# Stochastic Neural Networks for Causal Inference with Missing Confounders

**Yaxin Fang** [*]
Department of Anesthesiology, Perioperative and Pain Medicine
Stanford University
Palo Alto, CA 94305, USA
yxfang@stanford.edu

**Faming Liang**
Department of Statistics
Purdue University
West Lafayette, IN 47907, USA
fmliang@purdue.edu

## Abstract

Unmeasured confounding is a fundamental obstacle to causal inference from observational data. Latent-variable methods address this challenge by imputing unobserved confounders, yet many lack explicit model-based identification guarantees and are difficult to extend to richer causal structures. We propose Confounder Imputation with Stochastic Neural Networks (CI-StoNet), which parameterizes the conditional structure of a causal directed acyclic graph using a stochastic neural network and imputes latent confounders via adaptive stochastic-gradient Hamiltonian Monte Carlo. Under SUTVA and overlap, and assuming that the structural components of the data-generating process are well approximated by a capacity-controlled sparse deep neural network class, we establish model identification and consistent estimation of the mean potential outcome under a fixed intervention within this class. Although the latent confounder is identifiable only up to reparameterizations that preserve the joint treatment–outcome distribution, the causal estimand is invariant across this observationally equivalent class. We further characterize the effect of overlap on estimation accuracy. Empirical results on simulated and benchmark datasets demonstrate accurate performance, and the framework extends naturally to proxy-variable and multiple-cause settings with overlap diagnostics and bootstrap-based uncertainty quantification.

## 1 Introduction

Causal inference from observational studies is a topic of significant interest in fields such as genetics, economics, and social science. Under the potential outcome framework (Rubin, 1974), a fundamental condition for identifying causal effects is the strong ignorability condition (Rosenbaum & Rubin, 1983):

$$A \perp\!\!\!\perp \{Y(a) : a \in \mathcal{A}\} \mid Z, \tag{1}$$

where $A$ denotes the treatment variable taking values in the space $\mathcal{A}$, $Y(\cdot)$ denotes the outcome function, and $Z$ denotes confounders. A confounder refers to a variable that influences both the treatment and the outcome. The strong ignorability assumption requires that all confounders be observed. In observational studies, this condition is seldom fully met, thereby introducing the risk of substantial bias in causal effect estimation.

---

[*]The code of the experiments is available at: `https://github.com/nixay/Stochastic-Neural-Networks-for-Causal-Inference-with-Missing-Confounders`

One strategy to address the issue of missing confounders is to model them as latent variables. Wang & Blei (2018) proposed using a latent factor model to obtain a latent representation for multiple causes, enabling the capture of multiple-cause confounders under the assumption that no single-cause confounder exists. Kallus et al. (2018) tackled this problem under a proxy variable setting by leveraging the low-rank components of the proxy variables, obtained through matrix factorization, as an approximation to the true confounders. Louizos et al. (2017) also addressed the issue with proxy variables and introduced the causal effect variational autoencoder (CEVAE) to infer missing confounders from the observational distribution of the proxy, treatment, and outcome. These works represent significant advancements in causal inference using observational data; however, they have notable limitations. For instance, Imai & Jiang (2019) pointed out that Wang & Blei (2018) essentially models the substitute confounder as a deterministic function of treatments, leading it to converge to a function of the observed treatments rather than the true confounder. Kallus et al. (2018) focuses primarily on the linear regression setting, limiting its applicability to nonlinear models unless many proxies are available for a small number of latent variables. Rissanen & Marttinen (2021) examined the consistency of the causal effect estimator in Louizos et al. (2017) and showed that it fails to correctly estimate cause effects when the latent variable is misspecified or the data distribution is overly complex.

In this paper, we propose a latent variable imputation approach to address the issue of missing confounders. This new approach is built on the stochastic neural network (StoNet) (Sun & Liang, 2022; Liang et al., 2022) and sparse deep learning theory (Sun et al., 2022), effectively overcoming the limitations of existing approaches.

The core idea is to encode the causal DAG's Markov factorization in StoNet and to impute latent confounders through the resulting conditional distributions. Importantly, our theoretical guarantee is model-based: under SUTVA and overlap, and assuming the treatment and outcome mechanisms lie in a function class that can be well approximated by sparse DNNs, the causal functional $\mathbb{E}[Y(\boldsymbol{a})]$ is uniquely determined and is consistently estimable.

The StoNet is trained using an adaptive stochastic gradient MCMC algorithm (Liang et al., 2022; Deng et al., 2019), which allows for the simultaneous imputation of missing confounders and estimation of sparse StoNet parameters. We refer to the proposed approach as *Confounder Imputation with Stochastic Neural Networks* (CI-StoNet). In summary, it offers the following advantages in addressing the missing confounder issue:

(i) **Accurate Causal Effect Estimation:** This property is supported by StoNet's inherent ability to handle missing data, the consistency of sparse deep learning, and the convergence guarantee offered by the adaptive stochastic gradient MCMC algorithm.

(ii) **Complex nonlinear modeling.** CI-StoNet inherits the universal-approximation property of deep neural networks (DNNs), enabling effective modeling of complex nonlinear relationships across diverse applications.

(iii) **Structural flexibility.** The Markovian architecture of CI-StoNet provides structural flexibility for representing diverse dependency patterns in causal DAGs. It supports localized updates to each DNN module, promoting modular design and easy adaptation to varying causal relationships.

## 2 CI-StoNet for Missing Confounders

### 2.1 The CI-StoNet Approach

This section introduces the CI-StoNet approach or, more generally, a deep learning framework for performing causal inference in presence of missing confounders.

Consider the scenario of simple confounding, as depicted by Figure 1, which involves treatment $\boldsymbol{A} \in \{\boldsymbol{a}_1, \ldots, \boldsymbol{a}_m\}$, missing/latent confounders $\boldsymbol{Z}$, and an outcome $\boldsymbol{Y}$. The corresponding model is given by

$$\begin{aligned} \boldsymbol{A} &= g_1(\boldsymbol{Z}, \boldsymbol{e}_a), \\ \boldsymbol{Y} &= g_2(\boldsymbol{Z}, \boldsymbol{A}) + \boldsymbol{e}_y, \end{aligned} \quad (2)$$

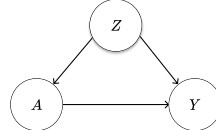

Figure 1: simple confounding

where $g_1(\cdot)$ and $g_2(\cdot)$ are unknown functions that can be nonlinear and highly complex, and $\boldsymbol{e}_a$ and $\boldsymbol{e}_y$ are random errors. In this paper, we assume $\boldsymbol{e}_y \sim N(0, \sigma_y^2 I_{d_y})$, where $d_y$ denotes the dimension of $\boldsymbol{Y}$. There is flexibility in specifying the distribution of $\boldsymbol{e}_a$. If $\boldsymbol{A}$ is continuous, then $\boldsymbol{e}_a$ can be assumed to follow a Gaussian distribution or any other continuous distribution. If $\boldsymbol{A}$ is mixed, the distribution of each component of $\boldsymbol{e}_a$ can be specified accordingly. This scenario has included multiple causes considered in Wang & Blei (2018), where $\boldsymbol{a}_i$ is a multi-dimensional vector, as a special case. Since $\boldsymbol{Z}$ is missing, we impute it from the conditional distribution:

$$\pi(\boldsymbol{Z}|\boldsymbol{A}, \boldsymbol{Y}) \propto \pi(\boldsymbol{Z})\pi(\boldsymbol{A}|\boldsymbol{Z})\pi(\boldsymbol{Y}|\boldsymbol{Z}, \boldsymbol{A}) \propto \pi(\boldsymbol{Z}|\boldsymbol{A})\pi(\boldsymbol{Y}|\boldsymbol{Z}, \boldsymbol{A}), \tag{3}$$

where $\pi(\cdot)$ denotes a distribution or conditional distribution in the appropriate context.

The latter part of Eq. (3) suggests that, mathematically, $\boldsymbol{A}$, $\boldsymbol{Z}$, and $\boldsymbol{Y}$ can be interpreted as the exogenous input, latent state and output of a stochastic model. Motivated by this view, we propose to perform the imputation using a CI-StoNet (see Figure 2 for its structure), formulated as:

$$\begin{aligned} \boldsymbol{Z} &= \mu_1(\boldsymbol{A}, \boldsymbol{\theta}_1) + \boldsymbol{e}_z, \\ \boldsymbol{Y} &= \mu_2(\boldsymbol{Z}, \boldsymbol{A}, \boldsymbol{\theta}_2) + \boldsymbol{e}_y, \end{aligned} \tag{4}$$

where $\mu_1(\cdot)$ and $\mu_2(\cdot)$ are two neural network functions, parameterized by $\boldsymbol{\theta}_1$ and $\boldsymbol{\theta}_2$, respectively; $\boldsymbol{e}_z \sim N(\boldsymbol{0}, \sigma_z^2 I_{d_z})$; and $\boldsymbol{e}_y$ is as de-

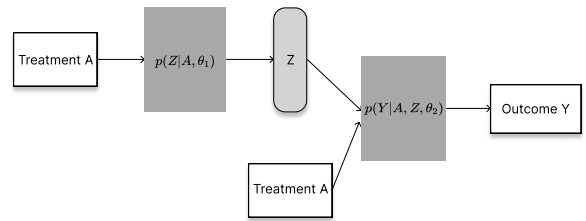

Figure 2: Diagram of CI-StoNet under simple confounding, where white rectangles represent variables from observed data; light-grey rounded-rectangles represent latent variable to impute; and dark-grey rectangles represent neural network modules to learn respective conditional distributions.

fined in (2). The two neural networks are interconnected through the latent variable $\boldsymbol{Z}$. Additionally, we impose the following assumptions on the models (2) and (4):

**Assumption 1.** *(i) $\boldsymbol{e}_a \perp\!\!\!\perp \boldsymbol{e}_y$, $\boldsymbol{e}_a \perp\!\!\!\perp \boldsymbol{Z}$, $\boldsymbol{e}_y \perp\!\!\!\perp (\boldsymbol{Z}, \boldsymbol{A})$; (ii) there exist sparse DNNs $\mu_1(\cdot)$ and $\mu_2(\cdot)$ such that (4) holds, $\boldsymbol{e}_z \sim N(\boldsymbol{0}, \sigma_z^2 I_{d_z})$, $\boldsymbol{e}_y \sim N(0, \sigma_y^2 I_{d_y})$, $\boldsymbol{e}_z \perp\!\!\!\perp \boldsymbol{e}_y$, and $\boldsymbol{e}_z \perp\!\!\!\perp \boldsymbol{A}$;*

Part (i) of Assumption 1 ensures that strong ignorability (1) holds. Part (ii) specifies the working model class by restricting the treatment and outcome mechanisms to a sparse DNN family (with $\boldsymbol{Z}$ random). This structural restriction enables tractable theoretical analysis and defines the pseudo-true parameter $\boldsymbol{\theta}^*$ that minimizes the KL divergence from the true data-generating law. See further details in 2.2.1.

Notably, the functional expression in (4) does not imply a causal mechanism $\boldsymbol{A} \to \boldsymbol{Z}$. For example, rain ($\boldsymbol{Z}$) causes a wetland ($\boldsymbol{A}$), but a wetland does not cause rain. Similarly, $\boldsymbol{Z}$ cannot be interpreted as a mediator due to the nonexistence of a causal mechanism $\boldsymbol{A} \to \boldsymbol{Z}$, although (4) has a mathematical structure similar to mediation models. For the time being, we assume that there is no mediator in the causal pathway between the treatment $\boldsymbol{A}$ and the outcome $\boldsymbol{Y}$, thereby ruling out any potential misinterpretation for the role of $\boldsymbol{Z}$. However, if a mediator does exist, issues related to the total causal effect estimation and the interpretation of $\boldsymbol{Z}$ will be addressed at the end of Section 2.2.

Under the missing data framework, the CI-StoNet can be trained by solving the following equation, which represents a Bayesian version of Fisher's identity (Song et al., 2020):

$$\nabla_{\boldsymbol{\theta}} \log \pi(\boldsymbol{\theta}|\boldsymbol{A}, \boldsymbol{Y}) = \int \nabla_{\boldsymbol{\theta}} \log \pi(\boldsymbol{\theta}|\boldsymbol{Z}, \boldsymbol{A}, \boldsymbol{Y})\pi(\boldsymbol{Z}|\boldsymbol{A}, \boldsymbol{Y}, \boldsymbol{\theta})d\boldsymbol{Z}, \tag{5}$$

where $\boldsymbol{\theta} = \{\boldsymbol{\theta}_1, \boldsymbol{\theta}_2\}$, $\boldsymbol{Z}$ is missing, $\pi(\boldsymbol{\theta}|\boldsymbol{Z}, \boldsymbol{A}, \boldsymbol{Y}) \propto \pi(\boldsymbol{\theta}_1)\pi(\boldsymbol{\theta}_2)\pi(\boldsymbol{Z}|\boldsymbol{A}, \boldsymbol{\theta}_1)\pi(\boldsymbol{Y}|\boldsymbol{Z}, \boldsymbol{A}, \boldsymbol{\theta}_2)$, $\pi(\boldsymbol{Z}|\boldsymbol{A}, \boldsymbol{Y}, \boldsymbol{\theta}) \propto \pi(\boldsymbol{Z}|\boldsymbol{A}, \boldsymbol{\theta}_1)\pi(\boldsymbol{Y}|\boldsymbol{Z}, \boldsymbol{A}, \boldsymbol{\theta}_2)$, and $\pi(\boldsymbol{\theta}_1)$ and $\pi(\boldsymbol{\theta}_2)$ denote the prior distributions imposed on $\boldsymbol{\theta}_1$ and $\boldsymbol{\theta}_2$, respectively. In this paper, we assume that the components of $\boldsymbol{\theta}$ are *a priori* independent and are subject to the following mixture Gaussian prior (Sun et al., 2022):

$$\pi(\boldsymbol{\theta}) = \prod_{i=1}^{K_n} (1 - \lambda_n)\phi(\boldsymbol{\theta}_i/\sigma_0) + \lambda_n \phi(\boldsymbol{\theta}_i/\sigma_1), \tag{6}$$

where $\lambda_n$ is the mixture proportion, $K_n$ is the total number of connections in the StoNet (i.e., the dimension of $\boldsymbol{\theta}$), $\phi(\cdot)$ represents the density function of the standard normal distribution, and $\sigma_0$ and $\sigma_1$ are the standard deviations of the two Gaussian components, respectively.

The identity (5) further suggests that the target equation

$$\nabla_{\boldsymbol{\theta}} \log \pi(\boldsymbol{\theta}|\boldsymbol{A}, \boldsymbol{Y}) = 0, \tag{7}$$

can be solved using an adaptive stochastic gradient MCMC algorithm, which iteratively alternates between latent variable imputation and parameter updates. In this paper, we employ the adaptive stochastic gradient Hamiltonian Monte Carlo (SGHMC) (Liang et al., 2022), as given in Algorithm 1, to solve equation (7).

---

**Algorithm 1:** Adaptive SGHMC

    0. Set the prior hyperparameters: $\lambda_n$, $\sigma_0$, and $\sigma_1$.

    1. (*Latent variable imputation*) Simulate $\boldsymbol{Z}$ from $\pi(\boldsymbol{Z}|\boldsymbol{A}, \boldsymbol{Y}, \boldsymbol{\theta})$ via Hamiltonian Monte Carlo updates:

$$\mathbf{v}^{(k+1)} = (1 - \epsilon_{k+1}\eta)\mathbf{v}^{(k)} + \epsilon_{k+1}\nabla_{\boldsymbol{Z}} \log \pi(\boldsymbol{Z}^{(k)}|\boldsymbol{A}, \boldsymbol{\theta}_1^{(k)}) + \epsilon_{k+1}\nabla_{\boldsymbol{Z}} \log \pi(\boldsymbol{Y}|\boldsymbol{Z}^{(k)}, \boldsymbol{A}, \boldsymbol{\theta}_2^{(k)}))$$
$$+ \sqrt{2\epsilon_{k+1}\eta}\mathbf{e}^{(k+1)},$$
$$\boldsymbol{Z}^{(k+1)} = \boldsymbol{Z}^{(k)} + \epsilon_t \mathbf{v}^{(k)},$$

    where $\boldsymbol{e}^{(k+1)} \sim N(0, I_{d_z})$, $d_z$ is the dimension of $\boldsymbol{Z}$, and $\epsilon_{k+1}$ is the learning rate.

    2. (*Parameter update*) Given $\boldsymbol{Z}^{(k+1)}$, update $\boldsymbol{\theta}_1$ and $\boldsymbol{\theta}_2$ separately:

$$\boldsymbol{\theta}_1^{(k+1)} = \boldsymbol{\theta}_1^{(k)} + \gamma_{k+1}\nabla_{\boldsymbol{\theta}_1} \log \pi(\boldsymbol{Z}^{(k+1)}|\boldsymbol{A}, \boldsymbol{\theta}_1^{(k)}) + \gamma_{k+1}\nabla_{\boldsymbol{\theta}_1} \log \pi(\boldsymbol{\theta}_1^{(k)}),$$
$$\boldsymbol{\theta}_2^{(k+1)} = \boldsymbol{\theta}_2^{(k)} + \gamma_{k+1}\nabla_{\boldsymbol{\theta}_2} \log \pi(\boldsymbol{Y}|\boldsymbol{Z}^{(k+1)}, \boldsymbol{A}, \boldsymbol{\theta}_2^{(k)}) + \gamma_{k+1}\nabla_{\boldsymbol{\theta}_2} \log \pi(\boldsymbol{\theta}_2^{(k)}).$$

---

In model (4), both $\sigma_z$ and $\sigma_y$ are scalar. They can be treated as hyperparameters to specify in simulations, while having minimal impact on the downstream inference. Notably, $\sigma_z$ is essentially non-identifiable in model (4), due to the universal approximation property of neural networks. In the inference stage, see equation (9), we provide a Bayesian estimator for $\sigma_z^2$ to facilitate imputation of missing confounders. Specifically, we impose an inverse gamma prior $\sigma_z^2 \sim \text{InvGamma}(\alpha, \beta)$, leading to the Bayesian estimator:

$$\hat{\sigma}_z^2 = \frac{\beta + \frac{1}{2}\sum_{j=1}^n (z_j - \mu_1(\boldsymbol{A}_j, \boldsymbol{\theta}_1))^2}{\frac{n}{2} + \alpha - 1}, \tag{8}$$

where we set $\alpha = \beta = 1$ for a flat prior, and $\boldsymbol{A}_j$ denotes the value of $\boldsymbol{A}$ in sample $j$. In simulations, its value can also be updated as in (8) along with iterations, while having minimal impact on the performance of the algorithm.

To enable causal inference, we introduce the following additional assumptions, which are standard conditions for causal effect identification:

**Assumption 2.**    *1.* **Stable unit treatment value assumption (SUTVA)***: the potential outcome of one subject are independent of the assigned treatment of another subject; that is, there is no interference between subjects and there is only a single version of each assigned treatment.*

    *2.* **Overlap***: The substitute confounder $\boldsymbol{Z}$ satisfies the overlap condition: $p(A \in \mathcal{A}|\boldsymbol{Z}) > 0$ for all sets $\mathcal{A}$ with positive measure, i.e., $p(\mathcal{A}) > 0$.*

Under Assumptions 1 and 2, the causal effect can be estimated in the following procedure:

**Causal effect estimation.**    After Algorithm 1 converges, with learned parameters $\hat{\boldsymbol{\theta}}^* = (\hat{\boldsymbol{\theta}}_1^*, \hat{\boldsymbol{\theta}}_2^*)$, for each observation $i$, draw $\mathcal{M}$ samples $\{\boldsymbol{z}_i^{(l)}\}_{l=1}^{\mathcal{M}}$ from $\pi(\boldsymbol{z} \mid \boldsymbol{a}_i; \hat{\boldsymbol{\theta}}_1^*)$. The expected

outcome $\mathbb{E}\{Y(\boldsymbol{a}) \mid \boldsymbol{\theta}^*\} = \int \mu_2(\boldsymbol{z}, \boldsymbol{a}; \boldsymbol{\theta}_2^*)\, \pi(\boldsymbol{z} \mid \boldsymbol{\theta}_1^*)\, d\boldsymbol{z}$ is then approximated by the Monte Carlo average

$$\widehat{\mathbb{E}(Y(\boldsymbol{a})|\hat{\boldsymbol{\theta}}^*)} = \frac{1}{n\mathcal{M}} \sum_{i=1}^{n} \sum_{l=1}^{\mathcal{M}} \mu_2(\boldsymbol{z}_i^{(l)}, \boldsymbol{a}, \hat{\boldsymbol{\theta}}_2^*). \tag{9}$$

A justification for the estimator is given in Appendix A5.3. The treatment effect estimator can then be derived accordingly.

## 2.2 Theoretical Guarantees

### 2.2.1 Model-based Identifiability

Following Rissanen & Marttinen (2021), we distinguish nonparametric identifiability from model-based identifiability. Nonparametric identifiability concerns whether the causal effect can be recovered solely from the observational distribution, whereas model-based identifiability asks whether the causal effect is unique within a restricted model class. Our results pertain to the latter. Concretely, we restrict the latent-variable model class by:

**Assumption 3.** *(Counfounding mechanism) The structural conditional mean functions $m_A(\boldsymbol{z}) = \mathbb{E}[\boldsymbol{A}|\boldsymbol{Z} = \boldsymbol{z}]$ and $m_Y(\boldsymbol{a}, \boldsymbol{z}) = \mathbb{E}[\boldsymbol{Y}|\boldsymbol{A} = \boldsymbol{a}, \boldsymbol{Z} = \boldsymbol{z}]$ belong to a function class $\mathcal{F}$ on a bounded domain such that there exists a sequence of sparse DNNs $\mu_{A,n}$ and $\mu_{Y,n}$ satisfying Assumption A10 with*

$$\|\mu_{A,n} - m_A\|_{L^2} + \|\mu_{Y,n} - m_Y\|_{L^2} \le \omega_n$$

*where $\omega_n$ is some sequence converging to 0 as $n \to \infty$*

This condition holds for many function classes $\mathcal{F}$, see a detailed discussion in Appendix A5.4. Under such restriction, let $\{P_{\boldsymbol{\theta}} : \boldsymbol{\theta} \in \Theta_n\}$ denote the family of observed-data laws induced by CI-StoNet under the working causal DAG and the sparse DNN specification (Assumption 1-(ii)). For the target functional $\psi_{\boldsymbol{\theta}}(\boldsymbol{a}) = \mathbb{E}_{P_{\boldsymbol{\theta}}}[Y(\boldsymbol{a})]$, we say $\psi_{\boldsymbol{\theta}}(\boldsymbol{a})$ is model-identifiable with respect to $P_{\boldsymbol{\theta}}$ if, whenever two parameter values $\boldsymbol{\theta}$, $\boldsymbol{\theta}'$, induce the same observed-data distribution (e.g. $p_{\boldsymbol{\theta}}(\boldsymbol{A}, \boldsymbol{Y})$ in simple confounding or $p_{\boldsymbol{\theta}}(\boldsymbol{A}, \boldsymbol{Y}, \boldsymbol{X})$ in proxy settings), it follows that $\psi_{\boldsymbol{\theta}}(\boldsymbol{a}) = \psi_{\boldsymbol{\theta}'}(\boldsymbol{a})$ for all $\boldsymbol{a}$.

In our setting, $\boldsymbol{Z}$ and $\boldsymbol{\theta}$ may be non-unique due to loss-invariant transformations, but the causal functional is invariant within each observational equivalence class; thus, Theorem A1(parameter recovery up to such transformations) and Theorem 1(consistency of $\hat{\psi}(\boldsymbol{a})$)), presented below, together yield model-based identification and consistent estimation of $\mathbb{E}[Y(\boldsymbol{a})]$ within the specified family. When the true mechanisms fall outside the sparse DNN family, Theorem 2 quantifies the resulting misspecification bias.

### 2.2.2 Theoretical Analysis

Let $P_0$ denote the true joint law of $(\boldsymbol{A}, \boldsymbol{Z}, \boldsymbol{Y})$, $P_{\boldsymbol{\theta}}$ be the distribution induced by the CI-StoNet with parameter $\boldsymbol{\theta}$. Define a pseudo-true StoNet parameter

$$\boldsymbol{\theta}^* = \arg\min_{\boldsymbol{\theta}} \mathrm{KL}(P_0, P_{\boldsymbol{\theta}}) := \arg\min_{\boldsymbol{\theta}} \int \log \frac{dP_0}{dP_{\boldsymbol{\theta}}} dP_0.$$

Let $\psi(P_0) = \int m_Y(\boldsymbol{a}, \boldsymbol{z}) p_{P_0}(\boldsymbol{z}) d\boldsymbol{z}$ be the true potential outcome, $\psi(P_{\boldsymbol{\theta}^*}) = \mathbb{E}(Y(\boldsymbol{a})|\boldsymbol{\theta}^*) = \int \mu_2(\boldsymbol{a}, \boldsymbol{z}) p_{P_{\boldsymbol{\theta}^*}}(\boldsymbol{z}) d\boldsymbol{z}$ the pseudo-true potential outcome, and $\psi(\hat{P}_{\boldsymbol{\theta}}) = \widehat{\mathbb{E}(Y(\boldsymbol{a})|\hat{\boldsymbol{\theta}}_n^*)} = \frac{1}{n\mathcal{M}} \sum_{i=1}^{n} \sum_{l=1}^{\mathcal{M}} \mu_2(\boldsymbol{z}_i^{(l)}, a, \hat{\boldsymbol{\theta}}_2^*)$. The error decomposition shows:

$$\|\psi(\hat{P}_{\boldsymbol{\theta}}) - \psi(P_0)\| \le \underbrace{\|\psi(\hat{P}_{\boldsymbol{\theta}}) - \psi(P_{\boldsymbol{\theta}^*})\|}_{\text{estimation error}} + \underbrace{\|\psi(P_0) - \psi(P_{\boldsymbol{\theta}^*})\|}_{\text{misspecification error}}.$$

**Theorem 1.** *(Estimation error)Suppose Assumptions 1-2 and the conditions in Lemma A1 and Theorem A1 (stated in Supplement A5) hold. Then*

$$\|\psi(\hat{P}_{\boldsymbol{\theta}}) - \psi(P_{\boldsymbol{\theta}^*})\| \xrightarrow{p} 0, \quad \text{as } \mathcal{M} \to \infty \text{ and } n \to \infty.$$

**Remark 1.** *As shown in the proof of Theorem 1, the consistency of the estimator (9) arises from the existence of the true sparse StoNet as well as the consistency of $\hat{\boldsymbol{\theta}}_n^*$. It is important to note that, due to the non-uniqueness of $\hat{\boldsymbol{\theta}}_n^*$ as discussed in Remark A1, the imputed latent confounders may differ from their true values. However, $\pi(\boldsymbol{z}|\boldsymbol{A},\hat{\boldsymbol{\theta}}_1^*)$ still serves as a consistent estimator (in terms of the density function) of $\pi(\boldsymbol{z}|\boldsymbol{A},\boldsymbol{\theta}_1^*)$, up to a loss-invariant transformation of $\hat{\boldsymbol{\theta}}_n^*$. Nevertheless, this does not affect the consistency of the estimator (9), which is a remarkable property.*

**Theorem 2.** *(Misspecification error) Fix a treatment level $a$. Let $P_0$ denote the true (complete-data) law of $(\boldsymbol{A},\boldsymbol{Z},\boldsymbol{Y})$, and let $\{P_{\boldsymbol{\theta}} : \boldsymbol{\theta} \in \Theta_n\}$ denote the family of (complete-data) laws induced by the CI-StoNet working model with the sparse-DNN architecture at sample size $n$. Assume that the structural conditional mean functions $m_A(\boldsymbol{z}) = \mathbb{E}_{P_0}[\boldsymbol{A} \mid \boldsymbol{Z} = \boldsymbol{z}]$ and $m_Y(\boldsymbol{a},\boldsymbol{z}) = \mathbb{E}_{P_0}[\boldsymbol{Y} \mid \boldsymbol{A} = \boldsymbol{a}, \boldsymbol{Z} = \boldsymbol{z}]$ satisfy Assumption 3, and AssumptionA10 with approximation rate $\omega_n$, and that the sparse-DNN architecture satisfies Assumption A13. Furthermore, assume the model outcome regression is uniformly bounded at any fixed treatment $a$: $\sup_{\boldsymbol{z}} |\mu_2(\boldsymbol{a},\boldsymbol{z};\boldsymbol{\theta}^*)| \leq C_{\mu_2}$.*

*Define the causal functional $\psi(P) := \mathbb{E}_P[Y(\boldsymbol{a})]$. In particular,*

$$\psi(P_0) = \int m_Y(\boldsymbol{a},\boldsymbol{z})\, p_{P_0}(\boldsymbol{z})\, d\boldsymbol{z}, \qquad \psi(P_{\boldsymbol{\theta}}) = \int \mu_2(\boldsymbol{a},\boldsymbol{z};\boldsymbol{\theta})\, p_{P_{\boldsymbol{\theta}}}(\boldsymbol{z}) d\boldsymbol{z}.$$

*Then there exist constants $C_1, C_2 < \infty$ (independent of $n$) such that*

$$KL(P_0, P_{\boldsymbol{\theta}^*}) \leq C_1\, \omega_n^2, \qquad \|\psi(P_0) - \psi(P_{\boldsymbol{\theta}^*})\| \leq C_2\, \omega_n.$$

A finite-sample error analysis for both the simple confounding case (Theorem A2) and the basic proxy case (Theorem A3) is provided in Supplement A5.5. The analysis shows how the estimation error bound of potential outcome worsens when there is bad overlap $P(\boldsymbol{A}|\boldsymbol{Z})$.

Notably, by approximating the treatment and outcome mechanisms using sparse DNNs, CI-StoNet imposes relatively mild structural restrictions on the confounding process. In contrast to Wang & Blei (2018), it does not rely on a "no single-cause confounder" assumption and thus accommodates both multi-cause and single-cause confounding within a unified framework. Moreover, unlike the variational autoencoder approach of Louizos et al. (2017), which may yield latent representations without consistency guarantees when large neural networks are used, CI-StoNet leverages sparse deep learning theory and Bayesian regularization to ensure parameter estimation consistency within the specified model class, thereby providing theoretically grounded causal effect estimation relative to the pseudo-true parameter.

Additionally, we note that the latent variable imputed in step (i) of Algorithm 1 cannot be used for causal effect estimation, as it may contain information related to colliders. Figure 3(a) illustrates this concept, where the collider variable $\boldsymbol{C}$ is influenced by both $\boldsymbol{A}$ and $\boldsymbol{Y}$. In step (i), we impute $\boldsymbol{Z}$ conditioned on both $\boldsymbol{A}$ and $\boldsymbol{Y}$. If a collider variable exists, the imputed latent variable may introduce spurious associations between $\boldsymbol{A}$ and $\boldsymbol{Y}$, potentially biasing the causal effect estimation. To mitigate this issue, we specifically impute the latent variables from $\pi(\boldsymbol{Z}|\boldsymbol{A},\hat{\boldsymbol{\theta}}_1^*)$, ensuring that any collider-related information is excluded from the analysis.

In Section 2.1, we assumed the absence of mediators to enable a clear interpretation of $\boldsymbol{Z}$ as a latent confounder. However, if a mediator $\boldsymbol{M}$ does exist, as illustrated in Figure 3(b), the imputed latent variable $\boldsymbol{Z}$ may inadvertently encapsulate information related to $\boldsymbol{M}$. Mathematically, the conditional distribution can be expressed as:

$$\pi(\boldsymbol{Y} \mid \boldsymbol{A},\boldsymbol{Z}) = \int \pi(\boldsymbol{Y} \mid \boldsymbol{A},\boldsymbol{Z},\boldsymbol{M})\,\pi(\boldsymbol{M} \mid \boldsymbol{A})\, d\boldsymbol{M},$$

which indicates that, without observing $\boldsymbol{M}$, its effect will be absorbed into $\boldsymbol{Z}$, making them

(a) Collider (b) Mediator

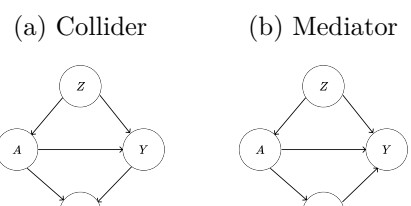

Figure 3: Other examples of causal structures: (a) existence of colliders, represented by $C$ ; (b) existence of mediators, represented by $M$.

statistically indistinguishable within the CI-StoNet framework. In this case, the treatment effect can still be estimated based on (9), where $\boldsymbol{Z}$ acts as a latent adjustment variable that facilitates estimation of the total causal effect. Although this precludes pathway-specific interpretations, it does not invalidate estimation of the total causal effect. If, however, the mediator $\boldsymbol{M}$ is known from domain knowledge or experimental design and there is no unmeasured confounding between $\boldsymbol{A}$ and $\boldsymbol{M}$, or between $\boldsymbol{M}$ and $\boldsymbol{Y}$, then the front-door criterion (Pearl, 2009) can be applied. In this case, $\boldsymbol{M}$ can be included as part of the latent confounder layer in the CI-StoNet to enable identification of the direct causal effect via front-door adjustment.

## 2.3 A Simulation Study

As a concept-proof example, we evaluated CI-StoNet using a simulation study with a nonlinear data-generating process for $\boldsymbol{A}$ and $\boldsymbol{Y}$ under both separable and non-separable confounding scenarios. We generate the latent confounders $Z_1, \ldots, Z_6$ as independent standard Gaussian random variables, and then draw $A_1, \ldots, A_9$ independently from the distribution, using inverse CDF:

$$p(\boldsymbol{a}_i|\boldsymbol{Z}) = \frac{\text{expit}(\xi(\boldsymbol{Z})\boldsymbol{a}_i)}{\int_{-1}^{1} \text{expit}(\xi(\boldsymbol{Z})\boldsymbol{a}_i)} 1_{\{-1 \leq \boldsymbol{a}_i \leq 1\}}, \quad i = 1, \ldots, 9,$$

where $\text{expit}(\xi(\boldsymbol{Z})\boldsymbol{a}_i) = \frac{\exp\{\xi(\boldsymbol{Z})\boldsymbol{a}_i\}}{1+\exp\{\xi(\boldsymbol{Z})\boldsymbol{a}_i\}}$, and $\xi(\boldsymbol{Z}) = \sum_{i=1}^{2} \beta_i \sin z_i + \sum_{j=3}^{4} \beta_j \cos z_j + \sum_{k=5}^{6} \frac{1}{1+\exp\{-\beta_k z_k + 0.5\}}$. We set $f_1(\boldsymbol{A}) = \boldsymbol{\theta}^T \boldsymbol{A}^{\otimes 2}$ and $f_2(\boldsymbol{A}) = \sum_{i<j} \boldsymbol{a}_i \boldsymbol{a}_j$, where $\boldsymbol{A}^{\otimes 2}$ represent an element-wise square operation, and generate $Y$ in two settings: (i) *Separable confounding.* the treatment and confounder impact the outcome separately: $Y = f_1(\boldsymbol{A}) - \boldsymbol{\theta}_0 f_2(\boldsymbol{A}) + \xi(\boldsymbol{Z}) + \epsilon$, where $\boldsymbol{\theta}_0 \sim U(-1,1)$ and $\epsilon \sim N(0,1)$. (ii) *Non-separable confounding.* there exists interaction between the treatment and confounder: $Y = f_1(\boldsymbol{A}) - \xi(\boldsymbol{Z})f_2(\boldsymbol{A}) + \xi(\boldsymbol{Z}) + \epsilon$, where $\epsilon \sim N(0,1)$.

For each setting, the experiment was conducted on 10 simulated datasets, each comprising 1000 training samples, 500 validation samples, and 500 test samples. The marginal treatment effects were calculated using the test set. Figure S1 compares the true and estimated marginal treatment effects across the 10 datasets. The plots show that most of the estimated marginal effects lie within half a standard deviation of the true marginal effects, indicating that CI-StoNet is able to estimate the marginal effect of each treatment with small bias.

## 3 Causal StoNet for Proxy Variables

In some applications, proxy variables for an unobserved confounder are available, though they may be noisy or only partially informative. Conditioning on such proxies can reduce, but not fully remove, confounding bias, making it natural to incorporate them as substitutes for the missing confounder. Kuroki & Pearl (2014) established identification results using matrix adjustment or spectral methods under specific assumptions on measurement error. The proximal causal inference framework of Tchetgen et al. (2020) and Miao et al. (2018) showed that causal effects can be identified when two types of proxies—treatment and outcome confounding proxies—are observed. In contrast, Louizos et al. (2017) proposed a variational autoencoder approach that leverages a basic proxy to learn a latent confounder representation.

Consider the causal structure with a basic proxy, as depicted in Figure 4(a). This causal structure suggests that $\boldsymbol{Z}$ can be imputed based on the following conditional distribution:

$$\pi(\boldsymbol{Z}|\boldsymbol{A},\boldsymbol{Y},\boldsymbol{X}) \propto \pi(\boldsymbol{Z})\pi(\boldsymbol{X}|\boldsymbol{Z})\pi(\boldsymbol{A}|\boldsymbol{Z})\pi(\boldsymbol{Y}|\boldsymbol{Z},\boldsymbol{A}) \propto \pi(\boldsymbol{Z}|\boldsymbol{X})\pi(\boldsymbol{A}|\boldsymbol{Z})\pi(\boldsymbol{Y}|\boldsymbol{Z},\boldsymbol{A}). \quad (10)$$

The decomposition of the conditional distribution (10) further suggests the StoNet structure:

$$\boldsymbol{Z} = \mu_1(\boldsymbol{X},\boldsymbol{\theta}_1) + \boldsymbol{e}_z, \quad \boldsymbol{A} = \mu_2(\boldsymbol{Z},\boldsymbol{\theta}_2,\boldsymbol{e}_a), \quad \boldsymbol{Y} = \mu_3(\boldsymbol{Z},\boldsymbol{A},\boldsymbol{\theta}_3) + \boldsymbol{e}_y, \quad (11)$$

where $\boldsymbol{e}_z$ and $\boldsymbol{e}_y$ are Gaussian random errors, while the form of $\boldsymbol{e}_a$ can be determined according to the types of treatments. These random errors are mutually independent and

(a)    (b)

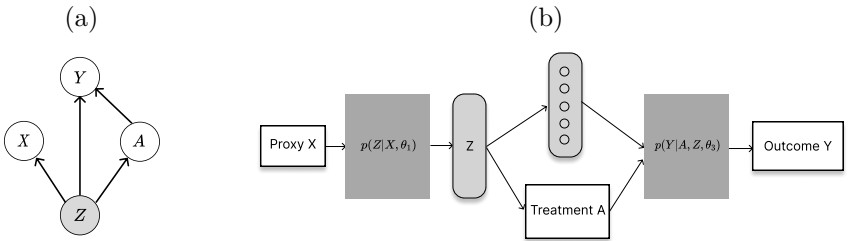

Figure 4: (a) Causal DAG: without dependence on the proxy; (b) Diagram of CI-StoNet under the proxy setting: white rectangles represent variables from observed data; light-grey rounded-rectangles represent hidden neurons; dark-grey rectangles represent network modules to learn respective conditional distributions.

are also independent of $\boldsymbol{X}$. Figure 4(b) illustrates the corresponding CI-StoNet structure. Alternatively, we can consider the following StoNet model:

$$\boldsymbol{Z} = \mu_1(\boldsymbol{X}, \boldsymbol{\theta}_1) + \boldsymbol{e}_z, \quad \boldsymbol{A} = \mu_2(\boldsymbol{Z}, \boldsymbol{\theta}_2) + \boldsymbol{e}_a, \quad \boldsymbol{Y} = \mu_3(\boldsymbol{Z}, \boldsymbol{A}, \boldsymbol{\theta}_3) + \boldsymbol{e}_y, \tag{12}$$

where $\boldsymbol{e}_z$, $\boldsymbol{e}_a$ and $\boldsymbol{e}_y$ are Gaussian random errors. Notably, the model (11) and the model (12) are asymptotically equivalent, even when $\boldsymbol{A}$ is a binary vector. In the binary case, their equivalence is supported by the result that, as shown in Liang (2003) and Duda et al. (2001), $\mu_2(\boldsymbol{Z}, \boldsymbol{\theta}_2)$ converges to the probability function $P(\boldsymbol{A} = \boldsymbol{1}|\boldsymbol{Z}, \boldsymbol{\theta}_2)$ as $n \to \infty$.

In this paper, we adopt the model (12) for computational simplicity. A gradient equation analogous to (7) can be constructed for the model. An adaptive SGHMC algorithm, similar to Algorithm 1, can be employed for its solution. Let $\{\boldsymbol{z}_i^{(l)} : l = 1, 2, \ldots, \mathcal{M}\}$ denote the samples simulated from $\pi(\boldsymbol{z}|\boldsymbol{x}_i, \hat{\boldsymbol{\theta}}_1^*)$. Then the expected outcome function $\mathbb{E}(Y(\boldsymbol{a}))$ can be estimated by the Monte Carlo average as

$$\widehat{\mathbb{E}(Y(\boldsymbol{a})|\hat{\boldsymbol{\theta}}_3^*)} = \frac{1}{n\mathcal{M}} \sum_{i=1}^{n} \sum_{l=1}^{\mathcal{M}} \mu_3(\boldsymbol{z}_i^{(l)}, \boldsymbol{a}, \hat{\boldsymbol{\theta}}_3^*). \tag{13}$$

### 3.1 Numerical Experiments

For simplicity, we consider a single binary treatment in our experiments. CI-StoNet is compared with the following baselines:

(i) Designed for average treatment effect (ATE): double selection estimator (**DSE**)(Belloni et al., 2014), approximate residual balancing estimator (**ARBE**) (Athey et al., 2018), targeted maximum likelihood estimator (**TMLE**) (van der Laan & Rubin, 2006), and deep orthogonal networks for unconfounded treatments (**DONUT**) (Hatt & Feuerriegel, 2021).

(ii) Designed for heterogeneous treatment effect: **X-learner** (Künzel et al., 2017), **Dragonnet**(Shi et al., 2019), causal multi-task deep ensemble (**CMDE**) (Jiang et al., 2023)), causal multi-task gaussian processes (**CMGP** (Alaa & van der Schaar, 2017)), causal effect variational autoencoder (**CEVAE**) (Louizos et al., 2017), generative adversarial networks (**GANITE**) (Yoon et al., 2018), and counterfactual regression net (**CFRNet** (Shalit et al., 2017)). For the baselines in part (ii), we use the code of Jiang et al. (2023) at GitHub.

For performance evaluation, we consider two metrics: (i) estimation accuracy of ATE, which is measured by the mean absolute error (MAE) of the ATE estimates; and (ii) estimation accuracy of CATE, which is measured by precision in estimation of heterogeneous effect (PEHE).

### 3.1.1 Simulated Examples

This example is designed to compare methods on problems with nonlinear treatment effect and nonlinear outcome function. We generated 10 datasets using the procedure as described in Section A1.2, with each dataset consisting of 2000 training samples, 500 validation samples, and 500 test samples. Table 1 shows that CI-StoNet provides accurate estimates for the heterogeneous treatment effect and outperforms the baselines.

Table 1: Comparison of different methods for estimation of heterogeneous treatment effects with proxy variables, where PEHE was computed over 10 datasets, 'In-sample PEHE' was computed with training and validation samples, and 'Out-of-sample PEHE' was computed with test samples.

|  | In-Sample PEHE | Out-of-Sample PEHE |
| --- | --- | --- |
| CI-StoNet | **0.3614(0.0328)** | **0.3731(0.0350)** |
| CMDE | 0.9019(0.0746) | 0.9059 (0.0699) |
| CMGP | 1.8823(0.0836) | 2.2116 (0.1682) |
| CEVAE | 0.6190(0.0350) | 0.6246 (0.0384) |
| Ganite | 1.2099(0.0558) | 1.1797 (0.0499) |
| X-learner-RF | 0.8308(0.0200) | 1.4272 (0.0132) |
| X-learner-Bart | 0.6489(0.0168) | 0.6570 (0.0151) |
| CFRNet-Wass | 1.7127(0.1668) | 1.7258 (0.1667) |
| CFRNet-MMD | 2.0238(0.0537) | 2.0250 (0.0582) |
| DragonNet | 0.4217(0.0356) | 0.4305 (0.0361) |

### 3.1.2 BENCHMARK DATASETS

We evaluated CI-StoNet on some benchmark datasets, including the Twins dataset and 10 datasets from Atlantic Causal Inference Conference (ACIC) 2019 Data Challenge. The results reported in Section A1.3 indicate that CI-StoNet outperforms the baselines.

## 4 CONCLUSION

By integrating StoNets with adaptive stochastic gradient MCMC, this paper develops a practical and theoretically grounded framework for causal inference with missing confounders. CI-StoNet encodes the dependence structure implied by the causal DAG and estimates parameters using sparse deep learning with Bayesian regularization. Although the latent confounder is identifiable only up to loss-invariant transformations, the induced causal effect is well-defined within the sparse StoNet class. The framework extends naturally to settings with multiple causes and proxy variables.

Within this framework, the causal functional is model-identified: observationally equivalent parameterizations yield the same causal effect. Moreover, the total error decomposes into statistical estimation error and misspecification error relative to the pseudo-true parameter, allowing explicit characterization of convergence and model approximation.

Despite its advantages, this study has some limitations. First, the structure and parameter estimation of CI-StoNet rely on correct specification of the underlying causal DAG. For example, in settings with multiple treatments, an unrecognized mediator may be absorbed into the learned substitute confounder, potentially biasing causal estimates, since the model is defined through the joint distribution of $(\boldsymbol{A}, \boldsymbol{Y}, \boldsymbol{Z})$. If the mediator is properly identified, however, the CI-StoNet structure can be modified accordingly to avoid this issue. Second, the current formulation of CI-StoNet does not provide principled, model-based uncertainty quantification for the causal effect estimator. Although the method does not inherently yield valid posterior intervals for the causal functional, we provide a practical post-processing procedure in Appendix A4 that constructs bootstrapped confidence intervals for the estimated causal effects. More formal uncertainty quantification could be achieved by adopting the original StoNet framework (Liang et al., 2022), which enables inference through its fully Bayesian formulation.

Finally, the Markovian structure of CI-StoNet affords substantial flexibility for modeling a wide range of causal structures. Section A2 extends CI-StoNet to two proxy-variable settings: (i) outcome depending on the proxy and (ii) treatment depending on the proxy. See that section for details.

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

# A Appendix

## A1 Supplementary Examples

### A1.1 Figures for the Simulation Study in Section 2.3

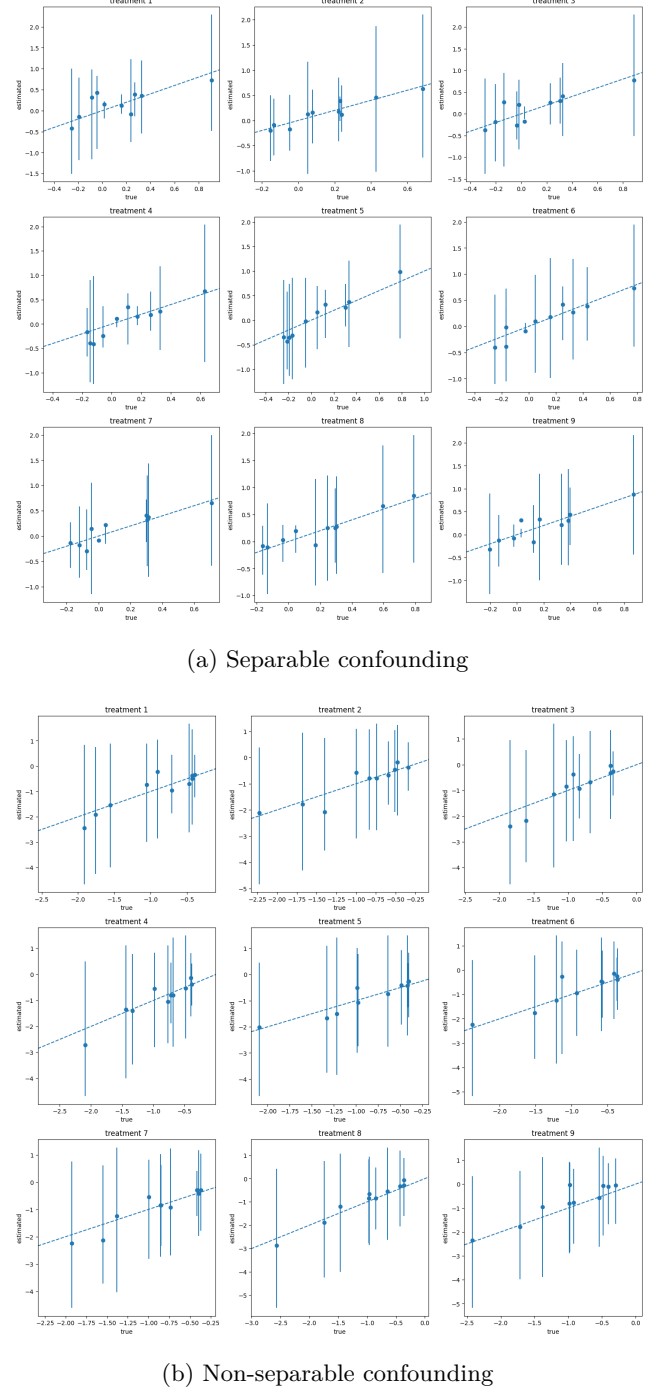

(a) Separable confounding

(b) Non-separable confounding

Figure S1: CI-StoNet results for the simulation study. Points denote the true marginal effects and error bars represent one standard error of the marginal effect estimator.

## A1.2 Simulated Examples

The simulated examples are used to compare the performance of different methods for problems with nonlinear treatment effect and nonlinear outcome function. We generated ten datasets using the following procedure, with each dataset comprising 2000 training samples, 500 validation samples, and 500 test samples.

1. Generate the confounder $\boldsymbol{z}_i = (z_{i,1}, \cdots, z_{i,5})$ independently from $N(0,1)$.

2. Generate $\gamma_i, r_{i,1}, \cdots, r_{i,100}$ independently from $N(\mu_i, 1)$ truncated in the interval $[-10, 10]$, with $\mu_i = \frac{1}{5} \sum_{k=1}^{5} z_{i,k}$. Set the proxy variable $\boldsymbol{x}_i = (x_{i,1}, \ldots, x_{i,100})$, with $x_{i,j} = \frac{\gamma_i + r_{i,j}}{\sqrt{2}}$, where $\boldsymbol{x}$ and $\boldsymbol{z}$ are dependent through $\mu$.

3. The propensity score $p(\boldsymbol{z}_i) = \frac{1}{4}(1 + \beta_{2,4}(\frac{1}{3}(\Phi(z_{i,1}) + \Phi(z_{i,3}) + \Phi(z_{i,5}))))$, where $\beta_{2,4}$ is the CDF of the beta distribution with shape parameters $(2, 4)$, and $\Phi$ denotes the CDF of the standard normal distribution. This ensures that $p(\boldsymbol{z}_i) \in [0.25, 0.5]$, thereby providing sufficient overlap. Treatment $A_i$ is hence generated from a Bernoulli distribution with the success probability $p(\boldsymbol{z}_i)$. Resampling from the treatment and control groups has been performed for ensuring that the dataset contains balanced samples for treatment group and control group.

4. To simulate the outcome, we set

$$\boldsymbol{y}_i = c(\boldsymbol{z}_i) + (\tau + \eta(\boldsymbol{z}_i))A_i + \sigma_y e_i,$$
$$c(\boldsymbol{z}_i) = \frac{5z_{i3}}{1 + z_{i4}^2} + 2z_{i5},$$

where $\eta(\boldsymbol{z}_i) = f(z_{i1})f(z_{i2}) - E(f(z_{i1})f(z_{i2}))$ and $f(w) = \frac{2}{1+\exp(-w+0.5))}$. That is, we set the treatment effect $\tau(\boldsymbol{z}_i) = \tau + \eta(\boldsymbol{z}_i)$, which is homogeneous for different individuals. We generated the samples under the setting $\tau = 3$, $\sigma_y = 0.25$, and $e_i \sim N(0,1)$.

## A1.3 Benchmark Datasets

We compare the performance of the proposed method on some benchmark datasets, including the Twins dataset and 10 datasets from Atlantic Causal Inference Conference (ACIC) 2019 Data Challenge.

**ACIC 2019 Datasets** We first worked on 10 ACIC 2019 datasets. This experiment focuses on comparing CI-StoNet with the baselines designed for ATE estimation. The results are summarized in Table S1, which indicates that CI-StoNet outperforms the baselines.

Table S1: ATE estimation across 10 ACIC 2019 datasets, where the number in the parentheses represents the standard deviation of the MAE, with additional benchmarks

| Method | In-Sample | Out-of-Sample |
|---|---|---|
| CI-StoNet | **0.0669 (0.0166)** | **0.0709 (0.0133)** |
| CMDE | 0.0802 (0.0166) | 0.0877 (0.0246) |
| CMGP | 0.1252 (0.0156) | 0.1349 (0.0170) |
| CEVAE | 0.0773 (0.0152) | 0.0875 (0.0154) |
| GANITE | 0.1622 (0.0390) | 0.1747 (0.0425) |
| X-Learner-RF | 0.1720 (0.0257) | 0.1903(0.0253) |
| X-Learner-BART | 0.0738 (0.0251) | 0.0817(0.0248) |
| CFRNet-Wass | 0.1024 (0.0241) | 0.1099(0.0256) |
| CFRNet-MMD | 0.1105 (0.0258) | 0.1208(0.0246) |

**Twins Data.** We analyzed a real-world dataset of twin births from 1989 to 1991 in the United States. The treatment variable is binary, with '1' denoting the heavier twin at birth.

The dataset contains 46 variables that include clinical information and socioeconomic status of parents, and we regard them as proxy variables for latent confounders. The outcome variable is binary, with '1' indicating twin mortality within the first year. We regard each twin-pair's records as potential outcomes, allowing us to find the true ATE. After data pre-processing, we obtained a dataset with 4,821 samples. In this final dataset, mortality rates for lighter and heavier twins are 16.9% and 14.42%, respectively, resulting in a true ATE of $-2.48\%$.

We conducted the experiment in three-fold cross validation, where we partitioned the dataset into three subsets, trained the model using two subsets and estimated the ATE using the remaining one. Table S2 (left panel) reports the averaged ATE over three folds and the standard deviation of the average. CI-StoNet yields a more stable ATE estimate (in RMSE) compared to the baseline methods.

Table S2: Comparison of different methods in average treatment effect (ATE) estimation for Twins data, where the number in the parentheses represents the standard deviation of the absolute error of ATE, and RMSE denotes the root mean squared error.

| Methods | With confounder *gestat10* | | Missing confounder *gestat10* | |
| --- | --- | --- | --- | --- |
| | Absolute Error of ATE | RMSE | Absolute Error of ATE | RMSE |
| CI-StoNet | 0.0099(0.0089) | **0.0133** | **0.0135(0.0071)** | **0.0153** |
| DSE | 0.0157(0.0176) | 0.0236 | 0.0211(0.0193) | 0.0286 |
| ARBE | 0.0152 (0.0201) | 0.0252 | 0.0168(0.0257) | 0.0307 |
| TMLE(Lasso) | 0.0855 (0.0599) | 0.1044 | 0.0932(0.0791) | 0.1222 |
| TMLE(ensemble) | 0.1042 (0.0779) | 0.1301 | 0.1238(0.0607) | 0.1379 |
| DONUT | 0.0490 (0.0128) | 0.0506 | 0.0490(0.0124) | 0.0505 |
| CMDE | 0.0108(0.0905) | 0.0911 | 0.0635(0.0905) | 0.1106 |
| CEVAE | 0.0249(**0.0002**) | 0.0249 | 0.0327(0.0633) | 0.0712 |
| Ganite | 0.3519 (0.1533) | 0.3838 | 0.4198(0.2278) | 0.4776 |
| X-learner-RF | **0.0056** (0.0257) | 0.0252 | 0.0157(0.0257) | 0.0301 |
| X-learner-Bart | 0.0194 (0.0192) | 0.0273 | 0.0251(0.0312) | 0.0400 |
| CFRNet-Wass | 0.0189 (0.0425) | 0.0465 | 0.0211(0.0254) | 0.0330 |
| CFRNet-MMD | 0.0439 (0.0146) | 0.0463 | 0.0619(0.0158) | 0.0639 |

Finally, to provide more convincing evidence that the proposed method performs well when confounders are missing, we conducted an experiment where a significant confounder, *gestat10* (gestational age), is intentionally omitted. In preprocessing the dataset, we followed Louizos et al. (2017) to focus on the same-sex twin pairs with birth weights less than 2 kg, and used the variable *gestat10* to generate "pseudo treatment assignments". Since *gestat10* is also an important factor for newborn mortality, it serves as a significant confounder. We removed *gestat10* from the dataset. The results in Table S2 (right panel) show that CI-StoNet exhibits robust performance in presence of missing confounders. In this scenario, it outperforms all baselines in both the absolute error of ATE and RMSE, indicating the superiority of CI-StoNet gained from latent confounder imputation.

## A2 Extension to other Causal Structures

The Markovian structure embedded in CI-StoNet provides it with great flexibility to model a wide range of causal structures. In this section, we extend CI-StoNet to handle other causal structures involving proxy variables. Specifically, we consider two scenarios: the outcome depending on the proxy, and the treatment depending on the proxy.

### A2.1 Outcome Depending on Proxy

When outcome depends on proxy, see Figure S2(a), the imputation of $\boldsymbol{Z}$ is based on the following decomposition:

$$\pi(\boldsymbol{Z}|\boldsymbol{A},\boldsymbol{Y},\boldsymbol{X}) \propto \pi(\boldsymbol{Z})\pi(\boldsymbol{X}|\boldsymbol{Z})\pi(\boldsymbol{A}|\boldsymbol{Z})\pi(\boldsymbol{Y}|\boldsymbol{Z},\boldsymbol{A},\boldsymbol{X}) \propto \pi(\boldsymbol{Z}|\boldsymbol{X})\pi(\boldsymbol{A}|\boldsymbol{Z})\pi(\boldsymbol{Y}|\boldsymbol{Z},\boldsymbol{A},\boldsymbol{X}).$$

Accordingly, the structure of the CI-StoNet can be arranged as follows:

$$
\begin{aligned}
\boldsymbol{Z} &= \mu_1(\boldsymbol{X}, \boldsymbol{\theta}_1) + \boldsymbol{e}_z, \\
\boldsymbol{A} &= \mu_2(\boldsymbol{Z}, \boldsymbol{\theta}_2) + \boldsymbol{e}_a, \\
\boldsymbol{Y} &= \mu_3(\boldsymbol{X}, \boldsymbol{Z}, \boldsymbol{A}, \boldsymbol{\theta}_3) + \boldsymbol{e}_y,
\end{aligned}
\tag{A1}
$$

where $\boldsymbol{e}_z$, $\boldsymbol{e}_a$, and $\boldsymbol{e}_y$ denote Gaussian random errors. The corresponding diagram is shown in Figure S2(b).

(a)                          (b)

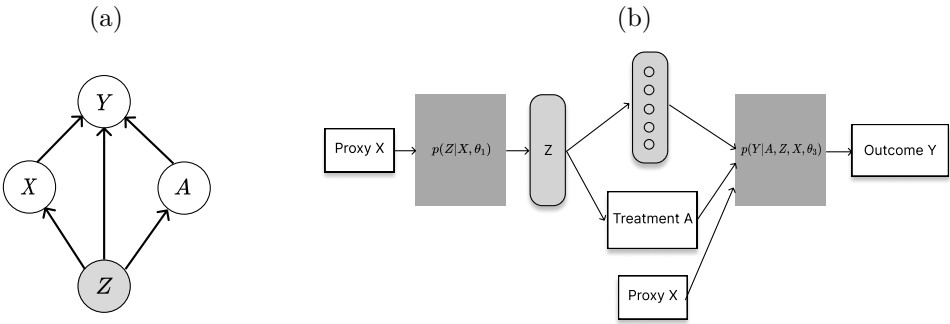

Figure S2: (a) Causal structure: outcome depends on the proxy; and (b) CI-StoNet structure for the scenario where outcome depends on the proxy.

## A2.2   Treatment Depending on Proxy

When the treatment depends on the proxy, see Figure S3(a), the imputation of $\boldsymbol{Z}$ is based on the decomposition:

$$
\pi(\boldsymbol{Z}|\boldsymbol{A}, \boldsymbol{Y}, \boldsymbol{X}) \propto \pi(\boldsymbol{Z})\pi(\boldsymbol{X}|\boldsymbol{Z})\pi(\boldsymbol{A}|\boldsymbol{Z}, \boldsymbol{X})\pi(\boldsymbol{Y}|\boldsymbol{Z}, \boldsymbol{A}) \propto \pi(\boldsymbol{Z}|\boldsymbol{X})\pi(\boldsymbol{A}|\boldsymbol{Z}, \boldsymbol{X})\pi(\boldsymbol{Y}|\boldsymbol{Z}, \boldsymbol{A}).
$$

The structure of the CI-StoNet can be arranged as follows:

$$
\begin{aligned}
\boldsymbol{Z} &= \mu_1(\boldsymbol{X}, \boldsymbol{\theta}_1) + \boldsymbol{e}_z, \\
\boldsymbol{A} &= \mu_2(\boldsymbol{Z}, \boldsymbol{X}, \boldsymbol{\theta}_2) + \boldsymbol{e}_a, \\
\boldsymbol{Y} &= \mu_3(\boldsymbol{Z}, \boldsymbol{A}, \boldsymbol{\theta}_3) + \boldsymbol{e}_y,
\end{aligned}
\tag{A2}
$$

where $\boldsymbol{e}_z$, $\boldsymbol{e}_a$, and $\boldsymbol{e}_y$ denote Gaussian random errors. The corresponding diagram is shown in Figure S3(b).

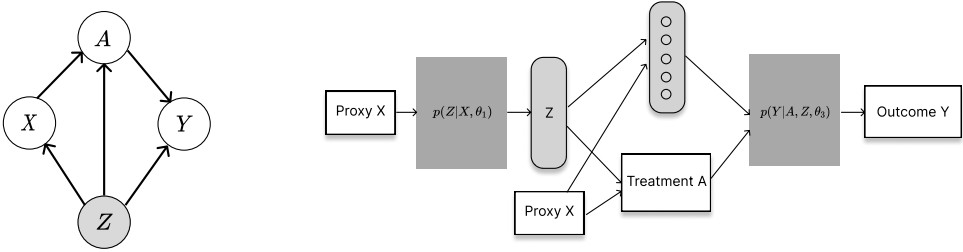

Figure S3: (a) Causal structure and (b) CI-StoNet structure for the scenario where treatment depends on the proxy.

Both models can be trained using an adaptive stochastic gradient MCMC algorithm, and the corresponding causal effects can be estimated based on the imputed confounders from $\pi(\boldsymbol{z}|\boldsymbol{X}, \hat{\boldsymbol{\theta}}_1^*)$.

For causal structures shown in Figures S2(a) and S3(a), $X$ is the proxy variable, $Z$ represents the missing confounder, and $A$ and $Y$ represents the treatment variable and outcome variable, respectively. The white nodes represent observed variables, while the light-grey node represent the unobserved variables. For the CI-StoNet structures shown in Figures S2(b) and S3(b), white rectangles represent variables from observed data; light-grey rounded-rectangles represent hidden neurons; and dark-grey rectangles represent network modules to learn respective conditional distributions.

## A3 EXPERIMENT ON DAG MISSPECIFICATION

We conducted an experiemnt where the true data generating process is treatment-depending proxy and outcome-depending proxy, but we used basic proxy to fit the data. Following is data-generating procedure

1. Confounder: $Z = (z_1, z_2, z_3, z_4, z_5) \sim N(0, I_5)$
2. Proxy: for $j \in \{1. \ldots, 50\}$, $X_j = h_1(Z) + \epsilon_X$, where $h_1(Z) = \sin(z_1) + 0.5z_2^2 + 0.3(z_3 + z_4 + z_5)$, $\epsilon_X \sim N(0, 0.5I_{50})$
3. Treatment: let $h_2(Z) = 0.7z_1 + 0.3z_2 - 0.2z_3$, $g(X) = \frac{1}{d_X}\sum_{j=1}^{d_X} |X_j|$. For outcome depending proxy and basic proxy, $A \sim \text{Bernoulli}(\text{expit}(h_2(Z)))$. For treatment-depending proxy, $A \sim \text{Bernoulli}(\text{expit}(h_2(Z) + 0.5g(X)))$.
4. Outcome: let $f(Z) = z_1^2 + 0.5z_2z_3$, $\tau(Z) = 3 + 0.5\sin(z_1)$, $q(X) = \frac{1}{d_X}\sum_{j=1}^{d_X} |X_j|$. For treatment-dependent proxy and basic proxy, $Y = f(Z) + A \cdot \tau(Z) + \epsilon_Y$. For outcome-dependent proxy, $Y = f(Z) + A \cdot \tau(Z) + \gamma q(X) + \epsilon_Y$. $\epsilon_Y \sim N(0, 0.5^2)$.

The true ATE is 3. For each scenario, 10 simulation datasets are generated, each dataset contains 2000 training samples, 500 validation samples, and 500 test samples. We use the basic proxy structure (equation 15) to model all three scenarios, where basic proxy scenario provides a correct baseline while outcome-depending and treatment-depending provide demonstration of DAG misspecification. In-sample MAE is calculated across train and validation set, and out-of-sample MAE is calculated across test set.

Table S3: Mean Absolute Error of ATE estimation under DAG misspecification

|  | In-Sample MAE | Out-of-Sample MAE |
|---|---|---|
| Basic Proxy | 0.0682 (0.0193) | 0.0702 (0.0207) |
| Outcome-depending proxy | 0.0705 (0.0188) | 0.0976 (0.0236) |
| Treatment-depending proxy | 0.0911 (0.0156) | 0.1239 (0.0178) |

## A4 DIAGNOSTICS AND UNCERTAINTY QUANTIFICATION

**Diagnostics on latent overlap** Although $Z$ is latent, we can still perform robustness diagnostics by sampling from its posterior. We show the procedure of diagnostics on latent overlap under simple confounding setting. Given that we have a overlap threshold $0 < \alpha < 1$, and a propensity score model $\hat{e}(A|Z)$, A simple stress test is:

1. For each unit $i$, draw $\boldsymbol{Z}_i^{(b)} \sim p_{\hat{\boldsymbol{\theta}}}(\boldsymbol{Z}_i | \boldsymbol{A}_i, \boldsymbol{Y}_i)$, b=1, ..., B
2. For each draw, compute propensities and a summary statistic of overlap

$$S_\alpha^{(b)} = \frac{1}{n}\sum_{i=1}^{n} \mathbf{1}\{\hat{e}_i^{(b)} < \alpha \text{ or } > 1 - \alpha\}$$

3. Report the mean $\overline{S}_\alpha = \frac{1}{B}\sum_b S_\alpha^{(b)}$ as diagnostic.

This diagnostic does not "verify" the latent overlap assumptions, which is theoretically untestable, but it allows practitioners to detect violations or fragility of overlap under posterior draws of $Z$, analogous to how overlap is assessed with fully observed covariates.

**Bootstrapped Confidence Interval** As explained in the imputation-regularized optimization paper (Liang et al., 2018a), the uncertainty in the "Monte Carlo average over imputed $Z$" reflects only the uncertainty due to the missing data (i.e., $Z$), and does not coincide with the uncertainty of the causal effect itself. Estimating the uncertainty of the causal effect requires additional methodology, such as bootstrapping or a method to compute the observed information matrix.

To derive a bootstrap confidence interval, consider the following computationally-light bootstrap procedure:

1. Fit CI-StoNet on the full dataset and save the final pruned parameters. The parameters $\hat{\boldsymbol{\theta}}$ are saved as the warm start for all bootstrap replicates.

2. Draw a bootstrap sample of size $n$, i.e., sampling indices $\{i_1^b, \ldots, i_n^b\} \sim$ i.i.d Uniform$(1, \ldots, n)$ with replacement, then construct the bootstrapped dataset $D^b = \{(A_{i_j^b}, X_{i_j^b}, Y_{i_j^b})\}_{j=1}^n$.

3. Warm-start model initialization. Initialize model parameters for this replicate by copying $\hat{\boldsymbol{\theta}}$.

4. Run a short SGD + SGHMC imputation phase to let the parameters to adjust to $D^b$.

5. Compute bootstrap ATE estimate. Use the trained parameter $\hat{\boldsymbol{\theta}}^b$, calculate $\tau^b = \hat{E}^b[Y(1)] - \hat{E}^b[Y(0)]$.

For the basic proxy scenario in DAG misspecification experiment (see details in A3), 100 iterations of such bootstrapping are conducted for each of the 10 generated dataset. Note that the true value of ATE is 3.

Table S4: Bootstrapped bounds for basic proxy setting

| DGP seed | $\hat{\tau}$ | $L_{\text{bootstrap}}$ | $U_{\text{bootstrap}}$ |
|---|---|---|---|
| 0 | 3.1365 | 3.0897 | 3.1581 |
| 1 | 3.0057 | 2.9758 | 3.0574 |
| 2 | 3.0074 | 2.9836 | 3.0720 |
| 3 | 3.1582 | 3.1066 | 3.2113 |
| 4 | 2.9718 | 2.9609 | 3.0404 |
| 5 | 3.0123 | 2.9774 | 3.0801 |
| 6 | 3.0515 | 3.0263 | 3.1113 |
| 7 | 3.0556 | 3.0128 | 3.0904 |
| 8 | 3.1846 | 3.1181 | 3.2119 |
| 9 | 3.0241 | 2.9787 | 3.0471 |

## A5 THEORETICAL PROOFS

The consistency of the estimator (9) can be established through several steps. First, we show that the estimator $\boldsymbol{\theta}^{(k)} = \{\boldsymbol{\theta}_1^{(k)}, \boldsymbol{\theta}_2^{(k)}\}$ obtained from Algorithm 1 converges in probability to a solution of (7), denoted by $\hat{\boldsymbol{\theta}}_n^*$ (see Lemma A1) A discussion on how to address the non-uniqueness of $\hat{\boldsymbol{\theta}}_n^*$ is followed. Next, we show that $\hat{\boldsymbol{\theta}}_n^*$ is a consistent estimator of $\boldsymbol{\theta}^*$, the true parameter vector of the sparse StoNet defined in (4) (see Theorem A1). Building on this result, we establish the consistency of the estimator (9) (see Theorem 1). These results are presented in the following.

### A5.1 CONVERGENCE OF $\boldsymbol{\theta}^{(k)}$

To train the CI-StoNet using the IRO algorithm, it requires that the full dataset is used at each iteration, making the algorithm difficult to scale up to large-scale neural networks. In

contrast, the adaptive SGHMC algorithm can use mini-batch data in parameter updating. As shown in Liang et al. (2022), the adaptive SGHMC algorithm solves equation (7) under the following conditions.

*Notations:* We let $\boldsymbol{D}$ denote a dataset of $n$ observations, and let $D_i$ denote the $i$-th observation of $\boldsymbol{D}$. For StoNet, $D_i$ has included both the input and output variables of the observation. For the CI-StoNet, $D_i$ includes the treatment and outcome, i.e., $D_i = \{\boldsymbol{A}_i, \boldsymbol{Y}_i\}$. For simplicity of notation, we re-denote the latent variable corresponding to $D_i$ by $Z_i$, and denote by $f_{D_i}(Z_i, \boldsymbol{\theta}) = -\log \pi(Z_i | D_i, \boldsymbol{\theta})$ the negative log-density function of $Z_i$. Let $\boldsymbol{Z} = (Z_1, Z_2, \ldots, Z_n)$, let $\boldsymbol{z} = (z_1, z_2, \ldots, z_n)$ be a realization of $\boldsymbol{Z}$, let $F_{\boldsymbol{D}}(\boldsymbol{Z}, \boldsymbol{\theta}) = \sum_{i=1}^{n} f_{D_i}(Z_i, \boldsymbol{\theta})$, and let $H(\boldsymbol{Z}, \boldsymbol{\theta}) = \nabla_{\boldsymbol{\theta}} \log \pi(\boldsymbol{Z} | \boldsymbol{A}, \boldsymbol{\theta})$. To study the convergence of the adaptive SGHMC algorithm presented in Algorithm 1, we make the following assumptions:

**Assumption A4.** *(i) (Boundedness) The function $F_{\boldsymbol{D}}(\cdot, \cdot)$ takes nonnegative real values, and there exist constants $A, B \geq 0$, such that $|F_{\boldsymbol{D}}(0, \boldsymbol{\theta}^*)| \leq A$, $\|\nabla_{\boldsymbol{Z}} F_{\boldsymbol{D}}(0, \boldsymbol{\theta}^*)\| \leq B$, $\|\nabla_{\boldsymbol{\theta}} F_{\boldsymbol{D}}(0, \boldsymbol{\theta}^*)\| \leq B$, and $\|H(0, \boldsymbol{\theta}^*)\| \leq B$.*

*(ii) (Smoothness) $F_{\boldsymbol{D}}(\cdot, \cdot)$ is $M$-smooth and $H(\cdot, \cdot)$ is $M$-Lipschitz: there exists some constant $M > 0$ such that for any $\boldsymbol{Z}, \boldsymbol{Z}' \in \mathbb{R}^{d_z}$ and any $\boldsymbol{\theta}, \boldsymbol{\theta}' \in \Theta$,*

$$\|\nabla_{\boldsymbol{Z}} F_{\boldsymbol{D}}(\boldsymbol{Z}, \boldsymbol{\theta}) - \nabla_{\boldsymbol{Z}} F_{\boldsymbol{D}}(\boldsymbol{Z}', \boldsymbol{\theta}')\| \leq M\|\boldsymbol{Z} - \boldsymbol{Z}'\| + M\|\boldsymbol{\theta} - \boldsymbol{\theta}'\|,$$
$$\|\nabla_{\boldsymbol{\theta}} F_{\boldsymbol{D}}(\boldsymbol{Z}, \boldsymbol{\theta}) - \nabla_{\boldsymbol{\theta}} F_{\boldsymbol{D}}(\boldsymbol{Z}', \boldsymbol{\theta}')\| \leq M\|\boldsymbol{Z} - \boldsymbol{Z}'\| + M\|\boldsymbol{\theta} - \boldsymbol{\theta}'\|,$$
$$\|H(\boldsymbol{Z}, \boldsymbol{\theta}) - H(\boldsymbol{Z}', \boldsymbol{\theta}')\| \leq M\|\boldsymbol{Z} - \boldsymbol{Z}'\| + M\|\boldsymbol{\theta} - \boldsymbol{\theta}'\|.$$

*(iii) (Dissipativity) For any $\boldsymbol{\theta} \in \Theta$, the function $F_{\boldsymbol{D}}(\cdot, \boldsymbol{\theta}^*)$ is $(m, b)$-dissipative: there exist some constants $m > \frac{1}{2}$ and $b \geq 0$ such that $\langle \boldsymbol{Z}, \nabla_{\boldsymbol{Z}} F_{\boldsymbol{D}}(\boldsymbol{Z}, \boldsymbol{\theta}^*) \rangle \geq m\|\boldsymbol{Z}\|^2 - b$.*

*(iv) (Gradient noise) There exists a constant $\varsigma \in [0, 1)$ such that for any $\boldsymbol{Z}$ and $\boldsymbol{\theta}$, $\mathbb{E}\|\nabla_{\boldsymbol{Z}} \hat{F}_{\boldsymbol{D}}(\boldsymbol{Z}, \boldsymbol{\theta}) - \nabla_{\boldsymbol{Z}} F_{\boldsymbol{D}}(\boldsymbol{Z}, \boldsymbol{\theta})\|^2 \leq 2\varsigma(M^2\|\boldsymbol{Z}\|^2 + M^2\|\boldsymbol{\theta} - \boldsymbol{\theta}^*\|^2 + B^2)$.*

**Assumption A5.** *The step size $\{\gamma_k\}_{k \in \mathbb{N}}$ is a positive decreasing sequence such that $\gamma_k \to 0$ and $\sum_{k=1}^{\infty} \gamma_k = \infty$. In addition, let $h(\boldsymbol{\theta}) = \mathbb{E}(H(\boldsymbol{Z}, \boldsymbol{\theta}))$, then there exists $\delta > 0$ such that for any $\boldsymbol{\theta} \in \Theta$, $\langle \boldsymbol{\theta} - \boldsymbol{\theta}^*, h(\boldsymbol{\theta}) \rangle \geq \delta\|\boldsymbol{\theta} - \boldsymbol{\theta}^*\|^2$, and $\liminf_{k \to \infty} 2\delta \frac{\gamma_k}{\gamma_{k+1}} + \frac{\gamma_{k+1} - \gamma_k}{\gamma_{k+1}^2} > 0$.*

**Assumption A6.** *(Solution of Poisson equation) For any $\boldsymbol{\theta} \in \Theta$, $\boldsymbol{z} \in \mathfrak{Z}$, and a function $V(\boldsymbol{z}) = 1 + \|\boldsymbol{z}\|$, there exists a function $\mu_{\boldsymbol{\theta}}$ on $\mathfrak{Z}$ that solves the Poisson equation $\mu_{\boldsymbol{\theta}}(\boldsymbol{z}) - \mathcal{T}_{\boldsymbol{\theta}} \mu_{\boldsymbol{\theta}}(\boldsymbol{z}) = H(\boldsymbol{\theta}, \boldsymbol{z}) - h(\boldsymbol{\theta})$, where $\mathcal{T}_{\boldsymbol{\theta}}$ denotes a probability transition kernel with $\mathcal{T}_{\boldsymbol{\theta}} \mu_{\boldsymbol{\theta}}(\boldsymbol{z}) = \int_{\mathfrak{Z}} \mu_{\boldsymbol{\theta}}(\boldsymbol{z}') \mathcal{T}_{\boldsymbol{\theta}}(\boldsymbol{z}, \boldsymbol{z}') d\boldsymbol{z}'$, such that*

$$H(\boldsymbol{\theta}_k, \boldsymbol{z}_{k+1}) = h(\boldsymbol{\theta}_k) + \mu_{\boldsymbol{\theta}_k}(\boldsymbol{z}_{k+1}) - \mathcal{T}_{\boldsymbol{\theta}_k} \mu_{\boldsymbol{\theta}_k}(\boldsymbol{z}_{k+1}), \quad k = 1, 2, \ldots. \tag{A3}$$

*Moreover, for all $\boldsymbol{\theta}, \boldsymbol{\theta}' \in \Theta$ and $\boldsymbol{z} \in \mathfrak{Z}$, we have $\|\mu_{\boldsymbol{\theta}}(\boldsymbol{z}) - \mu_{\boldsymbol{\theta}'}(\boldsymbol{z})\| \leq \varsigma_1 \|\boldsymbol{\theta} - \boldsymbol{\theta}'\| V(\boldsymbol{z})$ and $\|\mu_{\boldsymbol{\theta}}(\boldsymbol{z})\| \leq \varsigma_2 V(\boldsymbol{z})$ for some constants $\varsigma_1 > 0$ and $\varsigma_2 > 0$.*

**Lemma A1.** *(Theorem S1, Liang et al. (2022)) Suppose Assumptions A4-A6 hold. For Algorithm 1, if we set $\epsilon_k = C_\epsilon/(c_e + k^\alpha)$ and $\gamma_k = C_\gamma/(c_g + k^\alpha)$ for some constants $\alpha \in (0, 1)$, $C_\epsilon > 0$, $C_\gamma > 0$, $c_e \geq 0$ and $c_g \geq 0$, then there exists an iteration $k_0$ and a constant $\Lambda_0 > 0$ such that for any $k > k_0$,*

$$\mathbb{E}(\|\boldsymbol{\theta}^{(k)} - \hat{\boldsymbol{\theta}}_n^*\|^2) \leq \Lambda_0 \gamma_k, \tag{A4}$$

*where $\hat{\boldsymbol{\theta}}_n^*$ denotes a solution to Eq. (7), i.e., $\hat{\boldsymbol{\theta}}_n^* \in \mathcal{L} = \{\boldsymbol{\theta} : \nabla_{\boldsymbol{\theta}} \log \pi(\boldsymbol{\theta} | \boldsymbol{A}, \boldsymbol{Y}) = 0\}$; and $\Lambda_0 = \Lambda_0' + 6\sqrt{6} C_{\boldsymbol{\theta}}^{1/2}((3M^2 + \varsigma_2)C_{\boldsymbol{Z}} + 3M^2 C_{\boldsymbol{\theta}} + 3B^2 + \varsigma_2^2)^{1/2}$ for some positive constants $\Lambda_0'$, $C_{\boldsymbol{\theta}}$, and $C_{\boldsymbol{Z}}$.*

*Proof.* Lemma A1 is a restatement of Theorem S1 of Liang et al. (2022), and its proof is thus omitted. □

Refer to Lemma S1 of Liang et al. (2022) for the derivation of the constants $C_{\boldsymbol{\theta}}$ and $C_{\boldsymbol{Z}}$, which indicate the dependence of the convergence of $\boldsymbol{\theta}^{(k)}$ on the structure of the StoNet (4). As a consequence of the $l_2$-convergence (A4), we immediately have $\|\boldsymbol{\theta}^{(k)} - \hat{\boldsymbol{\theta}}_n^*\| \xrightarrow{p} 0$ as $k \to \infty$, where $\xrightarrow{p}$ denotes convergence in probability.

**Remark A1.** *For neural networks, it is known that their loss function is invariant under certain transformations of the connection weights, such as reordering hidden neurons within a layer or jointly changing the signs or scales of specific weights and biases, refer to, e.g., Liang et al. (2018b) and Sun et al. (2022) for detailed discussions. As a result, the solution $\hat{\boldsymbol{\theta}}_n^*$ is not unique, and all such solutions can be viewed as belonging to an equivalence class of unique solutions, defined by loss-invariant transformations. This equivalence class forms a reduced representation of the parameter space, where each member corresponds to a distinct network (i.e., not transformable into another via loss-invariant operations) and may have a different loss value. The consistency results established in this paper apply specifically to this reduced space of neural networks.*

## A5.2 CONSISTENCY OF $\hat{\boldsymbol{\theta}}_n^*$

### A5.2.1 CONSISTENCY OF THE IRO ALGORITHM

**The IRO Algorithm** The IRO algorithm (Liang et al., 2018a) starts with an initial weight setting $\hat{\boldsymbol{\theta}}^{(0)} = (\hat{\boldsymbol{\theta}}_1^{(0)}, \hat{\boldsymbol{\theta}}_2^{(0)})$ and then iterates between the imputation of latent confounders and regularized optimization for parameter updating:

- **Imputation:** simulate $\boldsymbol{z}_i^{(t+1)}$ from the predictive distribution:

$$\pi(\boldsymbol{z}_i \mid \boldsymbol{y}_i, \boldsymbol{a}_i, \hat{\boldsymbol{\theta}}^{(t)}, \boldsymbol{\sigma}_{CI}^2) \propto \pi(\boldsymbol{z}_i \mid \boldsymbol{a}_i, \hat{\boldsymbol{\theta}}_1^{(t)}, \sigma_z^2)\pi(\boldsymbol{y}_i \mid \boldsymbol{z}_i, \boldsymbol{a}_i, \hat{\boldsymbol{\theta}}_2^{(t)}, \sigma_y^2)$$

  where $t$ indexes iterations, and $\boldsymbol{\sigma}_{CI}^2 = (\sigma_z^2, \sigma_y^2)$.

- **Regularized optimization:** Given the pseudo-complete data $\{(\boldsymbol{y}_i, \boldsymbol{z}_i^{(t+1)}, \boldsymbol{a}_i) : i = 1, 2, \ldots, n\}$, update $\hat{\boldsymbol{\theta}}^{(t+1)}$ by maximizing the penalized log-likelihood function as follows:

$$\hat{\boldsymbol{\theta}}^{(t+1)} = \arg\max_{\boldsymbol{\theta}}\big\{\frac{1}{n}\sum_{i=1}^{n}\log\pi(\boldsymbol{y}_i, \boldsymbol{z}_i^{(t+1)}|\boldsymbol{a}_i, \boldsymbol{\theta}, \boldsymbol{\sigma}_{CI}^2) - \frac{1}{n}\log P_{\lambda_n}(\boldsymbol{\theta})\big\}. \quad (A5)$$

The penalty function $\frac{1}{n}\log P_{\lambda_n}(\boldsymbol{\theta})$ satisfies some conditions (see Assumption A9) such that $\hat{\boldsymbol{\theta}}^{(t+1)}$ forms a consistent estimator, uniformly over iterations, for the working parameter

$$\begin{aligned}
\boldsymbol{\theta}_*^{(t+1)} &= \arg\max_{\boldsymbol{\theta}}\mathbb{E}_{\hat{\boldsymbol{\theta}}^{(t)}}\log\pi(\boldsymbol{y}, \boldsymbol{z}|\boldsymbol{a}, \boldsymbol{\theta}, \boldsymbol{\sigma}_{CI}^2) \\
&= \arg\max_{\boldsymbol{\theta}}\int\log\pi(\boldsymbol{y}, \boldsymbol{z}|\boldsymbol{a}, \boldsymbol{\theta}, \boldsymbol{\sigma}_{CI}^2)\pi(\boldsymbol{z} \mid \boldsymbol{y}, \boldsymbol{a}, \hat{\boldsymbol{\theta}}^{(t)}, \sigma_z^2)\pi(\boldsymbol{y} \mid \boldsymbol{a}, \boldsymbol{\theta}^*, \sigma_y^2)d\boldsymbol{z}d\boldsymbol{y},
\end{aligned} \quad (A6)$$

where $\boldsymbol{\theta}^*$ denotes the true parameter value of the CI-StoNet model.

**Consistency of Parameter Estimation** The main proof for the consistency of parameter estimation is built on the theoretical framework developed in Liang et al. (2018a). Let $\tilde{\boldsymbol{x}} = (\boldsymbol{A}, \boldsymbol{Y}, \boldsymbol{Z})$ be the complete data, which is a collection of observed variable and latent variables. Define

$$G_n(\boldsymbol{\theta} \mid \hat{\boldsymbol{\theta}}^{(t)}) = \int\log\pi(\boldsymbol{y}, \boldsymbol{z}|\boldsymbol{a}, \boldsymbol{\theta}, \boldsymbol{\sigma}_{CI}^2)\pi(\boldsymbol{z} \mid \boldsymbol{y}, \boldsymbol{a}, \hat{\boldsymbol{\theta}}^{(t)}, \sigma_z^2)\pi(\boldsymbol{y} \mid \boldsymbol{a}, \boldsymbol{\theta}^*, \sigma_y^2)d\boldsymbol{z}d\boldsymbol{y},$$

$$\hat{G}_n(\boldsymbol{\theta} \mid \tilde{\boldsymbol{x}}, \hat{\boldsymbol{\theta}}^{(t)}) = \frac{1}{n}\sum_{i=1}^{n}\log\pi(\boldsymbol{y}_i, \boldsymbol{z}_i|\boldsymbol{a}_i, \boldsymbol{\theta}, \boldsymbol{\sigma}_{CI}^2), \quad \boldsymbol{z}_i \sim \pi(\boldsymbol{z}|\boldsymbol{y}_i, \boldsymbol{a}_i, \hat{\boldsymbol{\theta}}^{(t)}, \sigma_z^2),$$

**Lemma A2.** *(Theorem 1; Liang et al. (2018a)) Let $T$ denote the total number of iterations of the IRO algorithm. Under mild regularity conditions (See Assumptions 1-3 in Liang et al. (2018a)), the following uniform law of large numbers holds for any $T$, with $\log(T) = o(n)$:*

$$\sup_{\hat{\boldsymbol{\theta}}^{(t)}\in\boldsymbol{\theta}^T}\sup_{\boldsymbol{\theta}\in\Theta}\left|\hat{G}_n(\boldsymbol{\theta} \mid \tilde{\boldsymbol{x}}, \hat{\boldsymbol{\theta}}^{(t)}) - G_n(\boldsymbol{\theta} \mid \hat{\boldsymbol{\theta}}^{(t)})\right| \xrightarrow{p} 0, \quad (A7)$$

*as the sample size $n \to \infty$.*

**Assumption A7.** *For each $t = 1, 2, \ldots, T$, $G_n(\boldsymbol{\theta} \mid \hat{\boldsymbol{\theta}}^{(t)})$ has a unique maximum (up to loss-invariant transformations) at $\boldsymbol{\theta}_*^{(t)}$; for any $\epsilon > 0$, $\sup_{\boldsymbol{\theta} \in \Theta \backslash B_t(\epsilon)} G_n(\boldsymbol{\theta} \mid \hat{\boldsymbol{\theta}}^{(t)})$ exists, where $B_t(\epsilon) = \{\boldsymbol{\theta} \in \Theta : \|\boldsymbol{\theta} - \boldsymbol{\theta}_*^{(t)}\| < \epsilon\}$. Let $\delta_t = G_n(\boldsymbol{\theta}_*^{(t)} \mid \hat{\boldsymbol{\theta}}^{(t)}) - \sup_{\boldsymbol{\theta} \in \Theta \backslash B_t(\epsilon)} G_n(\boldsymbol{\theta} \mid \hat{\boldsymbol{\theta}}^{(t)})$, $\delta = \min_{t \in \{1,2,\ldots,T\}} \delta_t > 0$ holds.*

Assumption A7 restricts the shape of $G_n(\boldsymbol{\theta}|\hat{\boldsymbol{\theta}}^{(t)})$ around the global maximizer, ensuring that it is neither discontinuous nor too flat. Given the nonidentifiability of neural network models, Assumption A7 implicitly assumes that each $\boldsymbol{\theta}$ is unique up to loss-invariant transformations, such as reordering the hidden neurons within the same layer or simultaneously altering the signs or scales of certain weights and biases, see e.g., Liang et al. (2018b) and Sun et al. (2022) for further discussions. Alternatively, the optimal solutions can be considered as belonging to an equivalence class, subject to appropriate loss-invariant transformations, with the uniqueness assumption applying to this equivalence class.

Furthermore, consider the mapping $M(\boldsymbol{\theta})$ defined by

$$M(\boldsymbol{\theta}) = \arg\max_{\boldsymbol{\theta}'} \mathbb{E}_{\boldsymbol{\theta}} \log \pi(\boldsymbol{Y}, \boldsymbol{Z}|\boldsymbol{a}, \boldsymbol{\theta}', \boldsymbol{\sigma}_{CI}^2).$$

As argued in Liang et al. (2018a) and Nielsen (2000), it is reasonable to assume that the mapping is a contraction, as a recursive application of the mapping, i.e., setting

$$\hat{\boldsymbol{\theta}}^{(t+1)} = \boldsymbol{\theta}_*^{(t+1)} = M(\hat{\boldsymbol{\theta}}^{(t)}),$$

leads to a monotone increase of the target expectations $\mathbb{E}_{\hat{\boldsymbol{\theta}}^{(t)}} \log \pi(\boldsymbol{Y}, \boldsymbol{Z}|\boldsymbol{a}, \boldsymbol{\theta}, \boldsymbol{\sigma}_{CI}^2)$ for $t = 1, 2, \ldots$.

**Assumption A8.** *The mapping $M(\boldsymbol{\theta})$ is differentiable. Let $\rho_n(\boldsymbol{\theta})$ be the largest singular value of $\partial M(\boldsymbol{\theta})/\partial \boldsymbol{\theta}$. There exists a number $\rho^* < 1$ such that $\rho_n(\boldsymbol{\theta}) \leq \rho^*$ for all $\boldsymbol{\theta} \in \Theta$ for sufficiently large $n$ and almost every observed sequence of $(\boldsymbol{A}, \boldsymbol{Y})$.*

**Assumption A9.** *The penalty function $\frac{1}{n} \log P_{\lambda_n}(\boldsymbol{\theta})$ converges to 0 uniformly over the set $\{\boldsymbol{\theta}_*^{(t)} : t = 1, 2, \ldots, T\}$ as $n \to \infty$, where $\lambda_n$ is a regularization parameter and its value can depend on the sample size $n$.*

**Lemma A3.** *(Theorem 4; Liang et al. (2018a)) Suppose the conditions of Lemma A2, Assumptions A7-A9 hold, and $\sup_{n,t} \mathbb{E}\|\hat{\boldsymbol{\theta}}_n^{(t)}\| < \infty$ hold. Then for sufficiently large $t$ and almost every $(\boldsymbol{A}, \boldsymbol{Y})$-sequence, $\|\hat{\boldsymbol{\theta}}_n^{(t)} - \boldsymbol{\theta}^*\| \xrightarrow{p} 0$, as $n \to \infty$.*

### A5.2.2 VERIFICATION OF ASSUMPTION A9

To verify Assumption A9, we prove the following lemma.

**Lemma A4.** *Let $\boldsymbol{\theta} = (\boldsymbol{\theta}_1, \boldsymbol{\theta}_2, \ldots, \boldsymbol{\theta}_{K_n})^T$. Suppose that all components of $\boldsymbol{\theta}$ are a priori independent and they are subject to the following mixture Gaussian prior (6). Suppose $K_n \succ n$, $\boldsymbol{\theta}$ is sparse at a level of $m_n \prec \frac{n}{c \log(K_n/n)}$ for some constant $c > 1$, and $\min\{|\boldsymbol{\theta}_i| : \boldsymbol{\theta}_i \neq 0, i = 1, 2, \ldots, K_n\} > \delta_n$ for some constant $\delta_n = o(1)$. If we set $\sigma_1 = O(1)$ and set $(\lambda_n, \sigma_0)$ to satisfy the conditions:*

$$(\frac{n}{K_n})^c \prec \lambda_n \prec \frac{n}{K_n},$$
$$(\frac{n}{K_n})^c \prec \sigma_0 \prec \min\left\{1 - \frac{n}{K_n}, \frac{\delta_n}{\sqrt{c\log(K_n) - (c-1)\log(n)}}\right\}, \tag{A8}$$

*then the following result holds:*

$$\frac{1}{n}\left|\log \pi(\boldsymbol{\theta}) + K_n \log\left(\sqrt{2\pi}\sigma_0\right)\right| \to 0, \quad \text{as } n \to \infty. \tag{A9}$$

*Proof.* A straightforward calculation shows that

$$|\log \pi(\boldsymbol{\theta}) + K_n \log(\sigma_0)| \lesssim K_n |\log(1 - \lambda_n)| + (K_n - m_n)\frac{\sigma_0 \lambda_n}{\sigma_1(1 - \lambda_n)} + m_n \left|\log\left(\frac{\sigma_0 \lambda_n}{1 - \lambda_n}\right)\right|$$

$$- m_n \frac{\delta_n^2}{2\sigma_1^2} + \frac{m_n(1 - \lambda_n)\sigma_1}{\lambda_n \sigma_0} e^{-\frac{\delta_n^2}{2}(\frac{1}{\sigma_0^2} - \frac{1}{\sigma_1^2})}.$$

To ensure $K_n |\log(1 - \lambda)| \prec n$, we set

$$\lambda_n \prec 1 - e^{-n/K_n} \asymp \frac{n}{K_n}. \tag{A10}$$

To ensure $m_n \left|\log\left(\frac{\sigma_0 \lambda_n}{1 - \lambda_n}\right)\right| \prec n$, we set

$$\sigma_0 \succ (\frac{n}{K_n})^c \succ e^{-n/m_n}, \quad \lambda_n \succ (\frac{n}{K_n})^c \succ e^{-n/m_n}. \tag{A11}$$

To ensure $(K_n - m_n)\frac{\sigma_0 \lambda_n}{\sigma_1(1 - \lambda_n)} \prec n$, we set

$$\sigma_0 \prec 1 - \frac{n}{K_n} \prec \frac{n}{K_n}\frac{(1 - \lambda_n)}{\lambda_n}. \tag{A12}$$

To ensure $\frac{m_n(1 - \lambda_n)\sigma_1}{\lambda_n \sigma_0} e^{-\frac{\delta_n^2}{2}(\frac{1}{\sigma_0^2} - \frac{1}{\sigma_1^2})} \prec n$, we set

$$\sigma_0 \prec \frac{\delta_n}{\sqrt{c \log(K_n) - (c - 1)\log(n)}} \prec \frac{\delta_n}{\sqrt{|\log(n\lambda_n/m_n)|}}. \tag{A13}$$

Since $\delta_n \prec o(1)$ and $m_n \prec n$, we have $m_n \frac{\delta_n^2}{2\sigma_1^2} \prec n$.

As a summary of (A10)-(A13), we can set $(\lambda_n, \sigma_0)$ as stated in (A8), which ensures (A9) holds. $\square$

**Theorem A1.** *Suppose the regularity conditions give in Lemma A2 and Assumptions A7-A8 (given in Supplement A5) hold. Additionally, assume that the dimension of $\boldsymbol{\theta}$, denoted by $K_n$, increases with $n$ in a polynomial rate $K_n = O(n^\zeta)$ for some constant $\zeta > 1$, while the true StoNet is sparse with the number of nonzero connections $m_n \prec \frac{n}{c \log(K_n/n)}$ for some constant $c > 1$. Set the hyper-parameters of the prior (6) to satisfy the conditions:*

$$(\frac{n}{K_n})^c \prec \lambda_n \prec \frac{n}{K_n}, \quad \sigma_1 = O(1), \quad (\frac{n}{K_n})^c \prec \sigma_0 \prec \min\left\{1 - \frac{n}{K_n}, \frac{\delta_n}{\sqrt{c \log(K_n) - (c - 1)\log(n)}}\right\}. \tag{A14}$$

*Then $\|\hat{\boldsymbol{\theta}}_n^* - \boldsymbol{\theta}^*\| \xrightarrow{p} 0$ holds as $n \to \infty$, where $\boldsymbol{\theta}^*$ denotes the true parameter of the StoNet (4), and $\hat{\boldsymbol{\theta}}_n^*$ is up to a loss-invariant transformation.*

*Proof.* Since $\hat{\boldsymbol{\theta}}_n^*$ is a solution to equation (7), it serves as the maximum *a posteriori* (MAP) estimator of $\boldsymbol{\theta}$ with respect to the incomplete data (by treating $\boldsymbol{Z}$ as missing). By Lemma A3, we immediately have its consistency with respect to $\boldsymbol{\theta}^*$, i.e.,

$$\|\hat{\boldsymbol{\theta}}_n^* - \boldsymbol{\theta}^*\| \xrightarrow{p} 0, \quad \text{as } n \to \infty. \tag{A15}$$

Among the conditions of Lemma A3, we only need to verify Assumption A9, since the others are generally satisfied. Recall that we adopt the mixture Gaussian prior (6) in computing the MAP of $\boldsymbol{\theta}$. By Lemma A4, Assumption A9 is satisfied. This concludes the proof. $\square$

**Remark A2.** *In Theorem A1, we assume that the true sparse StoNet is of size $m_n = o(n)$. This assumption can be justified based on the theory established in Bölcskei et al. (2019), Schmidt-Hieber (2017), and Petersen & Voigtlaender (2018), where it is shown that a DNN of this size has been large enough to approximate many classes of functions, including affine, piecewise smooth, and $\alpha$-Hölder smooth functions. See Sun et al. (2022) for discussions on this issue. Additionally, Sun et al. (2022) showed that a sparse neural network of this size has been large enough to achieve the desired function approximation and posterior consistency, with the mixture Gaussian prior (6), as the sample size $n$ becomes large. Our theory allows $K_n$ to increase polynomially with $n$, which is typically satisfied by deep neural networks.*

### A5.3 Proof of Theorem 1

**Justification of the estimator (9)** To justify the pooled Monte Carlo average, fix an integrable test function $\varphi$ and define $\bar{\varphi}_i := \frac{1}{M} \sum_{l=1}^{M} \varphi(\boldsymbol{z}_i^{(l)})$. Conditional on $\boldsymbol{A}_i = \boldsymbol{a}_i$, the draws $\boldsymbol{z}_i^{(l)} \sim p(\cdot \mid \boldsymbol{a}_i)$ are i.i.d. (or ergodic), hence

$$\bar{\varphi}_i \xrightarrow{p} \mathbb{E}\{\varphi(\boldsymbol{z}) \mid \boldsymbol{a}_i\} \qquad \text{as } \mathcal{M} \to \infty,$$

where $\xrightarrow{p}$ denotes convergence in probability. Therefore, for any fixed $n$,

$$\frac{1}{n\mathcal{M}} \sum_{i=1}^{n} \sum_{l=1}^{\mathcal{M}} \varphi(\boldsymbol{z}_i^{(l)}) = \frac{1}{n} \sum_{i=1}^{n} \bar{\varphi}_i \xrightarrow{p} \frac{1}{n} \sum_{i=1}^{n} \mathbb{E}\{\varphi(\boldsymbol{z}) \mid \boldsymbol{a}_i\}, \qquad \text{as } \mathcal{M} \to \infty.$$

Under Assumption 2, $\{\boldsymbol{a}_i\}_{i=1}^{n}$ can be assumed to be i.i.d., the weak law of large numbers implies

$$\frac{1}{n} \sum_{i=1}^{n} \mathbb{E}\{\varphi(\boldsymbol{z}) \mid \boldsymbol{a}_i\} \xrightarrow{p} \mathbb{E}_A[\mathbb{E}\{\varphi(\boldsymbol{z}) \mid \boldsymbol{a}\}] = \int \varphi(\boldsymbol{z})\, p(\boldsymbol{z})\, d\boldsymbol{z}, \qquad \text{as } n \to \infty.$$

**Proof of Theorem 1**

*Proof.* Consider the joint density function:

$$\pi(\boldsymbol{Z}, \boldsymbol{Y} | \boldsymbol{A}, \boldsymbol{\theta}^*) = \pi(\boldsymbol{Z} | \boldsymbol{A}, \boldsymbol{\theta}_1^*)\pi(\boldsymbol{Y} | \boldsymbol{Z}, \boldsymbol{A}, \boldsymbol{\theta}_2^*),$$

under the assumption that the true model is a sparse StoNet (4) parameterized by $\boldsymbol{\theta}^*$. Then we have

$$\mathbb{E}(Y(\boldsymbol{a}) | \boldsymbol{\theta}^*) = \int \boldsymbol{y}\pi(\boldsymbol{z} | \boldsymbol{\theta}_1^*)\pi(\boldsymbol{y} | \boldsymbol{z}, \boldsymbol{a}, \boldsymbol{\theta}_2^*)d\boldsymbol{z}d\boldsymbol{y} = \int \mu_2(\boldsymbol{z}, \boldsymbol{a}, \boldsymbol{\theta}_2^*)\pi(\boldsymbol{z} | \boldsymbol{\theta}_1^*)d\boldsymbol{z}.$$

Let $\boldsymbol{z}_i^{(l)}$, for $l = 1, 2, \ldots, \mathcal{M}$, denote $\mathcal{M}$ independent samples drawn from $\pi(\boldsymbol{z} | \boldsymbol{a}_i, \hat{\boldsymbol{\theta}}_1^*)$. Let

$$\widehat{\mathbb{E}(Y(\boldsymbol{a}) | \hat{\boldsymbol{\theta}}_n^*)} = \frac{1}{n\mathcal{M}} \sum_{i=1}^{n} \sum_{l=1}^{\mathcal{M}} \mu_2(\boldsymbol{z}_i^{(l)}, \boldsymbol{a}, \hat{\boldsymbol{\theta}}_2^*).$$

By the standard property of Monte Carlo averages, as justified for the estimator (9), we have

$$\|\widehat{\mathbb{E}(Y(\boldsymbol{a}) | \hat{\boldsymbol{\theta}}_n^*)} - \mathbb{E}(Y(\boldsymbol{a}) | \hat{\boldsymbol{\theta}}_n^*)\| \xrightarrow{p} 0, \quad \text{as } n, \mathcal{M} \to \infty. \tag{A16}$$

On the other hand, by the consistency of $\hat{\boldsymbol{\theta}}_n^* = (\hat{\boldsymbol{\theta}}_1^*, \hat{\boldsymbol{\theta}}_2^*)$ (with respect to $\boldsymbol{\theta}^*$) as established in Lemma A1, we have

$$\|\mathbb{E}(Y(\boldsymbol{a}) | \hat{\boldsymbol{\theta}}_n^*) - \mathbb{E}(Y(\boldsymbol{a}) | \boldsymbol{\theta}^*)\| \xrightarrow{p} 0, \quad \text{as } n \to \infty, \tag{A17}$$

since $\mu_2(\cdot)$ is continuous respect to the parameters (as assumed for the neural network model).

Combining the convergence results in (A16) and (A17), we have

$$\|\widehat{\mathbb{E}(Y(\boldsymbol{a}) | \hat{\boldsymbol{\theta}}_n^*)} - \mathbb{E}(Y(\boldsymbol{a}) | \boldsymbol{\theta}^*)\| \leq \|\widehat{\mathbb{E}(Y(\boldsymbol{a}) | \hat{\boldsymbol{\theta}}_n^*)} - \mathbb{E}(Y(\boldsymbol{a}) | \hat{\boldsymbol{\theta}}^*)\| + \|\mathbb{E}(Y(\boldsymbol{a}) | \hat{\boldsymbol{\theta}}^*) - \mathbb{E}(Y(\boldsymbol{a}) | \boldsymbol{\theta}^*)\| \xrightarrow{p} 0,$$

as $n \to \infty$ and $\mathcal{M} \to \infty$. This concludes the proof. $\qquad \square$

### A5.4 Proof of Theorem 2

We can leverage the theory developed in (Sun et al., 2022) to justify the misspecification error of the causal effect.

**Assumption A10.** *(Restatement of Assumption A.2 of Sun et al. (2022)) The sparse DNN model $\mu_{A,n}$ and $\mu_{Y,n}$ satisfy the following conditions:*

1. *The network structure satisfies:*

$$r_n H_n \log n \;+\; r_n \log \overline{L} \;+\; s_n \log p_n \;\leq\; C_0 n^{1-\varepsilon},$$

*where $0 < \varepsilon < 1$ is a small constant, $r_n = |\gamma^*|$ denotes the connectivity of $\gamma^*$, $\overline{L} = \max_{1 \leq j \leq H_n - 1} L_j$ denotes the maximum hidden layer width, and $s_n$ denotes the input dimension of $\gamma^*$.*

2. *The network weights are polynomially bounded:*

$$\|\boldsymbol{\beta}^*\|_\infty \leq E_n, \qquad E_n = n^{C_1}$$

*for some constant $C_1 > 0$.*

For example, affine-system functions (Bolcskei et al. (2019)), piecewise-smooth functions with a fixed input dimension (Petersen & Voigtlaender (2018)), and bounded $\alpha$-Holder smooth function (Polson & Ročková (2018)). Assumption A10 clarifies the class of sparse DNNs that can approximate the unknown structural mean functions, and hence considered as "true sparse DNN" in the paper. Informally, Assumption A10 requires that the sparse deep network whose number of active weights and relevant inputs grows slower than the sample size, and whose weights are at most polynomially large in $n$. This ensures the "true" network lies in a capacity-controlled function class. By changing the network structure and network weight accordingly, we can establish different approximation error upper-bounding sequence $\omega_n$ for different function classes $\mathcal{F}$, see discussion in Sun et al. (2022) Section 2.2 for more details.

*Proof.* Let $\mathcal{G}_n$ denote the class of sparse DNNs compatible with the Assumption A10, we define for each sample size $n$ the pseudo-true sparse DNNs $(\mu^*_{A,n}, \mu^*_{Y,n})$ as minimizers of the approximation error within $\mathcal{G}_n$:

$$(\mu^*_{A,n}, \mu^*_{Y,n}) \in \arg \min_{(\mu_1, \mu_2) \in \mathcal{G}_n} \{\|\mu_{A,n} - m_A\|_{L^2} + \|\mu_{Y,n} - m_Y\|_{L^2}\}$$

by Assumption 3, there exists a sequence $\omega_n \to 0$ such that

$$\{\|\mu_{A,n} - m_A\|_{L^2} + \|\mu_{Y,n} - m_Y\|_{L^2}\} \lesssim \omega_n$$

with $\omega_n$ scaling at the chosen rates for chosen function $\mathcal{F}$ (e.g. $\omega_n \asymp n^{-\alpha/(2\alpha+d)}$ up to logarithmic factors for $\alpha$ - Hölder functions in dimension $d$).

Let $P_0$ denote the true joint law of $(\boldsymbol{A}, \boldsymbol{Z}, \boldsymbol{Y})$. For any parameter $\eta$ of sparse DNN, let $Q_\eta$ be the induced joint law under the corresponding generalized linear model (Gaussian or logistic) with regression function $\mu_\eta$, and let $P_{\boldsymbol{\theta}}$ be the distribution induced by the CI-StoNet with parameter $\boldsymbol{\theta}$. Following Sun et al. (2022), we define Kullback–Leibler divergence

$$d_0(q, p) = \int q \log \frac{q}{p},$$

and a family of distance $d_t(q, p) = \frac{1}{t} \int q[(\frac{q}{p})^t - 1]$ for any $t > 0$, which decreases to $d_0$ as $t$ decreases toward 0. For Gaussian regression with mean functions $m_1$ and $m_2$, they show that, for fixed noise scale and bounded regression functions with known $\sigma^2$

$$d_0(p_{m_1}, p_{m_2}) = \frac{1}{2\sigma^2}(m_1 - m_2)^2,$$

and for logistic they derive upper bounds

$$d_1(p_{m_1}, p_{m_2}) \leq \frac{1}{2}(m_1 - m_2)^2 + O((m_1 - m_2)^3).$$

Since $d_0(p_{m_1}, p_{m_2}) \leq d_1(p_{m_1}, p_{m_2})$, we have $d_0(p_{m_1}, p_{m_2}) \leq C(m_1 - m_2)^2$ for some constant $C > 0$ depending only on the uniform bound on $\mu$. Then it gives

$$\mathrm{KL}(P_0, Q_\eta) \lesssim \|\mu_\eta - \mu\|^2_{L^2},$$

where $\mu_\eta$ is the DNN-based regression function in $Q_\eta$ and $\mu^*$ is the true regression function. Putting things together, if the function class $\mathcal{F}$ is approximable at rate $\omega_n$ in $L_2$ by sparse DNNs, then there exist sparse DNN parameters $\eta_n$ such that

$$\mathrm{KL}(P_0, Q_{\eta_n}) \lesssim \omega_n^2.$$

In the StoNet definition, the deterministic parts $m_1$, $m_2$ are themselves neural networks. Therefore, for any sparse DNN architecture that satisfies Assumption A10, we can embed the same architecture into the StoNet by choosing $\boldsymbol{\theta}$ so that the StoNet's deterministic part coincides with the DNN, which yields a mapping $\eta \to \boldsymbol{\theta}(\eta)$ with $Q_\eta = P_{\boldsymbol{\theta}(\eta)}$. Therefore, defining a pseudo-true StoNet parameter

$$\boldsymbol{\theta}^* = \arg\min_{\boldsymbol{\theta}} \mathrm{KL}(P_0, P_{\boldsymbol{\theta}}),$$

then we have

$$\mathrm{KL}(P_0, P_{\boldsymbol{\theta}^*}) \leq \mathrm{KL}(P_0, P_{\boldsymbol{\theta}(\eta)}) = \mathrm{KL}(P_0, Q_\eta) \lesssim \omega_n^2.$$

Now consider the simple confounding case (recall the setup in 4). Let $\psi(P_0) = \int m_Y(\boldsymbol{a}, \boldsymbol{z}) p_{P_0}(\boldsymbol{z}) d\boldsymbol{z}$ and $\psi(P_{\boldsymbol{\theta}^*}) = \int \mu_2(\boldsymbol{a}, \boldsymbol{z}) p_{P_{\boldsymbol{\theta}^*}}(\boldsymbol{z}) d\boldsymbol{z}$, the approximation error between the true and the pseudo-true causal estimand is:

$$\|\psi(P_0) - \psi(P_{\boldsymbol{\theta}^*})\| = \|\int m_Y(\boldsymbol{a}, \boldsymbol{z}) p_{P_0}(\boldsymbol{z}) d\boldsymbol{z} - \int \mu_2(\boldsymbol{a}, \boldsymbol{z}) p_{P_{\boldsymbol{\theta}^*}}(\boldsymbol{z}) d\boldsymbol{z}\|$$

$$\leq \underbrace{\|\int [m_Y(\boldsymbol{a}, \boldsymbol{z}) - \mu_2(\boldsymbol{a}, \boldsymbol{z})] p_{P_0}(\boldsymbol{z}) d\boldsymbol{z}\|}_{A_1} + \underbrace{\|\int \mu_2(\boldsymbol{a}, \boldsymbol{z})[p_{P_0}(\boldsymbol{z}) - p_{P_{\boldsymbol{\theta}^*}}(\boldsymbol{z})] d\boldsymbol{z}\|}_{A_2}$$

$A_1$ can be bounded by

$$A_1 \leq \|m_Y(\boldsymbol{a}, \boldsymbol{z}) - \mu_2(\boldsymbol{a}, \boldsymbol{z})\|_{L_2(P_0(\boldsymbol{z}))} \lesssim \omega_n.$$

The first line is by Cauchy–Schwarz, and the second line is by sparse-DNN approximation results and StoNet/DNN equivalence.

Assume $|\mu_2(\boldsymbol{a}, \boldsymbol{z})| \leq C_{\mu_2}$, then

$$A_2 \leq C_{\mu_2} \int |p_{P_0(\boldsymbol{z})} - p_{\boldsymbol{\theta}^*}(\boldsymbol{z})| d\boldsymbol{z}$$

$$\leq 2C_{\mu_2} \sqrt{\frac{1}{2}\mathrm{KL}(P_{0,Z}, P_{\boldsymbol{\theta}^*, Z})}$$

$$\leq \sqrt{2}C_{\mu_2} \sqrt{\mathrm{KL}(P_0, P_{\boldsymbol{\theta}^*})}$$

$$\lesssim \omega_n.$$

From first line to second line we use Pinsker's inequality, and from the third line to the last line we used the previous result that $\mathrm{KL}(P_0, P_{\boldsymbol{\theta}^*}) \lesssim \omega_n^2$. From the second to the third line we use the data-processing inequality for KL divergence (monotonicity under marginalization):

$$\mathrm{KL}(P_{0,Z}, P_{\boldsymbol{\theta}^*, Z}) \leq \mathrm{KL}(P_0, P_{\boldsymbol{\theta}^*}).$$

Putting pieces together, we have

$$\|\psi(P_0) - \psi(P_{\boldsymbol{\theta}^*})\| \leq A_1 + A_2 \lesssim \omega_n + \omega_n \lesssim \omega_n.$$

Hence we propagated the approximation error of nuisance functions to the causal estimand and proved that the approximation error for causal estimand is $O(\omega_n)$. $\qquad\square$

Theorem 1 in our paper controls the estimation error, which is guaranteed by the properties of the sparse DNN. The misspecification error is determined solely by the approximation power of the sparse-DNN class (depth, width, sparsity rate). So the pirors only impact the misspecificaiton error indirectly, through controlling the model capacity.

A5.5 FINITE-SAMPLE ANALYSIS ON OVERLAP INFLATION

**Assumption A11** (Boundedness and overlap). *For each treatment $\boldsymbol{a} \in \mathcal{A}$, let $m^*(\boldsymbol{z}, \boldsymbol{a})$ denote the true outcome nuisance and $\hat{m}(\boldsymbol{z}, \boldsymbol{a})$ its estimator. Assume:*

1. *Outcome boundedness:*

$$\sup_z |m^*(\boldsymbol{z}, \boldsymbol{a})| \leq C_m, \qquad \sup_z |\hat{m}(\boldsymbol{z}, \boldsymbol{a})| \leq C_m.$$

2. *Latent propensity (positivity) bounds: there exist constants $0 < \kappa_z(\boldsymbol{a}) \leq K_z(\boldsymbol{a}) < \infty$ such that for all $z$ in the relevant support,*

$$\kappa_z(\boldsymbol{a}) \leq p^*(\boldsymbol{a} \mid \boldsymbol{z}) \leq K_z(\boldsymbol{a}), \qquad \kappa_z(\boldsymbol{a}) \leq \hat{p}(\boldsymbol{a} \mid \boldsymbol{z}) \leq K_z(\boldsymbol{a}).$$

3. *Observed propensity bounds: there exist constants $0 < \kappa_x(\boldsymbol{a}) \leq C_{ax}(\boldsymbol{a}) < \infty$ such that for all proxy $x_i$,*

$$\kappa_x(\boldsymbol{a}) \leq p^*(\boldsymbol{a} \mid \boldsymbol{x}_i) \leq C_{ax}(\boldsymbol{a}), \qquad \kappa_x(\boldsymbol{a}) \leq \hat{p}(\boldsymbol{a} \mid \boldsymbol{x}_i) \leq C_{ax}(\boldsymbol{a}).$$

4. *Marginal treatment density bound: For each treatment value $\boldsymbol{a} \in \mathcal{A}$, there exists $\kappa_A(\boldsymbol{a}) > 0$ such that*

$$p^*(\boldsymbol{a}) \geq \kappa_A(\boldsymbol{a}), \qquad \hat{p}(\boldsymbol{a}) \geq \kappa_A(\boldsymbol{a}).$$

*Define $\kappa(\boldsymbol{a}) := \min\{\kappa_z(\boldsymbol{a}), \kappa_x(\boldsymbol{a}), \kappa_A(\boldsymbol{a})\}$ and $K(\boldsymbol{a}) := K_z(\boldsymbol{a})$.*

**Lemma A5** (Bayes-bridge in simple confounding). *Fix $\boldsymbol{a} \in \mathcal{A}$ with $p(\boldsymbol{a}) > 0$. Then*

$$p(\boldsymbol{z}) = p(\boldsymbol{z} \mid \boldsymbol{a}) \, w(\boldsymbol{z}, \boldsymbol{a}), \qquad w(\boldsymbol{z}, \boldsymbol{a}) := \frac{p(\boldsymbol{a})}{p(\boldsymbol{a} \mid \boldsymbol{z})}.$$

*In particular, under Assumption A11,*

$$0 < w(\boldsymbol{z}, \boldsymbol{a}) \leq \frac{K(\boldsymbol{a})}{\kappa(\boldsymbol{a})}, \qquad |w(\boldsymbol{z}, \boldsymbol{a}) - 1| \leq \max\left\{1 - \frac{\kappa_A(\boldsymbol{a})}{K(\boldsymbol{a})}, \; \frac{K(\boldsymbol{a})}{\kappa(\boldsymbol{a})} - 1\right\}.$$

*The same statements hold for the estimated model with $\hat{w}(\boldsymbol{z}, \boldsymbol{a}) := \hat{p}(\boldsymbol{a})/\hat{p}(\boldsymbol{a} \mid \boldsymbol{z})$.*

*Proof.* Bayes' rule gives $p(\boldsymbol{z} \mid \boldsymbol{a}) = \frac{p(\boldsymbol{a}|\boldsymbol{z})p(\boldsymbol{z})}{p(\boldsymbol{a})}$, hence $p(\boldsymbol{z}) = p(\boldsymbol{z} \mid \boldsymbol{a})\frac{p(\boldsymbol{a})}{p(\boldsymbol{a}|\boldsymbol{z})}$. Under Assumption A11, $p(\boldsymbol{a} \mid \boldsymbol{z}) \geq \kappa(\boldsymbol{a})$ and $p(\boldsymbol{a} \mid \boldsymbol{z}) \leq K(\boldsymbol{a})$. Also $p(\boldsymbol{a}) = \int p(\boldsymbol{a} \mid \boldsymbol{z})p(\boldsymbol{z})d\boldsymbol{z} \leq K(\boldsymbol{a})$, and by Assumption A11-(4), $p(\boldsymbol{a}) \geq \kappa_A(\boldsymbol{a})$. Therefore $w(\boldsymbol{z}, \boldsymbol{a}) \leq K(\boldsymbol{a})/\kappa(\boldsymbol{a})$ and $w(\boldsymbol{z}, \boldsymbol{a}) \geq \kappa_A(\boldsymbol{a})/K(\boldsymbol{a})$, which implies the bound on $|w - 1|$. $\square$

**Theorem A2** (Simple-confounding bound for estimator (9)). *Fix $\boldsymbol{a} \in \mathcal{A}$. Let $m^*(\boldsymbol{z}, \boldsymbol{a}) = \mathbb{E}[Y \mid \boldsymbol{A} = \boldsymbol{a}, \boldsymbol{Z} = \boldsymbol{z}]$ and $\hat{m}(\boldsymbol{z}, \boldsymbol{a}) = \mu_2(\boldsymbol{z}, \boldsymbol{a}; \hat{\boldsymbol{\theta}}_2^*)$. Suppose Assumption A11 holds. Define the population causal estimand*

$$\psi^*(\boldsymbol{a}) := \int m^*(\boldsymbol{z}, \boldsymbol{a}) \, p^*(\boldsymbol{z}) \, d\boldsymbol{z}.$$

*Assume the latent draws used in (9) satisfy*

$$\boldsymbol{z}_i^{(l)} \overset{i.i.d.}{\sim} \hat{p}(\boldsymbol{z} \mid \boldsymbol{a}_i), \qquad i = 1, \ldots, n, \; l = 1, \ldots, \mathcal{M},$$

*conditional on $(\boldsymbol{a}_i)_{i=1}^n$ and the fitted model. Consider the estimator*

$$\hat{\psi}_{n,\mathcal{M}}(\boldsymbol{a}) := \frac{1}{n\mathcal{M}} \sum_{i=1}^n \sum_{l=1}^{\mathcal{M}} \hat{m}(\boldsymbol{z}_i^{(l)}, \boldsymbol{a}).$$

*Let the corresponding (model-based) conditional plug-in target be*

$$\Phi_{n,\text{cond}}(\hat{\boldsymbol{\theta}}; \boldsymbol{a}) := \frac{1}{n} \sum_{i=1}^n \int \hat{m}(\boldsymbol{z}, \boldsymbol{a}) \, \hat{p}(\boldsymbol{z} \mid \boldsymbol{a}_i) \, d\boldsymbol{z},$$

*and rewrite $\psi^*(\boldsymbol{a})$ as*

$$\psi^*(\boldsymbol{a}) = \Phi^*_{n,\mathrm{bridge}}(\boldsymbol{a}) := \frac{1}{n} \sum_{i=1}^{n} \int m^*(\boldsymbol{z}, \boldsymbol{a}) \, p^*(\boldsymbol{z} \mid \boldsymbol{a}_i) \, w^*(\boldsymbol{z}, \boldsymbol{a}_i) \, d\boldsymbol{z},$$

*where*

$$w^*(\boldsymbol{z}, \boldsymbol{a}_i) := \frac{p^*(\boldsymbol{a}_i)}{p^*(\boldsymbol{a}_i \mid \boldsymbol{z})}.$$

*For any $\delta \in (0,1)$, with probability at least $1 - \delta$,*

$$|\hat{\psi}_{n,\mathcal{M}}(\boldsymbol{a}) - \psi^*(\boldsymbol{a})| \leq \underbrace{C_m \sqrt{\frac{2 \log(2/\delta)}{n\mathcal{M}}}}_{\textit{Monte Carlo}} + \underbrace{\left| \Phi_{n,\mathrm{cond}}(\hat{\boldsymbol{\theta}}; \boldsymbol{a}) - \Phi^*_{n,\mathrm{cond}}(\boldsymbol{a}) \right|}_{\textit{estimation under } p(\boldsymbol{z}|\boldsymbol{a})}$$

$$+ \underbrace{\left| \Phi^*_{n,\mathrm{cond}}(\boldsymbol{a}) - \Phi^*_{n,\mathrm{bridge}}(\boldsymbol{a}) \right|}_{\textit{Bayes-bridge (overlap-inflated) bias}},$$

*where $C_m$ is from Assumption A11-(1), and*

$$\Phi^*_{n,\mathrm{cond}}(\boldsymbol{a}) := \frac{1}{n} \sum_{i=1}^{n} \int m^*(\boldsymbol{z}, \boldsymbol{a}) \, p^*(\boldsymbol{z} \mid \boldsymbol{a}_i) \, d\boldsymbol{z}.$$

*Moreover, the Bayes-bridge bias term satisfies*

$$\left| \Phi^*_{n,\mathrm{cond}}(\boldsymbol{a}) - \Phi^*_{n,\mathrm{bridge}}(\boldsymbol{a}) \right| = \frac{1}{n} \sum_{i=1}^{n} \left| \int m^*(\boldsymbol{z}, \boldsymbol{a}) \, p^*(\boldsymbol{z} \mid \boldsymbol{a}_i) \, (w^*(\boldsymbol{z}, \boldsymbol{a}_i) - 1) \, d\boldsymbol{z} \right|$$

$$\leq C_m \cdot \max_{1 \leq i \leq n} \max \left\{ 1 - \frac{\kappa_A(\boldsymbol{a}_i)}{K(\boldsymbol{a}_i)}, \ \frac{K(\boldsymbol{a}_i)}{\kappa(\boldsymbol{a}_i)} - 1 \right\}.$$

*Proof.* We prove the claimed decomposition and bounds.

**Step 1 (error decomposition).** Add and subtract $\Phi_{n,\mathrm{cond}}(\hat{\boldsymbol{\theta}}; \boldsymbol{a})$ and $\Phi^*_{n,\mathrm{cond}}(\boldsymbol{a})$:

$$|\hat{\psi}_{n,\mathcal{M}}(\boldsymbol{a}) - \psi^*(\boldsymbol{a})| = |\hat{\psi}_{n,\mathcal{M}}(\boldsymbol{a}) - \Phi^*_{n,\mathrm{bridge}}(\boldsymbol{a})|$$
$$\leq |\hat{\psi}_{n,\mathcal{M}}(\boldsymbol{a}) - \Phi_{n,\mathrm{cond}}(\hat{\boldsymbol{\theta}}; \boldsymbol{a})| + |\Phi_{n,\mathrm{cond}}(\hat{\boldsymbol{\theta}}; \boldsymbol{a}) - \Phi^*_{n,\mathrm{cond}}(\boldsymbol{a})| + |\Phi^*_{n,\mathrm{cond}}(\boldsymbol{a}) - \Phi^*_{n,\mathrm{bridge}}(\boldsymbol{a})|.$$

This gives the displayed three-term bound once each term is controlled.

**Step 2 (Monte Carlo term).** Conditional on $(\boldsymbol{a}_i)_{i=1}^{n}$ and the fitted model, by assumption the draws $\boldsymbol{z}_i^{(l)} \overset{i.i.d.}{\sim} \hat{p}(\boldsymbol{z} \mid \boldsymbol{a}_i)$ are independent across $(i, l)$, and $\hat{m}(\boldsymbol{z}_i^{(l)}, \boldsymbol{a}) \in [-C_m, C_m]$. Thus, Hoeffding's inequality implies for any $\epsilon > 0$,

$$\mathbb{P}\Big( \Big| \hat{\psi}_{n,\mathcal{M}}(\boldsymbol{a}) - \Phi_{n,\mathrm{cond}}(\hat{\boldsymbol{\theta}}; \boldsymbol{a}) \Big| > \epsilon \Big) \leq 2 \exp\Big( -\frac{n\mathcal{M}\,\epsilon^2}{2 C_m^2} \Big).$$

Setting the right-hand side to $\delta$ yields that with probability at least $1 - \delta$,

$$\Big| \hat{\psi}_{n,\mathcal{M}}(\boldsymbol{a}) - \Phi_{n,\mathrm{cond}}(\hat{\boldsymbol{\theta}}; \boldsymbol{a}) \Big| \leq C_m \sqrt{\frac{2 \log(2/\delta)}{n\mathcal{M}}}.$$

**Step 3 (Bayes-bridge bias bound).** By definition,

$$\Phi^*_{n,\mathrm{cond}}(\boldsymbol{a}) - \Phi^*_{n,\mathrm{bridge}}(\boldsymbol{a}) = \frac{1}{n} \sum_{i=1}^{n} \int m^*(\boldsymbol{z}, \boldsymbol{a}) \, p^*(\boldsymbol{z} \mid \boldsymbol{a}_i) \, (1 - w^*(\boldsymbol{z}, \boldsymbol{a}_i)) \, d\boldsymbol{z}.$$

Taking absolute values and using $\|m^*(\cdot, \boldsymbol{a})\|_\infty \le C_m$,

$$\left|\Phi^*_{n,\text{cond}}(\boldsymbol{a}) - \Phi^*_{n,\text{bridge}}(\boldsymbol{a})\right| \le \frac{C_m}{n}\sum_{i=1}^n \int p^*(\boldsymbol{z} \mid \boldsymbol{a}_i)\, |w^*(\boldsymbol{z}, \boldsymbol{a}_i) - 1|\, d\boldsymbol{z} \le C_m \cdot \sup_i \sup_{\boldsymbol{z}} |w^*(\boldsymbol{z}, \boldsymbol{a}_i) - 1|.$$

Under Assumption A11, for any $a$ and all $z$, $\kappa(\boldsymbol{a}) \le p^*(\boldsymbol{a} \mid \boldsymbol{z}) \le K(\boldsymbol{a})$. Moreover, $p^*(\boldsymbol{a}) = \int p^*(\boldsymbol{a} \mid \boldsymbol{z})p^*(\boldsymbol{z})d\boldsymbol{z} \le K(\boldsymbol{a})$, and by Assumption A11-(4), $p^*(\boldsymbol{a}) \ge \kappa_A(\boldsymbol{a})$. Hence,

$$\frac{\kappa_A(\boldsymbol{a})}{K(\boldsymbol{a})} \le \frac{p^*(\boldsymbol{a})}{p^*(\boldsymbol{a} \mid \boldsymbol{z})} \le \frac{K(\boldsymbol{a})}{\kappa(\boldsymbol{a})},$$

so for any $i$ (with $\boldsymbol{a}_i$ in place of $\boldsymbol{a}$),

$$\sup_{\boldsymbol{z}} |w^*(\boldsymbol{z}, \boldsymbol{a}_i) - 1| = \sup_{\boldsymbol{z}} \left|\frac{p^*(\boldsymbol{a}_i)}{p^*(\boldsymbol{a}_i \mid \boldsymbol{z})} - 1\right| \le \max\left\{1 - \frac{\kappa_A(\boldsymbol{a}_i)}{K(\boldsymbol{a}_i)}, \frac{K(\boldsymbol{a}_i)}{\kappa(\boldsymbol{a}_i)} - 1\right\}.$$

**Step 4 (estimation-under-$p(z|a)$ term).** The remaining term $\left|\Phi_{n,\text{cond}}(\hat{\boldsymbol{\theta}}; \boldsymbol{a}) - \Phi^*_{n,\text{cond}}(\boldsymbol{a})\right|$ is exactly the estimation error of the plug-in functional under the conditional latent laws:

$$\Phi_{n,\text{cond}}(\hat{\boldsymbol{\theta}}; \boldsymbol{a}) - \Phi^*_{n,\text{cond}}(\boldsymbol{a}) = \frac{1}{n}\sum_{i=1}^n \int \left[\hat{m}(\boldsymbol{z}, \boldsymbol{a})\hat{p}(\boldsymbol{z} \mid \boldsymbol{a}_i) - m^*(\boldsymbol{z}, \boldsymbol{a})p^*(\boldsymbol{z} \mid \boldsymbol{a}_i)\right]d\boldsymbol{z}.$$

If desired, it can be bounded further by adding and subtracting $m^*(\boldsymbol{z}, \boldsymbol{a})\hat{p}(\boldsymbol{z} \mid \boldsymbol{a}_i)$ and using $\|m^*(\cdot, \boldsymbol{a})\|_\infty \le C_m$, $\|\hat{m}(\cdot, \boldsymbol{a})\|_\infty \le C_m$:

$$\left|\Phi_{n,\text{cond}}(\hat{\boldsymbol{\theta}}; \boldsymbol{a}) - \Phi^*_{n,\text{cond}}(\boldsymbol{a})\right| \le \epsilon_m(\boldsymbol{a}) + \frac{2C_m}{n}\sum_{i=1}^n \|\hat{p}(\boldsymbol{z} \mid \boldsymbol{a}_i) - p^*(\boldsymbol{z} \mid \boldsymbol{a}_i)\|_{\text{TV}},$$

where $\epsilon_m(\boldsymbol{a}) := \sup_{\boldsymbol{z}} |\hat{m}(\boldsymbol{z}, \boldsymbol{a}) - m^*(\boldsymbol{z}, \boldsymbol{a})|$.

Combining Steps 1–3 yields the stated three-term high-probability bound, and Step 3 provides the explicit overlap-inflated Bayes-bridge bias bound. This completes the proof. $\qquad\square$

**Assumption A12** (Proxy conditional independence)**.** *In the basic proxy setting, assume*

$$\boldsymbol{A} \perp \boldsymbol{X} \mid \boldsymbol{Z},$$

*equivalently $p(\boldsymbol{a} \mid \boldsymbol{z}, \boldsymbol{x}) = p(\boldsymbol{a} \mid \boldsymbol{z})$ for all $(\boldsymbol{a}, \boldsymbol{z}, \boldsymbol{x})$.*

In consequence, we have the proxy Bayesian identity:

$$p(\boldsymbol{z} \mid \boldsymbol{x}, \boldsymbol{a}) = \frac{p(\boldsymbol{a} \mid \boldsymbol{z})\, p(\boldsymbol{z} \mid \boldsymbol{x})}{p(\boldsymbol{a} \mid \boldsymbol{x})}. \tag{A18}$$

**Lemma A6** (TV bound for proxy conditional $p(\boldsymbol{z} \mid \boldsymbol{x}, \boldsymbol{a})$)**.** *Assume Assumptions A11 and A12. Fix $\boldsymbol{x}$. Let $f(\boldsymbol{a}, \boldsymbol{z}|\boldsymbol{x}) := p(\boldsymbol{a} \mid \boldsymbol{z})p(\boldsymbol{z} \mid \boldsymbol{x})$ and $\hat{f}(\boldsymbol{a}, \boldsymbol{z}|\boldsymbol{x}) := \hat{p}(\boldsymbol{a} \mid \boldsymbol{z})\hat{p}(\boldsymbol{z} \mid \boldsymbol{x})$, with normalizers $g := p(\boldsymbol{a} \mid \boldsymbol{x}) = \int f(\boldsymbol{a}, \boldsymbol{z}|\boldsymbol{x})\, d\boldsymbol{z}$ and $\hat{g} := \hat{p}(\boldsymbol{a} \mid \boldsymbol{x}) = \int \hat{f}(\boldsymbol{a}, \boldsymbol{z}|\boldsymbol{x})\, d\boldsymbol{z}$. Then*

$$\|\hat{p}(\boldsymbol{z} \mid \boldsymbol{x}, \boldsymbol{a}) - p^*(\boldsymbol{z} \mid \boldsymbol{x}, \boldsymbol{a})\|_{\text{TV}} \le \frac{1}{2\, p^*(\boldsymbol{a} \mid \boldsymbol{x})}\|\hat{f} - f\|_1 + \frac{\|\hat{f}\|_1}{2\, p^*(\boldsymbol{a} \mid \boldsymbol{x})\, \hat{p}(\boldsymbol{a} \mid \boldsymbol{x})}\, |\hat{p}(\boldsymbol{a} \mid \boldsymbol{x}) - p^*(\boldsymbol{a} \mid \boldsymbol{x})|.$$

*Moreover, using Assumption A11, one has the crude bound*

$$\|\hat{f} - f\|_1 \le K(\boldsymbol{a})\, \|\hat{p}(\boldsymbol{z} \mid \boldsymbol{x}) - p^*(\boldsymbol{z} \mid \boldsymbol{x})\|_1 + \mathbb{E}_{p^*(\boldsymbol{z}|\boldsymbol{x})}[|\hat{p}(\boldsymbol{a} \mid \boldsymbol{z}) - p^*(\boldsymbol{a} \mid \boldsymbol{z})|].$$

*Proof.* Write $\hat{p}(\boldsymbol{z} \mid \boldsymbol{x}, \boldsymbol{a}) = \hat{f}(\boldsymbol{a}, \boldsymbol{z}|\boldsymbol{x})/\hat{g}$ and $p^*(\boldsymbol{z} \mid \boldsymbol{x}, \boldsymbol{a}) = f(\boldsymbol{a}, \boldsymbol{z}|\boldsymbol{x})/g$ with $g = p^*(\boldsymbol{a} \mid \boldsymbol{x})$ and $\hat{g} = \hat{p}(\boldsymbol{a} \mid \boldsymbol{x})$. Then

$$\left\|\frac{\hat{f}}{\hat{g}} - \frac{f}{g}\right\|_1 \le \left\|\frac{\hat{f} - f}{g}\right\|_1 + \left\|\hat{f}\left(\frac{1}{\hat{g}} - \frac{1}{g}\right)\right\|_1 = \frac{1}{g}\|\hat{f} - f\|_1 + \|\hat{f}\|_1\left|\frac{1}{\hat{g}} - \frac{1}{g}\right|.$$

Using $\left|\frac{1}{\hat{g}} - \frac{1}{g}\right| = \frac{|\hat{g}-g|}{g\hat{g}}$ and dividing by 2 yields the stated TV bound. For the crude bound on $\|\hat{f} - f\|_1$, add and subtract $\hat{p}(\boldsymbol{a} \mid \boldsymbol{z})p^*(\boldsymbol{z} \mid \boldsymbol{x})$:

$$\|\hat{f} - f\|_1 \leq \int \hat{p}(\boldsymbol{a} \mid \boldsymbol{z}) \, |\hat{p}(\boldsymbol{z} \mid \boldsymbol{x}) - p^*(\boldsymbol{z} \mid \boldsymbol{x})| \, d\boldsymbol{z} + \int p^*(\boldsymbol{z} \mid \boldsymbol{x}) \, |\hat{p}(\boldsymbol{a} \mid \boldsymbol{z}) - p^*(\boldsymbol{a} \mid \boldsymbol{z})| \, d\boldsymbol{z},$$

and apply $\hat{p}(\boldsymbol{a} \mid \boldsymbol{z}) \leq K(\boldsymbol{a})$. $\qquad\square$

**Assumption A13** (Gaussian latent working model stability). *Suppose the CI-StoNet working model for the latent $Z$ is Gaussian with fixed covariance:*

$$p(\boldsymbol{z} \mid u; \boldsymbol{\theta}_1) = \mathcal{N}(\mu_1(u; \boldsymbol{\theta}_1), \sigma_z^2 I),$$

*where $u = \boldsymbol{x}$ for proxy, i.e. $p(\boldsymbol{z} \mid \boldsymbol{x}; \boldsymbol{\theta}_1)$. Assume $\mu_1$ is uniformly Lipschitz in $\boldsymbol{\theta}_1$:*

$$\|\mu_1(u; \hat{\boldsymbol{\theta}}_1) - \mu_1(u; \boldsymbol{\theta}_1^*)\|_2 \leq L_1 \|\hat{\boldsymbol{\theta}}_1 - \boldsymbol{\theta}_1^*\|, \qquad \forall u.$$

**Lemma A7** (Pinsker–Gaussian TV bound). *Under Assumption A13, for any $u$,*

$$\left\|p(\cdot \mid u; \hat{\boldsymbol{\theta}}_1) - p(\cdot \mid u; \boldsymbol{\theta}_1^*)\right\|_{\mathrm{TV}} \leq \frac{L_1}{2\sigma_z} \|\hat{\boldsymbol{\theta}}_1 - \boldsymbol{\theta}_1^*\|.$$

*Proof.* By Pinsker's inequality, $\|P - Q\|_{\mathrm{TV}} \leq \sqrt{\frac{1}{2} D_{\mathrm{KL}}(P\|Q)}$. For Gaussians with common covariance $\sigma_z^2 I$,

$$D_{\mathrm{KL}}\big(\mathcal{N}(\mu^*, \sigma_z^2 I) \,\|\, \mathcal{N}(\hat{\mu}, \sigma_z^2 I)\big) = \frac{\|\mu^* - \hat{\mu}\|_2^2}{2\sigma_z^2}.$$

Hence

$$\|P - Q\|_{\mathrm{TV}} \leq \sqrt{\frac{1}{2} \cdot \frac{\|\mu^* - \hat{\mu}\|_2^2}{2\sigma_z^2}} = \frac{\|\mu^* - \hat{\mu}\|_2}{2\sigma_z} \leq \frac{L_1}{2\sigma_z} \|\hat{\boldsymbol{\theta}}_1 - \boldsymbol{\theta}_1^*\|,$$

where the last step uses Assumption A13. $\qquad\square$

**Theorem A3** (Finite-sample bound with explicit overlap inflation). *Fix $\boldsymbol{a} \in \mathcal{A}$ and suppose Assumptions A11–A13 hold. Let the sample-conditional causal target be*

$$\Phi_n^*(\boldsymbol{a}) := \frac{1}{n} \sum_{i=1}^{n} \mathbb{E}[Y(\boldsymbol{a}) \mid \boldsymbol{x}_i] = \frac{1}{n} \sum_{i=1}^{n} \int m^*(\boldsymbol{z}, \boldsymbol{a}) \, p^*(\boldsymbol{z} \mid \boldsymbol{x}_i) \, d\boldsymbol{z}.$$

*Define the plug-in functional*

$$\Phi_n(\hat{\boldsymbol{\theta}}; a) := \frac{1}{n} \sum_{i=1}^{n} \int \hat{m}(\boldsymbol{z}, \boldsymbol{a}) \, \hat{p}(\boldsymbol{z} \mid \boldsymbol{x}_i) \, d\boldsymbol{z},$$

*and the Monte Carlo approximation*

$$\hat{\Phi}_{n,M}(\boldsymbol{a}) := \frac{1}{nM} \sum_{i=1}^{n} \sum_{\ell=1}^{M} \hat{m}(\boldsymbol{z}_i^{(\ell)}, \boldsymbol{a}), \qquad \boldsymbol{z}_i^{(\ell)} \overset{i.i.d.}{\sim} \hat{p}(\boldsymbol{z} \mid \boldsymbol{x}_i).$$

*Then for any $\delta \in (0, 1)$, with probability at least $1 - \delta$,*

$$|\hat{\Phi}_{n,M}(\boldsymbol{a}) - \Phi_n^*(\boldsymbol{a})| \leq C_m \sqrt{\frac{2\log(2/\delta)}{nM}} + |\Phi_n(\hat{\boldsymbol{\theta}}; a) - \Phi_n^*(\boldsymbol{a})|.$$

*Moreover, define*

$$w_i^*(\boldsymbol{z}, \boldsymbol{a}) := \frac{p^*(\boldsymbol{a} \mid \boldsymbol{x}_i)}{p^*(\boldsymbol{a} \mid \boldsymbol{z})}, \qquad \hat{w}_i(\boldsymbol{z}, \boldsymbol{a}) := \frac{\hat{p}(\boldsymbol{a} \mid \boldsymbol{x}_i)}{\hat{p}(\boldsymbol{a} \mid \boldsymbol{z})},$$

$$\epsilon_{az}(\boldsymbol{a}) := \sup_z |\hat{p}(\boldsymbol{a} \mid \boldsymbol{z}) - p^*(\boldsymbol{a} \mid \boldsymbol{z})|, \quad \epsilon_{ax}(\boldsymbol{a}) := \frac{1}{n} \sum_{i=1}^{n} |\hat{p}(\boldsymbol{a} \mid \boldsymbol{x}_i) - p^*(\boldsymbol{a} \mid \boldsymbol{x}_i)|.$$

*Then*

$$|\Phi_n(\hat{\boldsymbol{\theta}}; \boldsymbol{a}) - \Phi_n^*(\boldsymbol{a})| \le \underbrace{\frac{2C_m C_{ax}(\boldsymbol{a})}{\kappa(\boldsymbol{a})} \cdot \frac{1}{n} \sum_{i=1}^n \|\hat{p}(\boldsymbol{z} \mid \boldsymbol{x}_i, \boldsymbol{a}) - p^*(\boldsymbol{z} \mid \boldsymbol{x}_i, \boldsymbol{a})\|_{\mathrm{TV}}}_{\textit{distribution/reconstruction error (overlap inflated)}}$$

$$+ \underbrace{\frac{C_{ax}(\boldsymbol{a})}{\kappa(\boldsymbol{a})} \epsilon_m(\boldsymbol{a})}_{\textit{outcome nuisance (overlap inflated)}} + \underbrace{\frac{C_m}{\kappa(\boldsymbol{a})} \epsilon_{ax}(\boldsymbol{a}) + \frac{C_m C_{ax}(\boldsymbol{a})}{\kappa(\boldsymbol{a})^2} \epsilon_{az}(\boldsymbol{a})}_{\textit{weight error (overlap inflated)}}.$$

*Proof.* We analyze the error $|\hat{\Phi}_{n,M}(\boldsymbol{a}) - \Phi_n^*(\boldsymbol{a})|$,

$$|\hat{\Phi}_{n,M}(\boldsymbol{a}) - \Phi_n^*(\boldsymbol{a})| \le \underbrace{|\hat{\Phi}_{n,M}(\boldsymbol{a}) - \Phi_n(\hat{\boldsymbol{\theta}}; \boldsymbol{a})|}_{A_1} + \underbrace{|\Phi_n(\hat{\boldsymbol{\theta}}; \boldsymbol{a}) - \Phi_n^*(\boldsymbol{a})|}_{A_2}$$

where $A_1$ represent the the error attributed to Monte-Carlo approximation, and $A_2$ represents the error attributed to estimation of nuisance function.

By Hoeffding's inequality, we have,

$$P(A_1 > \epsilon) \le 2 \exp\left\{ -\frac{nM\epsilon^2}{2C_m^2} \right\}$$

Equivalently, with probability at least $(1 - \delta)$,

$$A_1 \le C_m \sqrt{\frac{2 \log(2/\delta)}{nM}}$$

By (A18), we have

$$p^*(\boldsymbol{z} \mid \boldsymbol{x}_i) = p^*(\boldsymbol{z} \mid \boldsymbol{x}_i, \boldsymbol{a}) \frac{p^*(\boldsymbol{a} \mid \boldsymbol{x}_i)}{p^*(\boldsymbol{a} \mid \boldsymbol{z})} = p^*(\boldsymbol{z} \mid \boldsymbol{x}_i, \boldsymbol{a}) \, w_i^*(\boldsymbol{z}, \boldsymbol{a}),$$

and similarly $\hat{p}(\boldsymbol{z} \mid \boldsymbol{x}_i) = \hat{p}(\boldsymbol{z} \mid \boldsymbol{x}_i, \boldsymbol{a})\hat{w}_i(\boldsymbol{z}, \boldsymbol{a})$. Therefore

$$\Phi_n^*(\boldsymbol{a}) = \frac{1}{n} \sum_{i=1}^n \int m^*(\boldsymbol{z}, \boldsymbol{a}) \, p^*(\boldsymbol{z} \mid \boldsymbol{x}_i, \boldsymbol{a}) \, w_i^*(\boldsymbol{z}, \boldsymbol{a}) \, d\boldsymbol{z},$$

$$\Phi_n(\hat{\boldsymbol{\theta}}; \boldsymbol{a}) = \frac{1}{n} \sum_{i=1}^n \int \hat{m}(\boldsymbol{z}, \boldsymbol{a}) \, \hat{p}(\boldsymbol{z} \mid \boldsymbol{x}_i, \boldsymbol{a}) \, \hat{w}_i(\boldsymbol{z}, \boldsymbol{a}) \, d\boldsymbol{z}.$$

For the term $A_2$, we have

$$A_2 \le \frac{1}{n} \sum_{i=1}^n \left| \int \left( \hat{m} \, \hat{p}_{ia} \hat{w}_i - m^* \, p_{ia}^* w_i^* \right) d\boldsymbol{z} \right|,$$

where $\hat{p}_{ia} = \hat{p}(\boldsymbol{z} \mid \boldsymbol{x}_i, \boldsymbol{a})$ and $p_{ia}^* = p^*(\boldsymbol{z} \mid \boldsymbol{x}_i, \boldsymbol{a})$. Add and subtract $\hat{m} \, p_{ia}^* \hat{w}_i$ to get $A_2 \le B_1 + B_2$ with

$$B_1 := \frac{1}{n} \sum_{i=1}^n \left| \int \hat{m}(\boldsymbol{z}, \boldsymbol{a})\hat{w}_i(\boldsymbol{z}, \boldsymbol{a}) \big( \hat{p}_{ia}(\boldsymbol{z}) - p_{ia}^*(\boldsymbol{z}) \big) \, d\boldsymbol{z} \right|,$$

$$B_2 := \frac{1}{n} \sum_{i=1}^n \left| \int \big( \hat{m}(\boldsymbol{z}, \boldsymbol{a})\hat{w}_i(\boldsymbol{z}, \boldsymbol{a}) - m^*(\boldsymbol{z}, \boldsymbol{a})w_i^*(\boldsymbol{z}, \boldsymbol{a}) \big) p_{ia}^*(\boldsymbol{z}) \, d\boldsymbol{z} \right|.$$

*Bound $B_1$.* By Assumption A11,

$$\|\hat{w}_i(\cdot, \boldsymbol{a})\|_\infty \le \frac{\hat{p}(\boldsymbol{a} \mid \boldsymbol{x}_i)}{\inf_z \hat{p}(\boldsymbol{a} \mid \boldsymbol{z})} \le \frac{C_{ax}(\boldsymbol{a})}{\kappa(\boldsymbol{a})}.$$

Thus

$$B_1 \leq \frac{1}{n} \sum_{i=1}^{n} \|\hat{m}(\cdot, \boldsymbol{a})\|_\infty \|\hat{w}_i(\cdot, \boldsymbol{a})\|_\infty \|\hat{p}_{ia} - p_{ia}^*\|_1 \leq \frac{2 C_m C_{ax}(\boldsymbol{a})}{\kappa(\boldsymbol{a})} \cdot \frac{1}{n} \sum_{i=1}^{n} \|\hat{p}_{ia} - p_{ia}^*\|_{\mathrm{TV}}.$$

*Bound $B_2$.* Decompose $\hat{m}\hat{w}_i - m^* w_i^* = (\hat{m} - m^*)w_i^* + m^*(\hat{w}_i - w_i^*)$. For the first part,

$$\frac{1}{n} \sum_{i=1}^{n} \left| \int (\hat{m} - m^*) w_i^* \, p_{ia}^* \, d\boldsymbol{z} \right| \leq \frac{1}{n} \sum_{i=1}^{n} \|w_i^*\|_\infty \, \mathbb{E}_{p_{ia}^*}[|\hat{m} - m^*|] \leq \frac{C_{ax}(\boldsymbol{a})}{\kappa(\boldsymbol{a})} \, \epsilon_m(\boldsymbol{a}).$$

For the second part, using $|\frac{1}{u} - \frac{1}{v}| \leq \frac{|u-v|}{\inf(u,v)^2}$ and Assumption A11,

$$\begin{aligned}
|\hat{w}_i(\boldsymbol{z}, \boldsymbol{a}) - w_i^*(\boldsymbol{z}, \boldsymbol{a})| &= \left| \frac{\hat{p}(\boldsymbol{a} \mid \boldsymbol{x}_i)}{\hat{p}(\boldsymbol{a} \mid \boldsymbol{z})} - \frac{p^*(\boldsymbol{a} \mid \boldsymbol{x}_i)}{p^*(\boldsymbol{a} \mid \boldsymbol{z})} \right| \\
&\leq \frac{|\hat{p}(\boldsymbol{a} \mid \boldsymbol{x}_i) - p^*(\boldsymbol{a} \mid \boldsymbol{x}_i)|}{\kappa(\boldsymbol{a})} + \frac{C_{ax}(\boldsymbol{a})}{\kappa(\boldsymbol{a})^2} |\hat{p}(\boldsymbol{a} \mid \boldsymbol{z}) - p^*(\boldsymbol{a} \mid \boldsymbol{z})|.
\end{aligned}$$

Therefore,

$$\frac{1}{n} \sum_{i=1}^{n} \left| \int m^*(\boldsymbol{z}, \boldsymbol{a}) (\hat{w}_i(\boldsymbol{z}, \boldsymbol{a}) - w_i^*(\boldsymbol{z}, \boldsymbol{a})) p_{ia}^*(\boldsymbol{z}) \, d\boldsymbol{z} \right| \leq \frac{C_m}{\kappa(\boldsymbol{a})} \epsilon_{ax}(\boldsymbol{a}) + \frac{C_m C_{ax}(\boldsymbol{a})}{\kappa(\boldsymbol{a})^2} \epsilon_{az}(\boldsymbol{a}).$$

Combining the bounds for $B_1$ and $B_2$ gives the stated proxy inequality. $\square$

**Remark A3** (Overlap inflation). *In Theorem A3, the finite-sample bound contains explicit inverse-overlap factors $1/\kappa(\boldsymbol{a})$ and $1/\kappa(\boldsymbol{a})^2$, so for fixed nuisance estimation errors $(\epsilon_m, \epsilon_{ax}, \epsilon_{az})$ the bound worsens as $\kappa(\boldsymbol{a}) \downarrow 0$. In addition, Lemma A6 shows that controlling $\|\hat{p}(\boldsymbol{z} \mid \boldsymbol{x}, \boldsymbol{a}) - p^*(\boldsymbol{z} \mid \boldsymbol{x}, \boldsymbol{a})\|_{\mathrm{TV}}$ can introduce multiplicative dependence on $K(\boldsymbol{a}) = \sup_z p(\boldsymbol{a} \mid \boldsymbol{z})$ (and its estimated counterpart), which is particularly relevant when $A$ is continuous.*

## A6 EXPERIMENTAL SETTINGS

### A6.1 SIMULATED EXAMPLES

#### A6.1.1 MISSING CONFOUNDERS

For case with missing confounders, the hidden layers of the network consists of two modules. The first module takes the treatment variables as input and imputes the latent confounder, and the second module takes the concatenated vector of the imputed confounder and the treatment as input to model the outcome. For separable confounding and non-separable scenario, the first module contains two layers with size 32 and 6, and the second layer contains two layers with size 8 and 4. The variance of the noise term $\boldsymbol{e}_z$ and $\boldsymbol{e}_y$ in (4) are set as $10^{-5}$ and $10^{-3}$, respectively. The training consists of three stages - pre-training, training, and finetuning after pruning, with epochs being 100, 500, and 100, respectively. The network is trained like a plain vanilla DNN for pre-training and training, but the decay of imputation learning rate $\epsilon_k$ and network parameter learning rate $\gamma_k$ only starts at training. After training, the network is pruned and refined during the fine-tuning stage with smaller learning rate. Finetuning stage is usually optional and doesn't have dramatic improvement to the overall performance.

The initial imputation learning rate $\epsilon$ is set at $5 \times 10^{-4}$ for non-separable confounding and $10^{-3}$ for separable confounding, and decays with $\epsilon_k = \frac{\epsilon_k}{1 + \epsilon_k \times k^{0.95}}$. The initial parameter learning rate $\gamma$ is set as $5 \times 10^{-7}$ and $5 \times 10^{-6}$, for the first module and the second module, respectively, and decays with $\gamma_k = \frac{\gamma_k}{1 + \gamma_k \times k^{0.7}}$. For the mixture Gaussian prior 6, $\lambda_n = 10^{-6}$, $\sigma_0^2 = 10^{-4}$, and $\sigma_1^2 = 10^{-1}$.

A6.1.2 PROXY VARIABLE

For case with proxy variable, the hidden layers of the network consists of three modules. The first module takes the proxy variables as input and imputes the latent confounder, the second module takes the concatenated vector of the imputed confounder as input to model the treatment variable, and the third module takes the treatment variable as input and model the outcome. The first module contains two layers with size 64 and 32, the second layer contains one layer with size 16, and the third layer contains one layer with size 8.

The variance of the noise term $e_z$, $e_a$, and $e_y$ in (12) are set as $10^{-5}$, $10^{-4}$, and $10^{-3}$, respectively. The training consists of three stages - pre-training, training, and finetuning after pruning, with epochs being 50, 100, and 50, respectively.

The initial imputation learning rate are $\epsilon_1 = 10^{-3}$ and $\epsilon_2 = 10^{-4}$, and decays with $\epsilon_k = \frac{\epsilon_k}{1+\epsilon_k \times k^{0.8}}$. The initial parameter learning rates are set as $\gamma_1 = 5 \times 10^{-6}$, $\gamma_2 = 5 \times 10^{-5}$, and $\gamma_3 = 5 \times 10^{-7}$, for three modules, respectively, and decays with $\gamma_k = \frac{\gamma_k}{1+\gamma_k \times k^{0.6}}$. For the mixture Gaussian prior (6), $\lambda_n = 10^{-6}$, $\sigma_0^2 = 10^{-4}$, and $\sigma_1^2 = 10^{-2}$.

A6.2 BENCHMARK DATASET

The network structures for benchmark dataset is similar to proxy variable.

A6.2.1 ACIC

The first module contains two layers with size 64 and 32, the second layer contains one layer with size 16, and the third layer contains one layer with size 8.

The variance of the noise term $e_z$, $e_a$, and $e_y$ in (12) are set as $10^{-5}$, $10^{-4}$, and $10^{-3}$, respectively. The training consists of three stages - pre-training, training, and finetuning after pruning, with epochs being 50, 100, and 50, respectively.

The initial imputation learning rate are $\epsilon_1 = 5 \times 10^{-3}$ and $\epsilon_2 = 5 \times 10^{-4}$, and decays with $\epsilon_k = \frac{\epsilon_k}{1+\epsilon_k \times k^{0.8}}$. The initial parameter learning rates are set as $\gamma_1 = 10^{-6}$, $\gamma_2 = 10^{-5}$, and $\gamma_3 = 10^{-7}$, for three modules, respectively, and decays with $\gamma_k = \frac{\gamma_k}{1+\gamma_k \times k^{0.6}}$. For the mixture Gaussian prior (6), $\lambda_n = 10^{-6}$, $\sigma_0^2 = 2 \times 10^{-4}$, and $\sigma_1^2 = 10^{-2}$.

A6.2.2 TWINS

The first module contains two layers with size 64 and 32, the second layer contains one layer with size 16, and the third layer contains one layer with size 8.

The variance of the noise term $e_z$, $e_a$, and $e_y$ in (12) are set as $10^{-3}$, $10^{-5}$, and $10^{-7}$, respectively. The training consists of three stages - pre-training, training, and finetuning after pruning, with epochs being 100, 1000, and 200, respectively.

The initial imputation learning rate are $\epsilon_1 = 3 \times 10^{-3}$ and $\epsilon_2 = 5 \times 10^{-5}$, and decays with $\epsilon_k = \frac{\epsilon_k}{1+\epsilon_k \times k^{0.8}}$. The initial parameter learning rates are set as $\gamma_1 = 10^{-3}$, $\gamma_2 = 10^{-5}$, and $\gamma_3 = 10^{-10}$, for three modules, respectively, and decays with $\gamma_k = \frac{\gamma_k}{1+\gamma_k \times k^{0.95}}$. For the mixture Gaussian prior (6), $\lambda_n = 10^{-6}$, $\sigma_0^2 = 2 \times 10^{-5}$, and $\sigma_1^2 = 10^{-2}$.

