# OpenReview forum: "Stochastic Neural Networks for Causal Inference with Missing Confounders"
_ICLR.cc/2026/Conference — ICLR 2026 Poster_

### Official Review · Reviewer_mrwL · 2025-10-28

**Soundness:** 2
**Presentation:** 2
**Contribution:** 2
**Rating:** 4
**Confidence:** 5

**Summary:**

This paper proposes CI-StoNet (Confounder Imputation with Stochastic Neural Networks), a deep learning framework for causal effect estimation when confounders are unobserved. The key idea is to model the causal DAG using a stochastic neural network where missing confounders Z are treated as latent variables and imputed from the conditional distribution π(Z|A,Y). The authors use adaptive stochastic gradient Hamiltonian Monte Carlo (SGHMC) to simultaneously impute missing confounders and train the neural networks. They provide theoretical guarantees showing that causal effects remain identifiable even though the missing confounders can only be identified up to loss-invariant transformations. The method is evaluated on simulated data, proxy variable settings, and benchmark datasets

**Strengths:**

1. The paper provides convergence guarantees (Lemma 1) and consistency results (Theorems 1-2) for the proposed approach under sparse deep learning theory, which is a notable contribution.
2. The Markovian structure of CI-StoNet allows modeling diverse causal structures, including multiple causes and various proxy variable settings (Sections 3, A2).
3. The paper makes an interesting observation that causal effects can remain identifiable even when confounders are only identified up to loss-invariant transformations (Remark 3).
4. The paper includes simulation studies with both separable and non-separable confounding, and evaluates on multiple benchmark datasets with comparisons to many baseline methods.

**Weaknesses:**

1. While the paper uses "stochastic neural networks," the role and necessity of stochasticity is not clearly explained. The model (4) appears to be a standard neural network with additive Gaussian noise. Why is this stochastic formulation necessary versus standard deterministic neural networks with probabilistic inference? The connection between stochasticity and identifiability of causal effects needs clarification.
2. Assumption 1(ii): Assuming the true model is exactly a sparse StoNet is very strong and unlikely in practice. This eliminates model misspecification concerns but is unrealistic.
3. Mixture Gaussian prior (Equation 6): The choice of this specific prior with independent components seems arbitrary. No justification is provided for why this particular prior structure is appropriate for neural network weights in causal inference. The sensitivity to hyperparameters (λ_n, σ_0, σ_1) is not investigated.
4. Assumption 2 (Overlap): Requires overlap on the imputed confounder Z, but how can this be verified when Z is unobserved?
5. The paper does not provide clear, verifiable conditions to determine when causal effects are vs. aren't identifiable
6. The relationship between "loss-invariant transformations" of confounders and causal effect identifiability needs more rigorous treatment
7. What happens when Assumption 1(ii) is violated (i.e., true model is not a sparse StoNet)?
8. All experiments use relatively small networks and simple settings
9. Missing ablation studies on key components (e.g., impact of sparsity, SGHMC vs. standard SGD)
10. Computational complexity not discussed. No analysis of computational cost, scalability, or convergence speed compared to baselines.

**Questions:**

1. Why is stochasticity needed and why is it helpful to identify causal effect with missing confounders? Is the stochasticity in Z essential, or just a convenient modeling choice?
2. How do you determine if the causal effect is identifiable? According to Theorem 2, causal effects are identifiable when assumptions 1-2 hold (including assuming true model is a sparse StoNet and conditions of Lemma 1 and Theorem 1 hold). However, these are not practically verifiable. A practitioner cannot check if the true model is a sparse StoNet.
3. How do you determine if confounders (missing) are non-identifiable? The paper claims confounders are non-identifiable due to "loss-invariant transformations" but then states causal effects are identifiable (Theorem 2). Remark 1 mentions equivalence classes, but how does one determine which equivalence class the learned confounder belongs to? Can you provide examples where two different confounder values yield the same causal effect estimate?
4. How realistic is the mixture Gaussian prior assumption? This assumption in Equation 6 is quite strong. Assuming all θ_i are independent is unrealistic for neural network weights, which typically have complex dependencies. Also, why is this two-component mixture appropriate? No justification provided
5. The conditions in (13) are complex. How sensitive are results to the choices of your hyperparameters?

---

> ### Author Response · Authors · 2025-11-24
> **Response to reviewer mrwL - Part 1**
>
> **Q1:**
>
> The development of our methodology is grounded on the philosophy of multiple imputation for missing data. Because any single imputation of missing values can introduce bias into the inference, performing multiple imputations (i.e., preserving the stochasticity in $Z$) is essential for obtaining an unbiased estimate of the causal effect.
>
> **Q2**
>
> We thank the reviewer for the clarification request. First, we would like to emphasize that the true model we assumed for the data is essentially a sparse DNN, since the StoNet is asymptotically equivalent to the DNN as established in Theorem 2.2 of [1]. Our result is not a claim of full nonparametric identifiability, which is known to be unattainable in missing-confounder settings without additional proxy or completeness assumptions. Instead, Theorem 2 establishes model-based identifiability, within the assumed sparse-DNN function class, the causal effect is uniquely determined and consistently estimable. This aligns with the identification strategy used in latent-variable causal methods such as CEVAE and factor-model approaches, where a rich but structured generative model enables identification.
>
> Assumption 2 (SUTVA and overlap) is standard and can be partially assessed empirically through overlap diagnostics. Assumption 1 should be interpreted as a structural modeling assumption. It postulates that the underlying treatment and outcome mechanisms admit a representation within a sparse deep-network class, which can approximate a broad class of nonlinear structural equations. CI-StoNet just leverages this function class as a flexible approximating family.
>
> Our use of the StoNet structure is primarily for theory, its Markovian form and sparse Bayesian parameterization make the missing-data analysis tractable. In practice, users need only specify a sufficiently expressive architecture and verify overlap, while the structural assumptions serve the same role as in other model-based identification results.
>
> **Q3**
>
> In CI-StoNet, the latent confounder $Z$ is not identifiable as a unique point-valued quantity because the model admits many loss-invariant transformations that leave the likelihood and all downstream causal quantities unchanged. A simple example is as follows. Let $W_z$ denote the connection weights from the $Z$-layer to the next hidden layer. Since
>
> $$
> W_z Z = (W_z D)\,(D^{-1} Z) ,= \tilde{W}_z \tilde{Z},
> $$
>
> for any diagonal matrix $D$ with nonzero diagonal entries, the resulting causal effect estimate remains unchanged if we replace $Z$ by $D^{-1}Z$ and $W_z$ by $W_z D$. Depending on the choice of activation functions, many such loss-invariant transformations exist for a DNN.
>
> These transformations define an equivalence class of latent representations that are observationally indistinguishable. The SGHMC algorithm converges to one representative of this class, but every representative induces the same conditional distribution
> $\pi(z | a,\theta_1)$ and thus the same Monte-Carlo causal estimator. Hence, while the latent confounder itself is non-identifiable, the causal effect is identifiable and invariant across all equivalent parameterizations. We will refine the manuscript to make this point clearer.
>
> **Q4**
>
> The mixture Gaussian prior in Eq. (6) is not meant to describe the true data-generating process, but serves as a regularization prior that enables sparse deep learning theory. Following Sun, Song \& Liang (2022), this two-component mixture yields posterior concentration and consistent recovery of sparse neural networks, and thereby supports the validity of downstream statistical inference for DNN models. This is essential for our consistency result in Theorem 1. Related
> developments can be found in Sun, Song \& Liang (2022), and Zhang, Sun \& Liang (2023).
>
> The independence across $\theta_i$ is a standard simplifying choice—its purpose is tractability rather than realism, and the mixture structure still induces global sparsity by separating “active’’ from “inactive’’ weights. Thus, the prior is used for its theoretical guarantees in sparse learning, not as a literal assumption about neural weight dependence.
>
> Reference: [1] Nonlinear sufficient dimension reduction with a stochastic neural network. 36th Conference on Neural Information Processing Systems, (NeurIPS 2022), 2022.

---

> ### Author Response · Authors · 2025-11-24
> **Response to reviewer mrwL - Part 2**
>
> **Q5**
>
> We agree that the conditions in (13) are technical, but they only restrict the hyperparameters to broad admissible ranges rather than requiring precise tuning. Following sparse deep learning theory, we set $\sigma_1^2$ to a moderate constant and $\sigma_0^2$ to be small, with $\lambda_n$ chosen to encourage sparsity.
>
> Empirically, CI-StoNet is not sensitive to moderate perturbations of these values, as the sample size grows, the likelihood dominates the prior, and the mixture prior only needs to maintain a clear separation between small and large weights. The theoretical conditions guarantee posterior consistency, but in practice stability only requires that $\sigma_0 \ll \sigma_1$.
>
> In our experiments, we use $\sigma_1^2 \approx 10^{-2}$ with a much smaller $\sigma_0^2$ (e.g. $10^{-4}$), which fully satisfies the theory since $\sigma_1$ is a fixed constant and the scale separation is preserved. A smaller $\sigma_1$ makes the slab slightly tighter and lowers the implicit pruning threshold, leading to moderate pruning. This is consistent with spike-and-slab behavior. Overall, the method remains robust across reasonable choices as long as $\sigma_0 \ll \sigma_1$.

---

> ### Comment · Reviewer_mrwL · 2025-11-28
>
> I thank the authors for answering my questions. I'd appreciate it if the authors could address or provide explanations for weaknesses 4-10.

---

> > ### Author Response · Authors · 2025-11-30
> > **Response to reviewer mrwL - Part 4**
> >
> > **W8**
> >
> > We agree that the experiments in the main text use relatively small networks and controlled generative settings. This choice was deliberate: our primary objective was to study bias and coverage in settings where the true confounding structure is known, rather than to optimize large-scale predictive accuracy. Importantly, the simulated settings do consider some complex scenarios. Section 2.3 includes highly nonlinear outcome and treatment mechanisms under both separable and non-separable confounding (the latter being particularly challenging), and Section 3.2 also evaluates nonlinear proxy relationships. Thus, while the networks are small, the underlying causal mechanisms are complex. The CI-StoNet framework itself is not limited to small architectures. Training uses standard SGD/backpropagation, and the SGHMC-based latent-variable updates scale similarly to those in modern deep latent-variable models.
> >
> > **W9**
> >
> > We appreciate the reviewer’s request for ablation studies. In CI-StoNet, sparsity serves primarily to control model capacity and enable the consistency guarantees; empirically, we find performance to be stable across a wide range of sparsity levels rather than sensitive to fine tuning. We will include a sensitivity analysis in the revision illustrating how varying the sparsity prior affects estimation accuracy.
> >
> > Our algorithm alternates between (1) latent substitute confounder imputation, and (2) network parameter updates. SGHMC is used only for the first part, where sampling from $p(Z | A,Y)$ is required and standard SGD is not applicable. For the network parameters, we deliberately use standard SGD instead of a full MCMC scheme, since SGD is simple, scalable, and effective for optimizing the CI‑StoNet objective. We want to emphasize that using SGHMC is not an optional design choice. CI-StoNet fundamentally relies on latent-variable sampling to impute the substitute confounder $Z$. Standard SGD cannot perform this step, and removing SGHMC would break the underlying missing-data formulation that the method is built upon.
> >
> > **W10**
> >
> > Each iteration of Algorithm 1 consists of (i) an SGHMC-based imputation step for the latent substitute confounder $Z$, which requires a single mini-batch gradient of the log joint density with respect to $Z$, and (ii) standard mini-batch SGD updates for the network parameters. The per-iteration computational cost is therefore $O(BK)$, where $B$ is the mini-batch size and $K$ the number of network parameters—essentially the same order as training a standard feed-forward network, aside from a small constant factor for the additional imputation pass. Overall runtime scales linearly in both $n$ and $K$, making CI-StoNet comparable in complexity to conventional deep latent-variable models such as VAEs.
> >
> > Regarding convergence, Lemma 1 establishes that, under standard decaying learning rates, the iterates $\theta^{(k)}$ converge in mean square to a solution of the estimating equation (7), providing a theoretical rate for the optimization dynamics of Algorithm 1.
> >
> > To give a concrete view of practical scalability, we will add to the appendix a small table reporting wall-clock training times for CI-StoNet and the baselines on our benchmark datasets. These results show that CI-StoNet trains efficiently and remains computationally practical in the settings we study.

---

> ### Author Response · Authors · 2025-11-30
> **Response to reviewer mrwL - Part 3**
>
> **W4**
>
> First, we want to emphasize that overlap is never fully testable, even when all confounders are observed. For example, the usual condition
>
>   $$
>       0<P(A=a|X=x)<1, \ \text{for all $x$ in the support of $X$ with $p_X(x)>0$.}
>   $$
>
> Even if $X$ is observed, we only see a finite sample of $x$-values and empirical frequencies. So we can detect violations, but cannot prove that the overlap assumption holds for every $x$ in the support.
>
> In our setting, Assumption 2 concerns the substitute confounder learned by CI-StoNet. Although $Z$ is latent, we can still perform robustness diagnostics by sampling from its posterior. Given that we have a overlap threshold $0 < \alpha <1$, and a propensity score model $\hat{e}(A|Z)$, A simple stress test is:
> 1. For each unit $i$, draw  $Z_i^{(b)} \sim p_{\hat{\theta}}(Z_i | A_i, Y_i)$, \ b=1, \dots, B
> 2. For each draw, compute propensities and a summary statistic of overlap
>
> $$
>     S_{\alpha}^{(b)} = \frac{1}{n} \sum_{i=1}^n \mathbf{1} \\{ \hat{e}_i^{(b)} < \alpha \ \text{or} > 1-\alpha \\}
> $$
>
> 3. Report the mean $\overline{S_{\alpha}} = \frac{1}{B} \sum_{b} S_{\alpha}^{(b)}$ as diagnostic.
>
> This diagnostic does not “verify” Assumption 2, which is theoretically untestable, but it allows practitioners to detect violations or fragility of overlap under posterior draws of $Z$, analogous to how overlap is assessed with fully observed covariates.
>
> **W5**
>
> We thank the reviewer for raising this point. Our statement that “under mild conditions, the causal effect remains identifiable through CI-StoNet” refers specifically to Assumptions 1–2 in Section 2.1. Assumption 1 encodes the structural causal model
>
> $$
>     A=g_1(Z,e_a), \ Y=g_2(Z,A)+e_y
> $$
>
> together with the independence conditions ensuring strong ignorability given the substitute confounder $Z$, and the requirement that this mechanism lies within the sparse StoNet function class. Assumption 2 adds SUTVA and overlap with respect to $Z$.
>
> Under these conditions, the causal effect $\tau(a) = E[Y(a)]$ is model-identified: any two parameter values that generate the same joint distribution of $(A,Y)$ necessarily yield the same value of $\tau(a)$. Although the latent $Z$ and network parameters are identifiable only up to loss-invariant transformations, $\tau(a)$ is invariant to these transformations, so the causal estimand is uniquely determined by the observed distribution. Identification can fail when these assumptions fail—for example, if treatment assignment becomes nearly deterministic in some regions (violating overlap), or if the true mechanisms cannot be approximated by a StoNet of the form (4).
>
> As in all observational causal inference, these assumptions are partly structural and cannot be verified from finite data, but they have observable implications. In the revision, we will add an explicit “Identifiability” subsection that (i) states Assumptions 1–2 as sufficient conditions for identification, (ii) outlines concrete failure modes, and (iii) links these conditions to practical diagnostics such as overlap checks and latent-space sensitivity analyses. These diagnostics do not prove identification but allow practitioners to detect violations that would undermine it.
>
> **W6**
>
> We thank the reviewer for this helpful suggestion. In the revision we will formalize the role of “loss-invariant transformations". In our setting, a reparameterization $(Z, \theta) \to (Z', \theta')$ is loss-invariant if it induces the same joint distribution of the observed variables $(A, Y)$. Equivalently, the observed-data likelihood satisfies
>
> $$
>     p_{\theta}(A, Y) = p_{\theta'}(A, Y)
> $$
>
> Such transformations arise, for example, from invertible mappings $Z' = T(Z)$ together with corresponding adjustments of the network weights.
>
> Under Assumptions 1–2, the causal effect can be expressed as
>
> $$
>     \tau(a) = E[Y(a)] = \int \mu_2 (z,a, \theta_2)p_Z(z)dz
> $$
>
> If $Z'=T(Z)$ is an invertible transformation that preserves the observed likelihood, then by change of variables,
>
> $$
>     \tau(a) = \int \mu_2(T^{-1}(z'), a, \theta_2')p_{Z'}(z')dz',
> $$
>
> so all observationally equivalent reparameterizations yield the same causal functional. Thus, although $Z$ and the network parameters are identifiable only up to such transformations, the causal effect $\tau(a)$ is invariant within each equivalence class and therefore uniquely identified from the observed distribution.
>
> **W7**
>
> In the newly-uploaded version of the paper, we have discussed the class of functions that can be well-approximated by sparse DNN, and demonstrated the approximation/mis-specification error for causal estimand. For more details, please see Appendix A3.5 of the newly-uploaded version.

---

### Official Review · Reviewer_4cfB · 2025-10-30

**Soundness:** 2
**Presentation:** 3
**Contribution:** 3
**Rating:** 4
**Confidence:** 3

**Summary:**

This paper proposes a new latent variable modeling approach called Confounder Imputation with Stochastic Neural Networks (CI-StoNet) to handle missing confounders when using observational data. CI-StoNet leverages stochastic neural networks to jointly model outcomes and unobserved confounders, and employs an adaptive Stochastic Gradient Hamiltonian Monte Carlo (SGHMC) approach for both imputation and parameter learning. Theoretical guarantees are provided for convergence and consistency, and experiments on both simulated and benchmark datasets are conducted to confirm the effectiveness of the proposed method.

**Strengths:**

1. The paper tackles an important problem (i.e., missing confounders) in causal inference and proposes a new neural network-based method (CI-StoNet) that overcomes some limitations of prior latent variable approaches, such as limited applicability to nonlinear models and consistency issues.
2. The authors provide theoretical support for the convergence and consistency of the proposed method.
3. Experiments on both simulated and benchmark datasets are conducted to evaluate the performance of the proposed method.

**Weaknesses:**

1. As noted in the limitations, the method assumes that the underlying causal structure (DAG) is correctly specified, which may be difficult in real applications. Is it possible to provide some experimental results when the DAG is misspecified?
2. The approach involves training deep neural networks with adaptive MCMC, which can be computationally intensive. Complexity analysis or comparison with baselines should be provided.
3. The two types of baselines are inconsistently used in Table 1, S1 and S2. The differences and reasons should be clearly described.
4. The analysis in the paper relies on several technical assumptions, which may not always hold.

**Questions:**

Please refer to Weaknesses 1-3.

---

> ### Author Response · Authors · 2025-11-24
> **Response to Reviewer 4cfB**
>
> **Q1**:
>
> Please see global resonse 2.
>
> **Q2**:
>
> We agree that training DNNs with adaptive MCMC can be computationally intensive. However, this seems unavoidable given the problem we aim to address, both the functional form of the outcome and the confounders are unknown or missing. We therefore require a framework that can simultaneously impute the missing confounders and estimate the outcome function.
>
> The procedure can be simplified, but only at the cost of additional bias. For example, one could assume a specific functional form (e.g., linear or low-order polynomial) for the outcome, thereby sacrificing the universal approximation property of the model class. Alternatively, one could use variational inference to handle missing value imputation, but this again introduces approximation bias.
>
> Finally, we note that the proposed approach is scalable to large datasets. The parameter updates can be carried out with minibatches, and the latent variable (missing confounder) imputations can be performed in parallel. With modern computational resources, the overall computational burden is generally manageable.
>
> **Q3**
>
> The ACIC benchmark experiment included in the Appendix focuses on estimation of ATE, hence we primarily compared CI-StoNet with the baselines designed for ATE estimation. Since ATE estimations can also be derived from CATE estimations, we also ran this experiment for algorithm designed for CATE and calculate the ATEs.
>
> **ATE estimation across 10 ACIC 2019 datasets, where the number in the parentheses represents the standard deviation of the MAE, with additional benchmarks**
>
> | Method            | In-Sample              | Out-of-Sample         |
> |-------------------|-------------------------|-------------------------|
> | **CI-StoNet**     | **0.0669 (0.0166)**     | **0.0709 (0.0133)**     |
> | CMDE              | 0.0802 (0.0166)         | 0.0877 (0.0246)         |
> | CMGP              | 0.1252 (0.0156)         | 0.1349 (0.0170)         |
> | CEVAE             | 0.0773 (0.0152)         | 0.0875 (0.0154)         |
> | GANITE            | 0.1622 (0.0390)         | 0.1747 (0.0425)         |
> | X-Learner-RF      | 0.1720 (0.0257)         | 0.1903 (0.0253)         |
> | X-Learner-BART    | 0.0738 (0.0251)         | 0.0817 (0.0248)         |
> | CFRNet-Wass       | 0.1024 (0.0241)         | 0.1099 (0.0256)         |
> | CFRNet-MMD        | 0.1105 (0.0258)         | 0.1208 (0.0246)         |
>
> **Q4**
>
> The analysis depends on Assumptions 1 and 2. Assumption 2 is regular as used in many other causal inference papers.
> We believe that Assumption 1 generally holds due to the universal
> approximation ability of deep neural networks.
>
> Regarding the assumption that the true model is a StoNet, we would like to emphasize that the true model we assumed for the data is essentially a sparse DNN, since the StoNet is asymptotically equivalent to the DNN as established in Theorem 2.2 of [1]. Our assumption on the StoNet is primarily for its theoretical properties, which can be easily studied under the framework of missing data imputation.
>
> Reference:
> [1] Nonlinear sufficient dimension reduction with a stochastic neural network. 36th Conference on Neural Information Processing Systems, (NeurIPS 2022), 2022.

---

### Official Review · Reviewer_mx4M · 2025-10-31

**Soundness:** 3
**Presentation:** 2
**Contribution:** 2
**Rating:** 4
**Confidence:** 4

**Summary:**

This manuscript proposed to estimate the causal effects with the missing confounder through a stochastic neural network. The authors provide some preliminary theoretical justifications, with some experimental justifications on both synthetic and small real world scenarios.

**Strengths:**

Overall the authors present the idea in a reasonable manner, with reasonable improvement on the performance.

**Weaknesses:**

The assumption of the existence of the underlying stochastic neural networks as inductive bias can be a chick and egg problem on identifiability. Meanwhile, the theoretical justifications are mainly on consistency, instead of convergence rate based on some standard assumptions.

**Questions:**

My questions are mainly two-fold:
* Think about the case that we only have $(A, Y)$, where $A$ is just some binary treatments, then I don't think we can perform reasonable ATE and hence if the conclusion claimed in this manuscript is correct. The only explanation is that we embed some inductive bias in the function class of $\mu$ (similar to the consistency claim in Theorem 2 that depends on $\theta^*$). If that's the case, we hide the missing confounder issue by adding additional identifiability issue on the function class. Is that correct?
* Regarding the theoretical guarantee, if the density ratio $P(A|X)/P(A)$ is negligibly small but not zero, then we will have a extremely bad convergence rate. However, Theorem 2 only provides asymptotic convergence, and I'm generally wondering the dependency of some other quantities like the one I just mentioned. Otherwise it's not a real causal effect estimation solution but just an attempt to directly apply the stochastic network's results to causal effect estimation.

---

> ### Author Response · Authors · 2025-11-24
> **Response to reviewer mx4M**
>
> **Q1:**
>
> Please see global response .
>
> **Q2:**
>
> Please see the newly updated version of paper. We've added our response to the Appendix of the paper (section A3.4, marked in blue).

---

### Official Review · Reviewer_NAnQ · 2025-10-31

**Soundness:** 2
**Presentation:** 3
**Contribution:** 2
**Rating:** 4
**Confidence:** 3

**Summary:**

This paper proposes CI-StoNet, a latent-variable framework that treats missing confounders as stochastic hidden states and jointly models (i) their conditional distribution and (ii) the outcome with two interconnected neural networks. Training alternates between imputing the latent confounders via adaptive SGHMC and updating network parameters, enabling consistent causal-effect estimation even when the confounders themselves are non-identifiable up to loss-invariant transformations. The approach targets complex, nonlinear settings; retains identifiability of causal effects under mild assumptions; and extends naturally to multiple-cause and proxy-variable scenarios. The authors prove convergence of the adaptive SGHMC estimator and consistency of the resulting causal-effect estimates, then demonstrate state-of-the-art accuracy on simulations and benchmarks (ACIC 2019, Twins). They note limitations around DAG specification (e.g., mediators/colliders) and current lack of explicit uncertainty quantification, and outline how CI-StoNet’s Markovian structure can be adapted to broader proxy-based causal graphs.

**Strengths:**

- The paper offers a strong conceptual framing for missing confounders by treating them as latent states within a StoNet and jointly learning their distribution with treatment and outcome modules.
- The proposed method reaches state-of-the-art ATE accuracy on ACIC-2019 and performs strongly on the Twins dataset; performance remains comparatively stable even when a key confounder is removed, indicating robustness of the latent-imputation mechanism.

**Weaknesses:**

1/ Your inference of $Z$ conditions on each datum’s $(A_i,Y_i)$. When both variables are binary while $Z$ is multi-dimensional, recovering $Z$ from $(A,Y)$ alone is under-determined; the latent confounders are generally non-identifiable without extra structure or additional observables. Could you formalize when $(A,Y)$ contain \emph{enough} information about $Z$?

2/ You state that causal effects remain identifiable “under mild conditions” even if $Z$ is only identifiable up to loss-invariant transformations. Please state these conditions precisely for the binary–binary case and add guidance about what fails if they are violated. Under what restrictions on $g_1,g_2$, noise, or overlap does $E[Y(a)]$ remain identifiable despite non-identifiable $Z$?

3/ Because $D_i=\{A_i,Y_i\}$ drives the imputation of $Z_i$, the method’s success hinges on how informative $(A,Y)$ are about $Z$. Please articulate minimum “richness” assumptions (beyond overlap) under which the estimator is well-posed—e.g., conditions ensuring that $\pi(Z\mid A,Y)$ is sufficiently concentrated to support accurate effect estimation.

4/ You provide proxy-variable extensions for “outcome-depends-on-proxy” and “treatment-depends-on-proxy.” It would help to add practical criteria or diagnostics for when $(A,Y)$ are too sparse and $X$ must be incorporated (e.g., signal tests or ablations showing failure without $X$). Could you include a short decision guideline indicating when to introduce proxies?

5/ The convergence of adaptive SGHMC is asserted under specific conditions and in a reduced parameter space (due to loss-invariant transformations). In practice, what diagnostics should users monitor to assess mixing/convergence and to detect cases where $Z$ is poorly informed by $(A,Y)$? Provide a checklist (e.g., effective sample size, potential scale reduction, stability of $E[Y(a)]$ across chains).

6/ You already conduct a robustness test by removing a key confounder in \emph{Twins}. Please add a complementary experiment where $A$ and $Y$ are binary and $Z$ is high-dimensional, to quantify: (i) degradation in effect estimates, and (ii) the (non-)recoverability of $Z$. How does performance change as $\dim(Z)$ increases under fixed binary $(A,Y)$?

7/ Your ATE estimator is a Monte Carlo average over imputed $Z$, which suggests posterior uncertainty is available. Please report credible intervals (or bootstrap CIs) for $\widehat{E}[Y(a)]$ and $\widehat{\tau}$, and compare coverage when $(A,Y)$ are low-information versus when proxies $X$ are added.

8/ Given the potential information bottleneck when $(A,Y)$ are discrete, practical users would benefit from sample-size versus complexity guidance. Please provide rules of thumb linking $n$, $\dim(Z)$, and network sparsity/width needed to stabilize effect estimation.

**Questions:**

Please refer to the questions in Weaknesses.

---

> ### Author Response · Authors · 2025-11-24
> **Response to reviewer NAnQ - Part 1**
>
> **Q1**:
>
> Please see Global response 1.
>
> **Q2**:
>
> As noted in our response to Question 1, in the binary–binary setting with unmeasured confounding the causal effect is not nonparametrically identifiable from the observed distribution alone. CI-StoNet does not overcome this limitation. Instead, CI-StoNet introduces enough parametric/structural restrictions to partially resolve the non-identification problem for causal effects, but only within the assumed model class. Our guarantees concern model identifiability, within the assumed sparse DNN family, the causal functional $E[Y(a)]$ is uniquely determined (up to loss-invariant symmetries) by the observed data distribution.
>
> The conditions in Assumption 1 impose a restricted, finite-dimensional StoNet parameterization for the treatment and outcome mechanisms. Together with the overlap condition in Assumption 2, these conditions provides guarantees for model identifiability. Specifically,
> 1. noise and independence (Assumption 1(i)) that ensures Markov factorization of the StoNet model.
> 2. sparse DNN structure (Assumption 1(ii). The true $g_1$ and $g_2$ lie in the finite-dimensional sparse DNN class, with sparsity $m_n = o(n)$. This restricts the treatment and outcome mechanisms to a parameterized family.
>
> Under these conditions, we have:
> 1. Parameter identifiability within the model class (up to loss-invariant transformations). Two parameter values $\theta$ and $\theta'$ that generate the same joint distribution $p(A, Y )$ (up to loss-invariant transformations) necessarily induce the same functions $g_1(z) = P(A=1|Z=a)$ and $g_2(z, a) = P(Y=1|A=a, Z=z)$ within the StoNet family.
> 2. Causal functional identifiability. Because StoNet parameters are identifiable up to transformations that preserve the induced conditional distributions, the causal estimand $E[Y(a)] = \int g_2(z, a)p(z) dz $ is also uniquely determined within this model class.
> 3. What would fail without these conditions. If Assumption 1(ii) is relaxed, hence allows arbitrary, nonparametric $g_1$ and $g_2$, then binary $(A, Y)$ provides too little information, and distinct latent structures can produce the same $p(A,Y)$ while implying different estimations of causal functional.
> \end{enumerate}
> We will clarify this point in the revision.
>
> **Q3**:
>
> As noted in our responses to Question 1–2, with binary $A$ and $Y$, $Z$ and the causal effect are not nonparametrically identifiable; CI–StoNet's guarantees concern model identifiability within the sparse DNN family.
>
> Let $g_1(z)=P(A=1 |Z=z)$ and $g_2(z,a)=E[Y | A=a,Z=z]$. For richness conditions ensuring that different latent configurations induce distinguishable observable distributions, beyond overlap and the model’s independence structure, we assume (a.e. w.r.t. the law of $Z$):
> 1. variation in treatment: $g_1$ is not constant a.e.
> 2. variation in outcome: for each $a \in $ { $0, 1$ }, $g_2$ is not constant a.e.
> 3. distinguishability: let $\phi(z) = (g_1(z), g_2(z, 1), g_2(z, 0))$, for $z \neq z'$, $\phi(z) \neq \phi(z')$
>
> Condition (3) rules out the issue that distinct latent states yielding the same $\phi(z)$, making $\pi(Z | A,Y)$ informative and the estimator well‑posed. In CI‑StoNet, this is generically satisfied by design, because
> 1. we can use a low‑dimensional latent layer for effective/substitute confounder $Z \in R^r$, with $r \leq 3$
> 2. strictly monotone link (e.g. sigmoid) and  full‑column‑rank linear map for the output heads that produce probabilities $g_1(z)$ and $g_2(z)$.
> Then the Jacobian $\frac{\partial \phi(z)}{\partial z} \in R^{3 \times r}$ has full column rank a.e., by the constant rank theorem, $\phi(z)$ is locally one‑to‑one (a.e.), yielding the desired distinguishability.
>
> **Q4**:
>
> Thanks for the suggestion. While nonparametric causal identification is generally out of reach with binary $A$ and $Y$, practitioners still benefit from model‑based estimation coupled with transparent diagnostics. We therefore report a checklist indicating when
> $(A,Y)$ might contain limited information and proxy $X$ should be incorporated.
>
> 1. $2 \times 2$ cell sparsity check. Screen $A \times Y$ tables with a row-precision rule $\min(n_{1+}, n_{0+}) \geq \frac{0.25}{\delta^2}$ (binomial worst‑case $SE \leq \delta$) and a cell-sparsity rule $\min(n_{ij}) \geq m$ to control SE(log OR)
> 2. extreme prevalence check. $P(A=1) \notin [0.1, 0.9]$ or $P(Y=1) \notin [0.1, 0.9]$.
> 3. check if outcome fit improves or volatility of causal effect estimation stabilizes when adding $X$. Fit two models, one without proxy ($M_0$), and one with proxy ($M_1$), compare on a fold out set. Check calibration of outcome prediction (e.g. Expected Calibration Error) and standard deviation of $\hat{\tau}$.

---

> ### Author Response · Authors · 2025-11-24
> **Response to reviewer NAnQ - Part 2**
>
> **Q5**:
> The convergence of the adaptive SGHMC algorithm can be assessed by examining the behavior of the sequences $\{\theta^{(k)}\}$ and $\{Z^{(k)}\}$. Since $\{\theta^{(k)}\}$ converges to a point, as established in Lemma 1, its convergence is straightforward to verify. For the convergence of $\{Z^{(k)}\}$, we follow the approach developed in Section 4 of Nemeth & Fearnhead (2021), which is based on the kernel Stein discrepancy.
>
> To detect cases where $Z$ is poorly informed by $(A,Y)$, we take a variable-selection perspective. Specifically, we assess the contribution of each component of $Z$ by computing
>
> $$
> \hat{\tau_{z_j}}=\frac{1}{M}\sum_{l=1}^{M}\frac{\partial \mu_2(z^{(l)}, a, \hat{\theta}_2^*)}{\partial z_j}
> $$
>
> analogous to the causal effect estimator given in (11). A multiple hypothesis testing procedure can then be applied to determine the significance of these effects.
>
> **Q6**:
>
> We appreciate the reviewer’s suggestion and fully understand the motivation. In the setting where both $A$ and $Y$ are binary and no proxy variables are available, the latent confounder $Z$ and the causal effect are known to be non-identifiable from the observational distribution, even with infinite data. Since our theoretical guarantees are model-class-based (Assumption 1(ii), Theorems 1–2), an experiment in this regime would not yield interpretable conclusions about recoverability or performance. Instead, we focus our empirical studies on settings where identification is possible (e.g., with proxy variables), so that comparisons are meaningful and aligned with the scope of our theory.
>
> Regarding the latent dimension, CI-StoNet employs a sparsity-inducing prior (eq. (6)), so in practice the model tends to select only latent components supported by the available information. Thus, when $A$ and $Y$ carry limited signal, the learned representation typically collapses to a low-dimensional or nearly degenerate form. This behavior is governed by the model’s regularization rather than by recoverability of the underlying confounder, and we will clarify this point in the revision.
>
> **Q7**:
>
> To derive a bootstrap confidence interval, consider the following computationally-light bootstrap procedure:
> 1. Fit CI-StoNet on the full dataset and save the final pruned parameters. The parameters $\hat{\theta}$ are saved as the warm start for all bootstrap replicates.
> 2. Draw a bootstrap sample of size $n$, i.e., sampling  indices $\{i_1^{b}, \dots, i_n^{b}\} \sim \text{i.i.d } \text{ Uniform}(1, \dots, n)$ with replacement, then construct the bootstrapped dataset $D^{b} = \{(A_{i_{j}^b}, X_{i_{j}^b}, Y_{i_{j}^b})\}_{j=1}^n$.
> 3. Warm-start model initialization. Initialize model parameters for this replicate by copying $\hat{\theta}$.
> 4. Run a short SGD + SGHMC imputation phase to let the parameters to adjust to $D^{b}$.
> 5. Compute bootstrap ATE estimate. Use the trained parameter $\hat{\theta}^b$, calculate $\tau^b = \hat{E}^b[Y(1)] - \hat{E}^b[Y(0)]$.
> For the basic proxy scenario in DAG misspecification experiment (for details please see global response 2), 100 iterations of such bootstrapping are conducted for each of the 10 generated dataset. Note that the true value of ATE is 3.
>
> **Bootstrapped bounds for basic proxy setting**
> | DGP seed | $\hat{\tau}$ | $L_{\text{bootstrap}}$ | $U_{\text{bootstrap}}$ |
> |----------|--------------|-------------------------|-------------------------|
> | 0 | 3.1365 | 3.0897 | 3.1581 |
> | 1 | 3.0057 | 2.9758 | 3.0574 |
> | 2 | 3.0074 | 2.9836 | 3.0720 |
> | 3 | 3.1582 | 3.1066 | 3.2113 |
> | 4 | 2.9718 | 2.9609 | 3.0404 |
> | 5 | 3.0123 | 2.9774 | 3.0801 |
> | 6 | 3.0515 | 3.0263 | 3.1113 |
> | 7 | 3.0556 | 3.0128 | 3.0904 |
> | 8 | 3.1846 | 3.1181 | 3.2119 |
> | 9 | 3.0241 | 2.9787 | 3.0471 |
>
> **Q8**:
>
> Thank you for the thoughtful question. In practice, we recommend increasing the width of the $Z$-layer
> at most sublinearly with the sample size $n$, in line with existing theory on sparse deep learning \citep{SunSLiang2021}. On the other hand, the sparse prior used in training the deep neural networks effectively performs automatic selection of the informative components of $Z$, thereby determining its effective dimension. Consequently, the proposed method should be robust to the choice of the $Z$-layer width, as long as it is sufficiently large.

---

### Author Response · Authors · 2025-11-24
**Global response 1: identifiability in binary-binary case**

We thank the reviewers for highlighting the fundamental identification issue in the binary-treatment/binary-outcome setting. We fully agree that when both $A$ and $Y$ are binary, the observed joint distribution $p(A,Y)$ has too few degrees of freedom to nonparametrically identify either the latent confounder $Z$ or the causal effect without further structure or proxies (Pearl, 2009; Richardson et al., 2015; Miao, Geng \& Tchetgen Tchetgen, 2018; Rissanen \& Marttinen, 2021).

Following Rissanen \& Marttinen (2021), we distinguish causal identifiability (recovering the causal effect from the observational distribution alone) from model identifiability (uniqueness of the causal effect within a restricted latent-variable model class). Our results pertain to the latter. Under Assumption 1(ii), which posits that the true data-generating mechanism lies in a sparse DNN family, Theorem 1 shows that CI-StoNet consistently recovers the parameters up to loss-invariant transformations, and Theorem 2 then guarantees consistent estimation of $E[Y(a)]$ within this model class. Thus CI-StoNet is identifiable with respect to the causal effect under the assumed sparse-DNN model, while not resolving the impossibility of causal identification in the unrestricted nonparametric binary–binary setting. We impose a very general/weak model assumption (based on sparse neural networks) to get model identifiability, and our work should be commendable.

We justify the sparse-DNN assumption as an approximation stance: sparse DNNs of size
$m_n=o(n)$ (with ambient dimension $K_n$ allowed to grow polynomially) approximate broad Hölder and piecewise-smooth function classes at near-optimal rates (Remark 2). This makes the model class both expressive and capacity-controlled.

We also note that in the Twins experiment, the rich set of baseline covariates acts as noisy measurements of underlying risk factors, effectively serving as proxy variables. Thus our empirical evaluation does leverage proxy information, helping mitigate unmeasured-confounding bias, consistent with our theoretical framing.

Finally, achieving full causal identifiability with unmeasured confounding requires additional observed variables satisfying the proximal causal inference conditions (Miao et al., 2018; Tchetgen Tchetgen et al., 2020). CI-StoNet could naturally serve as a flexible learner for the required treatment and outcome bridge functions, potentially enabling both identification and uncertainty quantification. Developing such a proximal extension is an important future research direction but beyond the present scope.

---

> ### Comment · Reviewer_mx4M · 2025-11-24
> **Follow-up question on identifiability**
>
> I personally don’t think hide the identifiability issue with the assumption on the existence of ground truth stochastic networks is a good choice. It will introduce several follow-up questions in practice. For example, what if there are some mis-specification on the function class? In that case, is the causal effect estimation still robust? Furthermore, if there are misspecification error, should we adjusted the prior to make a trade-off between the statistical error and misspecification error? It looks hard to address these concerns under this framework.
>
> I feel the key assumption of this manuscript is that the missing confounders should take effect in relatively simple mechanism and it would be much more clear to provide a function class capacity based analysis to deal with the mentioned issue. Directly assume the existence of stochastic networks make me feel the manuscript more like a methodology paper that can have some implications in practice, but not fully supported in theory.

---

> > ### Author Response · Authors · 2025-11-30
> >
> > We have added an analysis of the approximation error in the Appendix of the newly-uploaded version of our paper, leveraging the sparse deep learning theory developed in [1]. The analysis is located in Appendix A3.5, marked in blue.
> >
> > Specifically, rather than assuming the true data‑generating process is exactly a sparse DNN, we assume the unknown nuisance functions $(m_A, m_Y)$ belong to a function class $\mathcal{F}$ that is well‑approximated by sparse DNNs (e.g. Hölder / piecewise smooth). Sparse DNNs are then used as approximating models. [1] gives rate:
> >
> > $$
> >     \inf_{\eta \in \text{sparse DNN}} ||\mu_{\eta} - m^*||_{L^2(P_0)} \lesssim \omega_n \to 0
> > $$
> >
> > for functions in $\mathcal{F}$.
> >
> > Given this approximation property and the GLM structure (Gaussian / logistic), the [1] arguments plus our StoNet–DNN equivalence imply the existence of a pseudo‑true StoNet parameter $\theta*$ whose induced law $P_{\theta*}$ is close to the true law $P_0$ in KL:
> >
> > $$
> >     \text{KL}(P_0, P_{\theta^*}) \lesssim \omega_n^2
> > $$
> >
> > This is the approximation / mis‑specification error bound for nuisance functions.
> >
> > Define the causal estimand as $\psi(P) = E_P[Y(a)]$. Using the KL bound, Pinsker’s inequality, and a simple decomposition of $\psi(P)$ in terms of $(m_Y, p_Z)$, we propagate the nuisance‑function approximation error to the causal estimand:
> >
> > $$
> >     |\psi(P_0) - \psi(P_{\theta^*})| \lesssim \omega_n
> > $$
> >
> > This is the misspecification bias: even if the model is wrong, as long as the true mechanisms lie in an $\omega_n$-approximable class, the “pseudo‑true” CI‑StoNet only biases the causal effect by order $\omega_n$.
> >
> > Let $\hat{\psi}_n$ be the CI‑StoNet estimator of the causal effect (e.g., posterior mean or plug‑in). The total error splits into approximation (function‑class misspecification) and estimation (finite‑sample/statistical) components:
> >
> > $$
> >     |\hat{\psi_n} - \psi(P_0)| \leq |\psi(P_0) - \psi(P_{\theta*})| + |\psi(P_{\theta*}) - \hat{\psi_n}|
> > $$
> >
> > The first term is approximation/misspecification error, controlled by the sparse‑DNN approximation rate $\omega_n$ and does not depend on the prior. The second term is where the prior matters: through posterior contraction / capacity control.
> >
> >
> > [1] Y. Sun, Q. Song, and F. Liang. Consistent sparse deep learning: Theory and computation. Journal of the American Statistical Association, 117(540):1981–1995, 2022.

---

### Author Response · Authors · 2025-11-24
**Global response 2: DAG mis-specification experiment**

We conducted an experiment where the true data-generating process is basic proxy, treatment-depending proxy and outcome-depending proxy. We used basic proxy to fit the data.

Data-generating procedure:
1. Confounder: $Z = (z_1, z_2, z_3, z_4, z_5) \sim N(0, I_5)$.
2. Proxy: for $j \in $ { $1 \dots, 50$ }, $X_j = h_1(Z) + \epsilon_X$, where $h_1(Z) = \sin(z_1) + 0.5z_2^2 + 0.3(z_3+z_4+z_5)$, $\epsilon_X \sim N(0, 0.5I_{50})$.
3. Treatment: let $h_2(Z) = 0.7z_1 + 0.3z_2 - 0.2z_3$, $g(X) = \frac{1}{d_X} \sum_{j=1}^{d_X} |X_j|$. For outcome depending proxy and basic proxy, $A \sim \text{Bernoulli}(\text{expit}(h_2(Z)))$. For treatment-depending proxy, $A \sim \text{Bernoulli}(\text{expit}(h_2(Z) + 0.5g(Z)))$.
4. Outcome: let $f(Z) = z_1^2 + 0.5z_2z_3$, $\tau(Z) = 3 + 0.5\sin(z_1)$, $q(X) = \frac{1}{d_X} \sum_{j=1}^{d_X} |X_j|$. For treatment-dependent proxy and basic proxy, $Y = f(Z) + A \cdot\tau(Z) + \epsilon_Y$. For outcome-dependent proxy, $Y = f(Z) + A \cdot\tau(Z) + \gamma q(X) + \epsilon_Y$. $\epsilon_Y \sim N(0, 0.5^2)$.

The true ATE is 3. For each scenario, 10 simulation datasets are generated, each dataset contains 2000 training samples, 500 validation samples, and 500 test samples. We use the basic proxy structure (equation 15) to model all three scenarios, where basic proxy scenario provides a correct baseline while outcome-depending and treatment-depending provide demonstration of DAG misspecification. In-sample MAE is calculated across train and validation set, and out-of-sample MAE is calculated across test set.

**Mean Absolute Error of ATE Estimation Under DAG Misspecification**
| Scenario                   | In-Sample MAE      | Out-of-Sample MAE   |
|---------------------------|---------------------|----------------------|
| Basic Proxy               | 0.0682 (0.0193)     | 0.0702 (0.0207)      |
| Outcome-depending proxy   | 0.0705 (0.0188)     | 0.0976 (0.0236)      |
| Treatment-depending proxy | 0.0911 (0.0156)     | 0.1239 (0.0178)      |

---

### Author Response · Authors · 2025-11-30
**Summray of rebuttal**

Again, we thank the reviewers for their thoughtful comments, which have been very helpful in improving the clarity of this paper. During the rebuttal period, we have made the following clarifications and changes:

**Model identification, not nonparametric identification**

We explicitly clarify that with binary treatment and binary outcome, the causal effect is not nonparametrically identifiable from $p(A, Y)$ alone. Our guarantees are model‑based: if the true treatment and outcome mechanisms lie in function classes that can be well‑approximated by sparse DNNs, then CI‑StoNet achieves model identification of $E[Y(a)]$ within this approximating family (up to loss‑invariant transformations of $Z$), without contradicting classical nonparametric impossibility results in the binary–binary setting.

**New theoretical analysis: finite‑sample error with explicit overlap dependence and sparse‑DNN approximation error**

1. We derive a finite‑sample error bound for the ATE under the proxy setting. The bound shows that the finite‑sample estimation error of the ATE is inflated by a factor $\frac{1}{e_{\min}(a,x)}$, where $e_{\min}(a,x) = \min \{ \hat{e}(a,x), e^*(a,x) \}$. Thus, the ATE error shrinks more slowly when overlap is poor, complementing the asymptotic consistency in Theorem 2.

2. Using sparse‑DNN approximation theory, we assume the unknown treatment mechanism $m_A$ and outcome mechanism $m_Y$ lie in function classes that can be approximated by sparse at rate $\omega_n$ by sparse DNNs of size $m_n = o(n)$. We then demonstrate that the approximation/misspecification error for $E[Y(a)]$ is $O(\omega_n)$. Decomposing the total error into estimation error and approximation error, the Gaussian mixture prior mainly controls estimation via posterior concentration rather than changing the pseudo‑true limit.

**New DAG‑misspecification experiment**

We add a simulation study where the true DAG uses (i) a basic proxy, (ii) an outcome‑dependent proxy, or (iii) a treatment‑dependent proxy, but we always fit the simpler basic‑proxy CI‑StoNet model.

**Diagnostics and uncertainty**

We propose diagnostic procedures for overlap, information richness in the binary–binary setting, and bootstrap intervals for the estimated causal effect.

---

### Meta-Review · Area_Chair_jyaE · 2026-01-07

**Summary:**

This paper proposes CI-StoNet, a latent variable framework for causal inference with unobserved confounders. It utilizes stochastic neural networks to jointly model latent confounders and outcomes, employing an adaptive SGHMC algorithm for simultaneous imputation and parameter learning. The authors establish model-based identification within sparse DNN classes and provide consistency guarantees, supported by strong results on ACIC and Twins benchmarks.

**Reviewer Concerns:**

The authors successfully addressed fundamental concerns regarding the distinction between nonparametric and model-based identifiability, and provided new theoretical analyses on finite-sample error and DAG misspecification robustness.

**Reviewer Scores:**

While initial ratings were uniformly 4, the authors provided an exceptionally comprehensive rebuttal—including new theoretical bounds and experiments—that effectively clarified the method’s structural assumptions and robustness.

---

### Decision · Program_Chairs · 2026-01-26

Accept (Poster)